# Kv2.1 mediates spatial and functional coupling of L-type calcium channels and ryanodine receptors in mammalian neurons

Nicholas C Vierra[1,2], Michael Kirmiz[2], Deborah van der List[1,2], L Fernando Santana[1], James S Trimmer[1,2]*

[1]Department of Physiology and Membrane Biology, School of Medicine, University of California, Davis, Davis, United States; [2]Department of Neurobiology, Physiology, and Behavior, University of California, Davis, Davis, United States

**Abstract** The voltage-gated K+ channel Kv2.1 serves a major structural role in the soma and proximal dendrites of mammalian brain neurons, tethering the plasma membrane (PM) to endoplasmic reticulum (ER). Although Kv2.1 clustering at neuronal ER-PM junctions (EPJs) is tightly regulated and highly conserved, its function remains unclear. By identifying and evaluating proteins in close spatial proximity to Kv2.1-containing EPJs, we discovered that a significant role of Kv2.1 at EPJs is to promote the clustering and functional coupling of PM L-type $Ca^{2+}$ channels (LTCCs) to ryanodine receptor (RyR) ER $Ca^{2+}$ release channels. Kv2.1 clustering also unexpectedly enhanced LTCC opening at polarized membrane potentials. This enabled Kv2.1-LTCC-RyR triads to generate localized $Ca^{2+}$ release events (*i.e.*, $Ca^{2+}$ sparks) independently of action potentials. Together, these findings uncover a novel mode of LTCC regulation and establish a unique mechanism whereby Kv2.1-associated EPJs provide a molecular platform for localized somatodendritic $Ca^{2+}$ signals in mammalian brain neurons.

DOI: https://doi.org/10.7554/eLife.49953.001

*For correspondence:
jtrimmer@ucdavis.edu

**Competing interests:** The authors declare that no competing interests exist.

## Introduction

The members of the Kv2 family of voltage-gated K+ (Kv) channels, Kv2.1 and Kv2.2, are among the most abundant and widely expressed K+ channels in mammalian brain neurons (*Trimmer, 2015*). Kv2 channels are present in high-density clusters localized to neuronal somata, proximal dendrites, and axon initial segments (*Trimmer, 1991*; *Du et al., 1998*; *Bishop et al., 2015*; *Kirmiz et al., 2018a*). In hippocampal and cortical neurons, Kv2 channels conduct most of the delayed rectifier K+ current (*Murakoshi and Trimmer, 1999*; *Du et al., 2000*; *Guan et al., 2007*). Detailed studies have revealed the significant influence of neuronal Kv2.1-mediated currents on action potential duration and repetitive firing (e.g., *Du et al., 2000*; *Liu and Bean, 2014*; *Kimm et al., 2015*, etc.). In addition to its important role in modulating intrinsic electrical activity, Kv2.1 serves a non-canonical structural (*i.e.*, nonconducting) function in tethering the plasma membrane (PM) to the endoplasmic reticulum (ER) to form ER-PM junctions (EPJs) (*Fox et al., 2015*; *Bishop et al., 2018*; *Johnson et al., 2018*; *Kirmiz et al., 2018a*; *Kirmiz et al., 2018b*). Although Kv2.1 clustering at EPJs is tightly regulated and independent of K+ conductance (*Kirmiz et al., 2018b*), the physiological impact of concentrating this Kv channel at an EPJ is not known.

In brain neurons, EPJs occupy approximately 10% of the PM surface area, predominantly within the soma and proximal dendrites (*Wu et al., 2017*). By electron microscopy, the ER at many neuronal EPJs appears as a micron-diameter, flattened vesicle less than 10 nm from the PM, a structure

also called a 'subsurface cistern' (*Rosenbluth, 1962*; *Tao-Cheng, 2018*). While the specific functions of neuronal subsurface cisterns remain unclear, in most eukaryotic cells, EPJs represent domains specialized for maintenance of $Ca^{2+}$, lipid, and metabolic homeostasis (*Gallo et al., 2016*; *Chang et al., 2017*).

L-type voltage-gated $Ca^{2+}$ channels (LTCCs) are prominently expressed in neurons throughout the brain (*Catterall, 2011*; *Zamponi et al., 2015*). Their important role in brain is underscored by studies showing genetic variation in the *CACNA1C* gene encoding Cav1.2, the major voltage-sensing and pore forming α1 subunit expressed in brain, is associated with neurodevelopmental, psychiatric and neurological disorders (*Splawski et al., 2004*; *Ferreira et al., 2008*; *Bozarth et al., 2018*). Given their diverse and crucial roles in neuronal function, LTCCs are subjected to multimodal regulation to ensure their activity is coupled to overall cellular state, especially as related to intracellular $[Ca^{2+}]$ (*Lipscombe et al., 2013*; *Hofmann et al., 2014*; *Neely and Hidalgo, 2014*). In both neurons and non-neuronal cells, Cav1.2-containing LTCCs are clustered at specific sites on the PM where they participate in supramolecular protein complexes that couple LTCC-mediated $Ca^{2+}$ entry to specific $Ca^{2+}$ signaling pathways (*Dai et al., 2009*; *Rougier and Abriel, 2016*). In neurons, LTCCs in dendritic spines participate in a complex whose output contributes to short- and long-term synaptic plasticity (*Da Silva et al., 2013*; *Simms and Zamponi, 2014*; *Stanika et al., 2015*; *Wiera et al., 2017*). Neocortical and hippocampal pyramidal neurons and dentate granule cells also have substantial LTCC populations in the soma and proximal dendrites (*Westenbroek et al., 1990*; *Hell et al., 1993*; *Tippens et al., 2008*; *Berrout and Isokawa, 2009*; *Marshall et al., 2011*; *Kramer et al., 2012*) representing the 'aspiny' regions (*Spruston and McBain, 2007*) of these neurons. Many current models of $Ca^{2+}$-dependent activation of transcription factors posit that somatic LTCCs uniquely contribute to transcription factor activation by mediating $Ca^{2+}$ influx within specialized and compartmentalized signaling complexes (*Wheeler et al., 2008*; *Ma et al., 2012*; *Matamales, 2012*; *Wheeler et al., 2012*; *Ma et al., 2014*; *Cohen et al., 2015*; *Yap and Greenberg, 2018*; *Wild et al., 2019*). However, relatively little research has focused on the molecular mechanisms underlying the spatial and functional compartmentalization of the prominent somatic population of LTCCs compared to those on dendrites and at synapses.

Neuronal somata lack PM compartments analogous to dendritic spines, and fundamental questions remain as to how discrete $Ca^{2+}$ signaling events can occur in the absence of such compartmentalization. In many non-neuronal cells, LTCCs are clustered at EPJs that represent specialized microdomains for LTCC-dependent and -independent $Ca^{2+}$ signaling (*Helle et al., 2013*; *Lam and Galione, 2013*; *Henne et al., 2015*; *Burgoyne et al., 2015*; *Gallo et al., 2016*; *Chung et al., 2017*; *Dickson, 2017*). For example, Cav1.2-mediated $Ca^{2+}$ entry is spatially and functionally coupled to ER ryanodine receptor (RyR) $Ca^{2+}$ release channels at EPJs constituting the cardiomyocyte junctional dyad (*Shuja and Colecraft, 2018*). Localized $Ca^{2+}$ release events (spreading <2 μm from the point of origin) called $Ca^{2+}$ sparks arise from clusters of RyRs located in the ER of EPJs and are triggered *via* local $Ca^{2+}$-induced $Ca^{2+}$ release (CICR), a feed-forward phenomenon in which cytosolic $Ca^{2+}$ binding to RyRs triggers their opening (*Cheng et al., 1993*; *Cheng and Lederer, 2008*). As indicated above, EPJs are abundant on neuronal somata (*Wu et al., 2017*), and neuronal somata have prominent LTCC- and RyR-mediated CICR (*Friel and Tsien, 1992*; *Isokawa and Alger, 2006*; *Berrout and Isokawa, 2009*). Localized RyR-mediated $Ca^{2+}$ release events occur in the somata and proximal dendrites of cultured and acute slice preparations of hippocampal pyramidal neurons (*Koizumi et al., 1999*; *Berrout and Isokawa, 2009*; *Manita and Ross, 2009*; *Miyazaki et al., 2012*), but a specific molecular structure underlying these events has not been described.

Given the well-characterized spatial and functional coupling of LTCCs and RyRs at EPJs in myocytes and previous observations of somatodendritic clustering of the LTCC Cav1.2 in hippocampal neurons (*Westenbroek et al., 1990*; *Hell et al., 1993*), our finding that Kv2.1 clusters are often juxtaposed to RyRs previously led us to hypothesize that Kv2.1 channels cluster with LTCCs to form $Ca^{2+}$'micro-signaling domains' (*Antonucci et al., 2001*; *Misonou et al., 2005a*). More recently, heterologously expressed Kv2.1 and Cav1.2 were found to colocalize in dissociated cultured hippocampal neurons (CHNs) (*Fox et al., 2015*). However, the spatial association of Kv2.1 with endogenous LTCCs and RyRs in brain neurons has not been determined. Here, we examined the subcellular distribution of Kv2.1, LTCCs, and RyRs in hippocampal neurons and used an unbiased proteomic analysis of brain tissue to identify LTCCs and RyRs as proteins in close spatial proximity to clustered Kv2.1. Using heterologous cells and CHNs, we investigated the impact of Kv2.1 clustering on the spatial

coupling and functional properties of LTCCs and RyRs. We also defined how the localization and function of LTCCs and RyRs are affected by the loss of Kv2.1 in mouse CHNs lacking Kv2.1. Together, our findings establish a functional interaction between Kv2.1, LTCCs, and RyRs, reveal a significant influence of Kv2.1 in shaping neuronal LTCC activity, and support a critical role for Kv2.1 in the generation of somatodendritic $Ca^{2+}$ signals.

## Results

### Kv2.1 channels spatially associate with LTCCs and RyRs in brain neurons

In mature CHNs, endogenous Cav1.2 channels are distributed to PM-localized clusters on the soma and proximal dendrites, distinct from their punctate localization in the more distal postsynaptic compartments that also contain the scaffolding protein PSD-95 (*Di Biase et al., 2008*) (*Figure 1A*). To investigate the spatial relationship between somatic Kv2.1 and Cav1.2 channels, we examined rat CHNs immunolabeled for Kv2.1 and Cav1.2, and also for Kv4.2 channels, which exhibit more uniform PM localization in CHNs than either Kv2.1 or Cav1.2 (*Shibata et al., 2003*). In CHNs expressing detectable levels of all three immunolabeling signals, presumed to be pyramidal neurons based on their morphological characteristics (*Benson et al., 1994*; *Antonucci et al., 2001*; *Obermair et al., 2003*), we observed clusters of Kv2.1 that were spatially associated with smaller Cav1.2 clusters but not Kv4.2 clusters (*Figure 1B*). Triple immunolabeling for Kv2.1, Cav1.2, and RyRs demonstrated that many of the clusters of spatially associated Kv2.1 and Cav1.2 channels were colocalized with RyRs (*Figure 1C*). We also observed more prominent spatial overlap of Kv2.1, Cav1.2, and RyR immunolabeling in a subset of CHNs (*Figure 1D*). Analysis of Pearson's Correlation Coefficient (PCC) of Cav1.2 and either Kv2.1 or Kv4.2 pixel intensity demonstrated a greater spatial correlation between Cav1.2 and Kv2.1 immunolabeling than that of Cav1.2 and Kv4.2 (*Figure 1E*). While the absolute PCC values indicate that the majority of somatic Kv2.1 and Cav1.2 immunolabeling did not co-occur within the same pixels, our data suggested that a subset Cav1.2 channels could be found in close proximity (if not overlapping) with Kv2.1. In support of a spatial association between Kv2.1, Cav1.2, and RyRs, we determined that there was a positive correlation between the PCC of Kv2.1 and Cav1.2, and the PCC of Cav1.2 and RyRs within the same cell (*Figure 1F*), suggesting that increased association between Cav1.2 and Kv2.1 was also associated with greater spatial coupling of Cav1.2 to RyRs.

To better evaluate the subcellular distribution of LTCCs relative to Kv2.1 clusters, we next performed super-resolution structured illumination (SIM) imaging of immunolabeled CHNs. These images revealed that Kv2.1 clusters often encompassed smaller clusters of Cav1.2 as well Cav1.3 (*Figure 1G–H*). For these super-resolution images, we performed an object-based analysis (rather than a pixel intensity correlation-based measurement such as PCC) to determine whether the localization of somatic Kv2.1 and LTCCs were co-dependent. The approach we used relied on evaluation of the nearest-neighbor distances (NND) of Kv2.1 and Cav1.2 or Cav1.3 cluster centroids and a comparison of these values to the predicted NNDs if Kv2.1 and LTCCs were randomly distributed (*Shivanandan et al., 2013*; *Helmuth et al., 2010*). We found that the spatial distributions of somatic Kv2.1 and Cav1.2 puncta significantly correlated (p<0.001 versus the null hypothesis that the spatial distributions of Kv2.1 and Cav1.2 puncta were independent) and could not be recapitulated in images in which their relative positions had been iteratively randomized in silico. We also observed similar expression patterns of endogenous Cav1.3 and RyRs in CHNs, with Cav1.3 clusters spatially associated with RyR clusters (*Figure 1I*), in agreement with a recent report (*Sahu et al., 2019*). Together, these data suggested a spatial correlation between Kv2.1 and LTCCs.

We next evaluated how phosphorylation-dependent bidirectional changes in Kv2.1 clustering influenced the localization of somatic Cav1.2 and RyRs in rat CHNs. One stimulus that results in dephosphorylation of Kv2.1 and dispersal of Kv2.1 clusters in CHNs is acute elevation in intracellular $Ca^{2+}$ in response to treatment with the excitatory neurotransmitter glutamate (*Misonou et al., 2004*; *Misonou et al., 2006*). In contrast, suppression of neuronal activity with tetrodotoxin (TTX) causes an increase in Kv2.1 phosphorylation and clustering (*Cerda and Trimmer, 2011*; *Romer et al., 2019*). We found that glutamate stimulation of CHNs not only reduced Kv2.1 clustering, but also significantly decreased the colocalization between Cav1.2 and RyRs, decreased the size

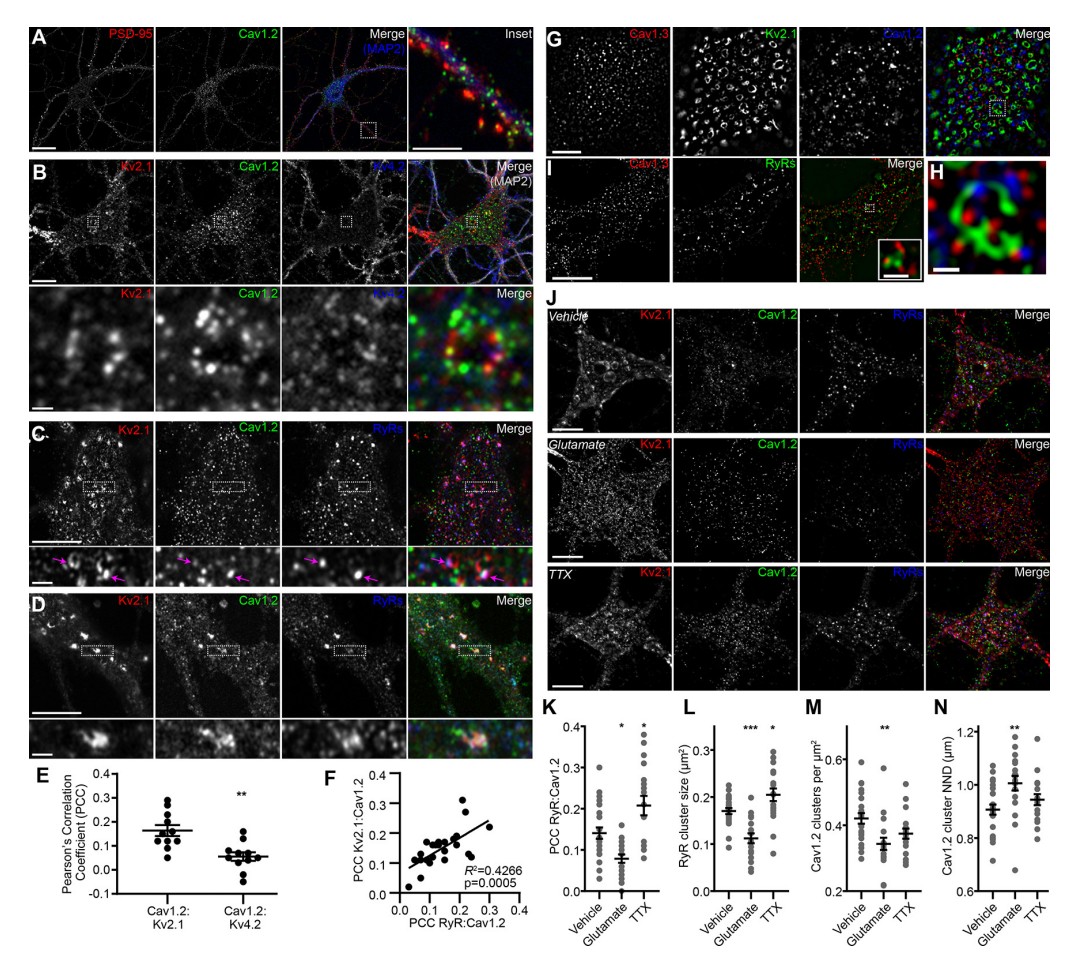

**Figure 1.** Kv2.1 reversibly associates with Cav1.2 and RyRs in cultured hippocampal neurons. (**A**) Single optical section image of a rat CHN immunolabeled for PSD-95, Cav1.2, and MAP2 (scale bar: 20 µm). Note large population of somatic Cav1.2 channels distinct from excitatory synapses located primarily on more distal dendrites. Inset of merged panel shows expanded view of dendritic PSD-95 and Cav1.2 immunolabeling marked by box (inset scale bar: 5 µm). (**B**) Single optical section of the soma of rat CHN immunolabeled for Kv2.1, Cav1.2, and Kv4.2 (scale bar: 10 µm). The row of panels below the main panels shows an expanded view of somatic immunolabeling in the region marked by the box in the main panels (scale bar: 1 µm). (**C**) Single confocal optical section of the soma of rat CHN immunolabeled for Kv2.1, Cav1.2, and RyRs (scale bar: 5 µm). The row of panels below the main panels shows an expanded view of somatic immunolabeling in the region marked by the box in the main panels; arrows indicate selected regions of colocalized Kv2.1, Cav1.2, and RyR immunolabeling (inset scale bar: 1 µm). (**D**) As in E, but in a CHN displaying more prominent colocalization of clustered Kv2.1, Cav1.2, and RyRs. (**E**) Pearson's correlation coefficient (PCC) values of somatic Cav1.2 and Kv2.1 or Kv4.2 immunolabeling (each point represents a single neuron; **p=0.0013; two-tailed *t*-test). (**F**) Scatter plot demonstrating the positive correlation of paired measurements of the PCC values of Kv2.1 vs. Cav1.2 and RyRs vs. Cav1.2 immunolabeling in rat CHNs. (**G**) Super resolution (N-SIM) image of the basal membrane of the soma of a rat CHN immunolabeled for Kv2.1, Cav1.2, and Cav1.3 (scale bar: 5 µm). (**H**) Expanded view of the boxed region in the merged image of G (scale bar: 1.25 µm). (**I**) Super resolution (N-SIM) image of the basal membrane of the soma of a rat CHN immunolabeled for Cav1.3 and RyRs (scale bar: 5 µm). Inset in merged panel shows a higher magnification view of the boxed area (inset scale bar: 0.625 µm). (**J**) Single optical sections of representative rat CHNs treated with vehicle (control), 10 µM glutamate, or 500 nM tetrodotoxin (TTX), and immunolabeled for Kv2.1, Cav1.2, and RyRs (scale bar: 10 µm). (**K–N**) Morphology and spatial distribution of the indicated parameters determined from rat CHNs treated with vehicle, glutamate, or TTX (each point represents one cell; one-way ANOVA followed by Tukey's *post-hoc* test). (**K**) *p=0.0239 (vhl. vs. glut.); *p=0.0134 (vhl. vs. TTX). (**L**) ***p=0.003 (vhl. vs. glut.); *p=0.0407 (vhl. vs. TTX). (**M**) **p=0.0045 (vhl. vs. glut.). (**N**) **p=0.0062 (vhl. vs. glut.).

DOI: https://doi.org/10.7554/eLife.49953.002

of somatic RyR clusters, and increased the distance between somatic Cav1.2 clusters (*Figure 1J–N*). Conversely, suppression of neuronal excitability with TTX produced an effect opposite of glutamate treatment, producing increased spatial coupling between RyRs and Cav1.2, and increasing the size of individual RyR clusters (*Figure 1J–N*). We also found that glutamate stimulation reduced the

number of Cav1.2 clusters present on the PM, consistent with previous observations that acute $Ca^{2+}$ influx results in endocytosis of Cav1.2 channels (*Hall et al., 2013*). Together, these data show that bidirectional changes in Kv2.1 clustering are coupled to corresponding changes in the spatial distributions of Cav1.2 and RyRs on CHN somata.

We next assessed the localization of Kv2.1, Cav1.2, and RyRs in brain sections. Previous immuno-histochemical analyses showed that in hippocampal neurons, Cav1.2 localizes to distinct clusters on somata and proximal dendrites (*Westenbroek et al., 1990*; *Hell et al., 1993*), a spatial pattern similar to that of Kv2.1 (*Trimmer, 1991*; *Scannevin et al., 1996*; *Kirizs et al., 2014*). Similar to previous observations, in low magnification images of mouse and rat hippocampus, we observed Cav1.2 immunolabeling concentrated in CA1 neuron somata, with increasing labeling in area CA2/CA3 neurons, and greatest labeling in dentate gyrus (DG) granule cell somata and dendrites (*Figure 2A–E*). In higher magnification confocal images of DG granule cell bodies in rat hippocampus, we found that Kv2.1 clusters tended to colocalize with Cav1.2 clusters (*Figure 2F*). The somata of rat CA1 pyramidal neurons exhibited a spatial association of Cav1.2, Kv2.1, and RyR immunolabeling that was qualitatively comparable to that seen in CHNs (*Figure 2G*). Similar labeling was observed in high-magnification images of mouse brain sections (*Figure 2H–I*). Kv2.2, which also clusters at EPJs through the same mechanism as Kv2.1 (*Kirmiz et al., 2018b*), similarly colocalized with Cav1.2 immunolabeling in rat CA1 pyramidal cells and DG granule cells (*Figure 2—figure supplement 1A and B*).

## Crosslinking-based proteomic analyses support that Kv2.1 channels are in close spatial proximity to LTCCs and RyRs in brain neurons

We interrogated proteins within the Kv2.1 nano-environment using a crosslinking- and mass spectrometry-based proteomics approach to further determine whether LTCCs and RyRs were in close spatial proximity (having lysine residues within $\approx$ 12 Å of one another) to Kv2.1. We affinity immuno-purified (IPed) Kv2.1 from mouse brain homogenates that were subjected to chemical cross-linking during homogenization. This strategy previously allowed us to identify the ER-resident VAP proteins as Kv2 channel binding partners (*Kirmiz et al., 2018a*). Importantly, we also performed parallel IPs from brain homogenates prepared from Kv2.1 knockout (KO) mice (*Jacobson et al., 2007*; *Speca et al., 2014*) using the same Kv2.1 antibody, to identify proteins IPing in a Kv2.1-independent manner. To further improve the recovery of peptides IPed with Kv2.1, we performed on-bead trypsin digestion, as opposed to the in-gel digestion we had done previously (*Kirmiz et al., 2018a*). Similar to our earlier findings, enriched in the control Kv2.1 IPs (and absent from the Kv2.1 KO brain IPs) were the VAP isoforms VAPA and VAPB (*Table 1*). In addition, among the 50 most abundant proteins specifically present in Kv2.1 IPs (*i.e.*, from WT and not Kv2.1 KO brain samples) were numerous proteins involved in $Ca^{2+}$ signaling and/or previously reported to localize to neuronal EPJs. These included RyR isoforms RyR2 and RyR3, the LTCC $\alpha$ subunits Cav1.2 and Cav1.3, various Cav$\beta$ auxiliary subunits of LTCCs, as well as other proteins involved in $Ca^{2+}$ signaling and homeostasis (*Table 1*). Taken together with our imaging analyses, these findings indicate that Kv2.1 is in close spatial proximity to LTCCs and RyRs at EPJs in mouse brain neurons. We note that while Cav1.2 is the predominant LTCC $\alpha$1 subunit in hippocampus (*Hell et al., 1993*; *Davare et al., 2001*; *Moosmang et al., 2005*; *Lacinova et al., 2008*; *Sinnegger-Brauns et al., 2009*), where its localization on neuronal somata overlaps with Kv2.1, it was not as highly represented in these proteomic analyses as was Cav1.3, perhaps as these analyses were performed on whole brain samples.

## Kv2.1 organizes the localization of LTCCs

Because our immunolabeling and proteomics results indicated that endogenous Cav1.2 channels spatially associated with clustered Kv2.1 in hippocampal neurons, we investigated how the subcellular localization of Cav1.2 (expressed with the LTCC auxiliary subunits $\alpha_2\delta_1$ and $\beta$3) was influenced by the presence of Kv2.1 in heterologous HEK293T cells. HEK293T cells lack endogenous Kv2.1 or Kv2.2 channels (*Yu and Kerchner, 1998*), and have little to no expression of LTCCs (*Berjukow et al., 1996*; *Geiger et al., 2012*). Expression of conducting or nonconducting Kv2 channels in these cells induces EPJ formation (*Fox et al., 2015*; *Bishop et al., 2018*; *Kirmiz et al., 2018a*; *Kirmiz et al., 2018b*). Using total internal reflection fluorescence (TIRF) microscopy to visualize Cav1.2-GFP expressed in HEK293T cells, we observed small Cav1.2 clusters (average area

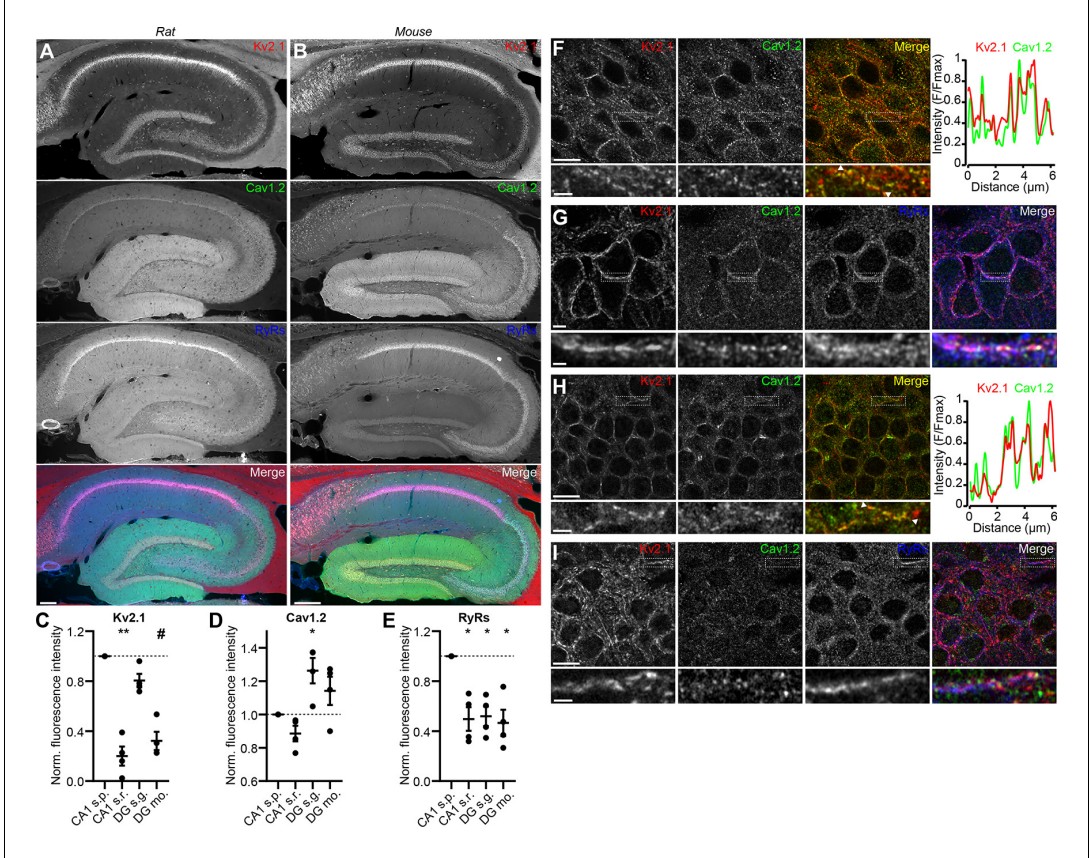

**Figure 2.** Kv2.1 spatially associates with Cav1.2 and RyRs in brain neurons. (**A**) Panels show exemplar images of the hippocampus acquired from a brain section from an adult rat immunolabeled for Kv2.1 (red), Cav1.2, (green) and RyRs (blue), and the merged image (scale bar: 200 µm). (**B**) As in A but acquired from an adult mouse brain section. (**C–E**) Summary graphs of normalized mean fluorescence intensity of Kv2.1, Cav1.2, and RyR immunolabeling from ROIs from various laminae within CA1 (s.p.: *stratum pyramidale*; s.r.: *stratum radiatum*) and DG (s.g.: *stratum granulosum*; mo: molecular layer) in WT mouse brain sections. Fluorescence intensity values were normalized to CA1 s.p. for each mouse. Each point corresponds to an individual mouse (one-way ANOVA followed by Dunnett's *post-hoc* test vs. CA1 s.p.). (**C**) **p=0.0025, #p=0.0573 (**D**) *p=0.0408 (**E**) *p=0.0198 (CA1 s.r.), *p=0.0324 (DG s.g.), *p=0.0107 (DG s.m.) (**F**) Confocal optical section obtained from the dentate gyrus of a rat brain section immunolabeled for Kv2.1 (red) and Cav1.2 (green) (scale bar: 10 µm). The row below the main panels shows expanded views of immunolabeling in the region marked by the box in the main panels; arrowheads indicate region selected for intensity profile line scan (scale bar: 2 µm). Line scan obtained from inset is shown to the right. (**G**) Confocal optical section obtained from the pyramidal cell layer of hippocampal area CA1 in a rat brain section immunolabeled for Kv2.1 (red), Cav1.2 (green), and RyRs (blue) (scale bar: 10 µm). The row below the main panels shows expanded view of immunolabeling in the region marked by the box in the main panels (scale bar: 2 µm). (**H**) As in F but acquired from a mouse brain section. (**I**) As in G but acquired from a mouse brain section.
DOI: https://doi.org/10.7554/eLife.49953.003

The following figure supplement is available for figure 2:

**Figure supplement 1.** Cav1.2 spatially associates with Kv2.2 in brain neurons.
DOI: https://doi.org/10.7554/eLife.49953.004

0.27 ± 0.24 µm²) adjacent to or overlapping with cortical ER, marked by the general ER marker BFP-SEC61β (*Figure 3A,C*). However, in the presence of Kv2.1, the organization of Cav1.2 was dramatically altered, such that Cav1.2 now co-assembled with Kv2.1 into significantly larger clusters (1.05 ± 0.67 µm²) that showed greater colocalization with the ER (as indicated by the PCC of Cav1.2-GFP and BFP-Sec61β) than in the absence of Kv2.1 (*Figure 3B,D–F*). To confirm that these large Kv2.1 clusters were present in the PM, we labeled cell surface Kv2.1 with guangxitoxin-633 (GxTX-633), a membrane impermeant, Kv2 channel-specific toxin conjugated to a fluorescent dye (*Tilley et al., 2014*) (*Figure 3B*). The Kv2.1-induced rearrangement of Cav1.2 was accompanied by an increased occurrence of larger Cav1.2 clusters and a reduced occurrence of smaller Cav1.2 clusters, and a nearly linear relationship between the sizes of Cav1.2 and Kv2.1 clusters (*Figure 3F*).

**Table 1.** LTCC subunits and other Ca$^{2+}$ signaling proteins specifically copurifying with Kv2.1

| Protein | Rank | Mean | SEM (n = 3) |
|---|---|---|---|
| Kv2.1 | 1 | 100.000 | NA |
| Kv2.2 | 3 | 31.638 | 0.518 |
| VAPA | 5 | 25.344 | 1.733 |
| RyR3 | 10 | 12.477 | 0.881 |
| Cavβ4 | 12 | 11.133 | 1.411 |
| VAPB | 15 | 7.600 | 1.393 |
| Cavβ2 | 18 | 5.623 | 0.79 |
| Cav1.3 | 19 | 5.730 | 1.652 |
| Cavβ3 | 23 | 5.070 | 1.033 |
| Hippocalcin | 24 | 4.583 | 0.831 |
| Neurocalcin-delta | 25 | 4.590 | 0.856 |
| SR/ER calcium ATPase 2 | 28 | 4.226 | 2.4 |
| Hippocalcin-like protein 1 | 29 | 4.360 | 0.288 |
| Cavβ1 | 33 | 3.800 | 0.697 |
| Calcineurin catalytic subunit γ | 35 | 3.583 | 0.718 |
| RyR2 | 36 | 3.140 | 0.903 |
| Calcineurin subunit B | 37 | 3.197 | 0.469 |
| Calcium-transporting ATPase | 39 | 2.873 | 0.447 |
| SR/ER calcium ATPase 1 | 40 | 2.530 | 1.21 |
| Cav1.2 | 43 | 2.427 | 0.766 |

Values in table are spectral counts normalized to Kv2.1 over three independent experiments.

DOI: https://doi.org/10.7554/eLife.49953.005

Because TIRF microscopy illuminates subcellular structures that can be up to 100 nm away from the PM, we tested whether the observed co-clustering of Cav1.2 with Kv2.1 was occurring within the PM itself. We performed cell surface immunolabeling of intact cells coexpressing Kv2.1 and a Cav1.2 construct possessing an extracellular hemagglutinin epitope tag [Cav1.2-HA, (*Obermair et al., 2004*). Following cell surface immunolabeling of Cav1.2-HA channels, cells were permeabilized and immunolabeled for total Cav1.2-HA, then imaged by TIRF microscopy (*Figure 3G*). Similar to cells expressing fluorescently tagged channels, we found that cell surface Cav1.2-HA also co-clustered with Kv2.1 (*Figure 3H*). We also determined that cell surface Cav1.2-HA immunolabeling corresponded to approximately 70% of the total Cav1.2 visible in the TIRF field regardless of Kv2.1 coexpression (*Figure 3I*), suggesting that Kv2.1 did not alter the steady-state partitioning of Cav1.2 between PM and intracellular pools. Importantly, similar to results obtained evaluating total Cav1.2, cell surface Cav1.2-HA cluster size was also larger in the presence of Kv2.1, indicating recruitment of cell surface Cav1.2 into larger clusters induced by Kv2.1 (*Figure 3I*). We also found that coexpression with the related but distinct Kv1.5 channel did not impact the clustering of cell surface Cav1.2 channels as did coexpression with Kv2.1 (*Figure 3—figure supplement 1A*). As another measure of the impact of Kv2.1 expression on the organization of PM Cav1.2, we assessed the coefficient of variation (CV: SD/mean) of Cav1.2-HA immunolabeling intensity. The CV is used as a measure of non-uniformity of subcellular distribution, with clustered distributions having high CV values and uniform or diffuse signals having low CV values (*Bishop et al., 2015*; *Jensen et al., 2017*; *Bishop et al., 2018*; *Kirmiz et al., 2018a*; *Kirmiz et al., 2018b*). We found that cells coexpressing Kv2.1 had higher CV values for cell surface Cav1.2 than did cells coexpressing Kv1.5 (*Figure 3—figure supplement 1D*). Cell surface Kv2.1 labeling also exhibited greater colocalization with cell surface Cav1.2 than did cell surface labeling for Kv1.5 (as indicated by PCC values, *Figure 3—figure supplement 1E*).

We next established that the impact of Kv2.1 expression on Cav1.2 clustering did not require Kv2.1 K$^+$ conductance, as coexpression of a K$^+$-impermeable point mutant (Kv2.1$_{P404W}$) (*Lee et al., 2003*; *Kirmiz et al., 2018b*) induced clustering of Cav1.2 comparable to WT Kv2.1 (*Figure 3J–K*).

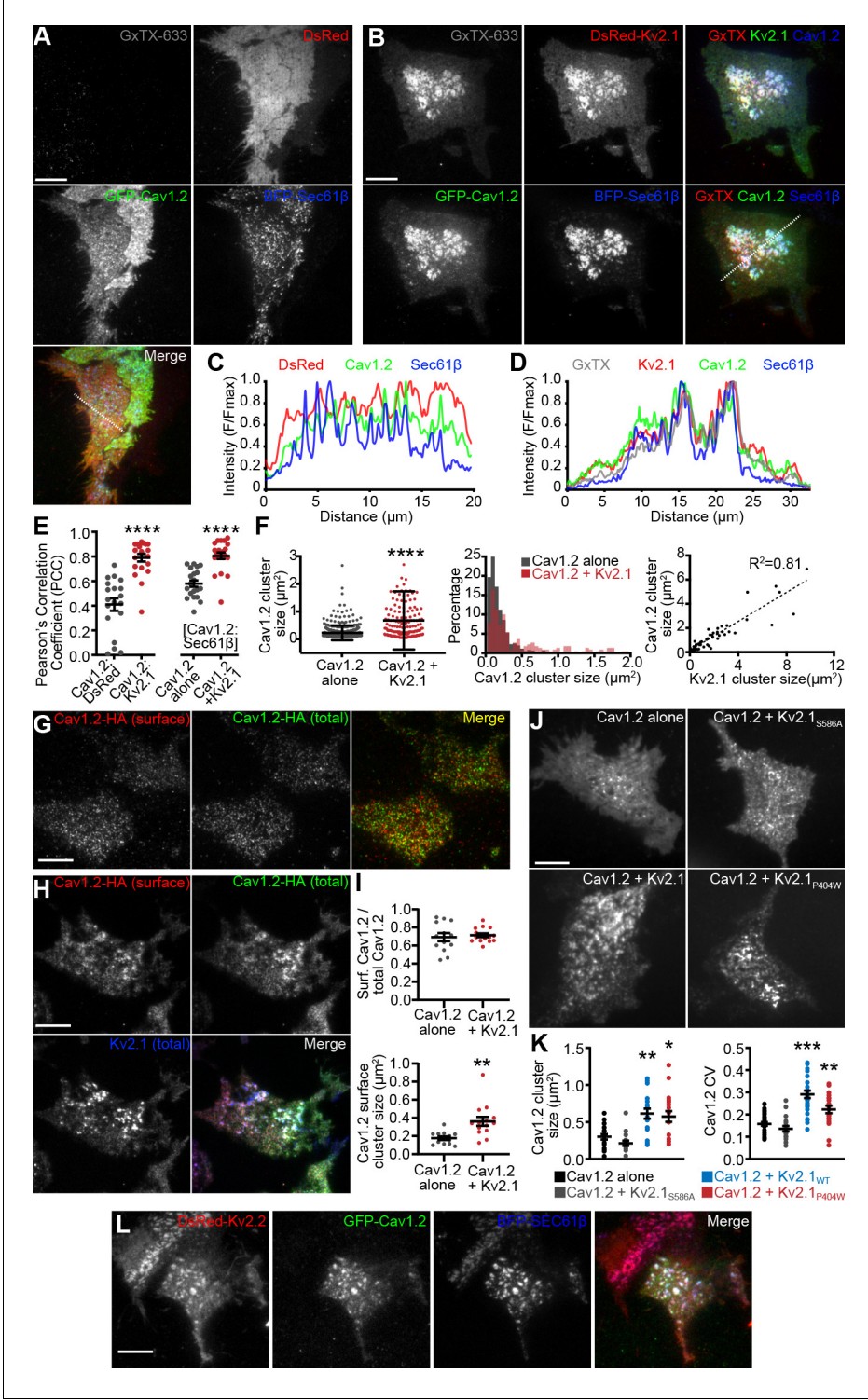

**Figure 3.** LTCCs are recruited to Kv2-induced EPJs. (**A**) TIRF images of a HEK293T cell cotransfected with DsRed (red), GFP-Cav1.2 (green), BFP-SEC61β (blue) and LTCC auxiliary subunits Cavβ3 and Cavα$_2$δ$_1$ (not shown) and labeled with GxTX-633 (scale bar: 10 μm). (**B**) TIRF images of a HEK293T cell cotransfected with DsRed-Kv2.1 (red), GFP-Cav1.2 (green), BFP-SEC61β (blue) and LTCC auxiliary subunits Cavβ3 and Cavα$_2$δ$_1$ (not shown) and labeled with GxTX-633 (scale bar: 10 μm). (**C**) Line scan of pixel intensities from the ROI depicted in the merged image of panel A. (**D**) Line scan of pixel intensities from the ROI depicted in the merged image of panel B. (**E**) Pearson's correlation coefficient (PCC) values of Cav1.2-GFP and DsRed or DsRed-Kv2.1 fluorescence (left) or Cav1.2-GFP

*Figure 3 continued on next page*

*Figure 3 continued*

and BFP-Sec61β with or without DsRed-Kv2.1 (right) (each point represents a single cell; ****p<0.0001; Mann-Whitney test). (F) Summary graphs of Cav1.2 cluster size (left panel), the cluster size frequency distribution (center panel), and a scatter plot of paired measurements of Kv2.1 and Cav1.2 cluster sizes (left panel) measured from HEK293T cells transfected with GFP-Cav1.2, Cavβ3, and Cavα$_2$δ$_1$ alone (black) or additionally cotransfected with DsRed-Kv2.1 (red). Bars are mean ± SD (****p<10$^{-15}$, two-tailed *t*-test, n = 3 cells). (G) TIRF images of a HEK293T cell transfected with Cav1.2-HA, Cavβ3, and Cavα$_2$δ$_1$, and immunolabeled for cell surface Cav1.2-HA (red) and total Cav1.2-HA (green) (scale bar: 10 μm). (H) TIRF images of a HEK293T cell transfected with Cav1.2-HA, Kv2.1-GFP, Cavβ3, and Cavα$_2$δ$_1$, and immunolabeled for cell surface Cav1.2-HA (red) and total Cav1.2-HA (green) (scale bar: 10 μm). (I) Upper panel: ratio of cell surface Cav1.2-HA cluster area versus total Cav1.2-HA cluster area present in the TIRF field obtained from cells expressing Cav1.2-HA and auxiliary subunits with or without Kv2.1 (each point represents one cell; p=0.6755, two-tailed *t*-test). Lower panel: mean area of Cav1.2-HA clusters present in the TIRF field measured from cells expressing Cav1.2-HA and auxiliary subunits with or without Kv2.1 (each point represents one cell; **p=0.0020, two-tailed *t*-test). (J) TIRF images GFP-Cav1.2 in HEK293T cells cotransfected with GFP-Cav1.2, Cavβ3 and Cavα$_2$δ$_1$, either alone or with the non-clustered Kv2.1$_{S586A}$ point mutant, Kv2.1$_{WT}$, or the nonconducting Kv2.1$_{P404W}$ point mutant (scale bar: 10 μm and holds for all panels). (K) Summary graph of Cav1.2 cluster size (left) and coefficient of variation (CV) values of GFP-Cav1.2 fluorescent signal intensity (right) measured from HEK293T cells cotransfected with GFP-Cav1.2 and the indicated Kv2.1 isoforms. Each point corresponds to a single cell. (cluster size: **p=0.0004, *p=0.0017 vs. Cav1.2 alone; CV: ***p<0.0001, **p=0.0040 vs. Cav1.2 alone; one-way ANOVA followed by Dunnett's *post-hoc* test). (L) TIRF images of a HEK293T cell cotransfected with DsRed-Kv2.2 (red), GFP-Cav1.2 (green), BFP-SEC61β (blue) and Cavβ3 and Cavα$_2$δ$_1$ (not shown) (scale bar: 10 μm).
DOI: https://doi.org/10.7554/eLife.49953.006

The following figure supplements are available for figure 3:

**Figure supplement 1.** Kv2.1 increases clustering of surface Cav1.2 channels.
DOI: https://doi.org/10.7554/eLife.49953.007
**Figure supplement 2.** Cav1.3s is recruited to Kv2-induced EPJs.
DOI: https://doi.org/10.7554/eLife.49953.008

Conversely, coexpression with a Kv2.1 point mutant (Kv2.1$_{S586A}$), deficient in clustering (*Lim et al., 2000*) and in inducing EPJ formation (*Kirmiz et al., 2018a*; *Kirmiz et al., 2018b*), had no effect on Cav1.2 clustering (*Figure 3J–K*). We also found that Kv2.2 channels similarly recruited Cav1.2 into large clusters (*Figure 3L*). We also determined that the localization of GFP-tagged Cav1.3 was similarly altered upon coexpression with Kv2.1 or Kv2.2, implying a common mechanism for co-clustering of these two neuronal LTCCs with Kv2 channels (*Figure 3—figure supplement 2A–C*). In contrast, Kv2.1 coexpression did not alter the PM localization of the T-type Ca$^{2+}$ channel Cav3.1 (*Figure 3—figure supplement 2D–F*). This observation suggests that the Kv2.1-mediated spatial reorganization of LTCCs is specific and related to their association with Kv2.1 suggested by our Kv2.1 IP experiments, a notion also supported by the absence of T-type Ca$^{2+}$ channels in these IP experiments. Together, these data demonstrate that clustered but not non-clustered Kv2 channels enhance LTCC clustering and increase their localization to EPJs as a nonconducting function.

## Neuronal Kv2.1 channels functionally associate with endogenous LTCCs and RyRs

Kv2.1 fused to fluorescent proteins such as GFP clusters at neuronal EPJs similar to untagged or endogenous Kv2.1 (*Antonucci et al., 2001*; *Kirmiz et al., 2018a*; *Kirmiz et al., 2018b*). To begin to evaluate Ca$^{2+}$ signals at neuronal Kv2.1-associated EPJs, we fused the genetically-encoded Ca$^{2+}$ indicator GCaMP3 (derived from GFP) to K$^+$-conducting and -nonconducting Kv2.1 channel isoforms and expressed these constructs in rat CHNs. GCaMP3 has previously been used to study near-membrane Ca$^{2+}$ signaling microdomains in astrocytes (*Shigetomi et al., 2010*), and its higher basal fluorescence relative to newer GCaMP variants facilitated identification of transfected neurons. When expressed in HEK293T cells, GCaMP3-Kv2.1$_{WT}$ and GCaMP3-Kv2.1$_{P404W}$ were comparably expressed in surface-localized clusters as reported by both GxTX-633 labeling and GCaMP3 fluorescence (*Figure 4—figure supplement 1*). In rat CHNs, GCaMP3-Kv2.1 exhibited clustered localization similar to other fluorescently tagged Kv2.1 isoforms (*Figure 4A*) and reported global Ca$^{2+}$ spikes, as indicated by the synchronized increase in fluorescence across the PM at sites where the construct was clustered and also in regions with diffuse GCaMP3-Kv2.1 expression (*Figure 4B*, *Video 1*). In

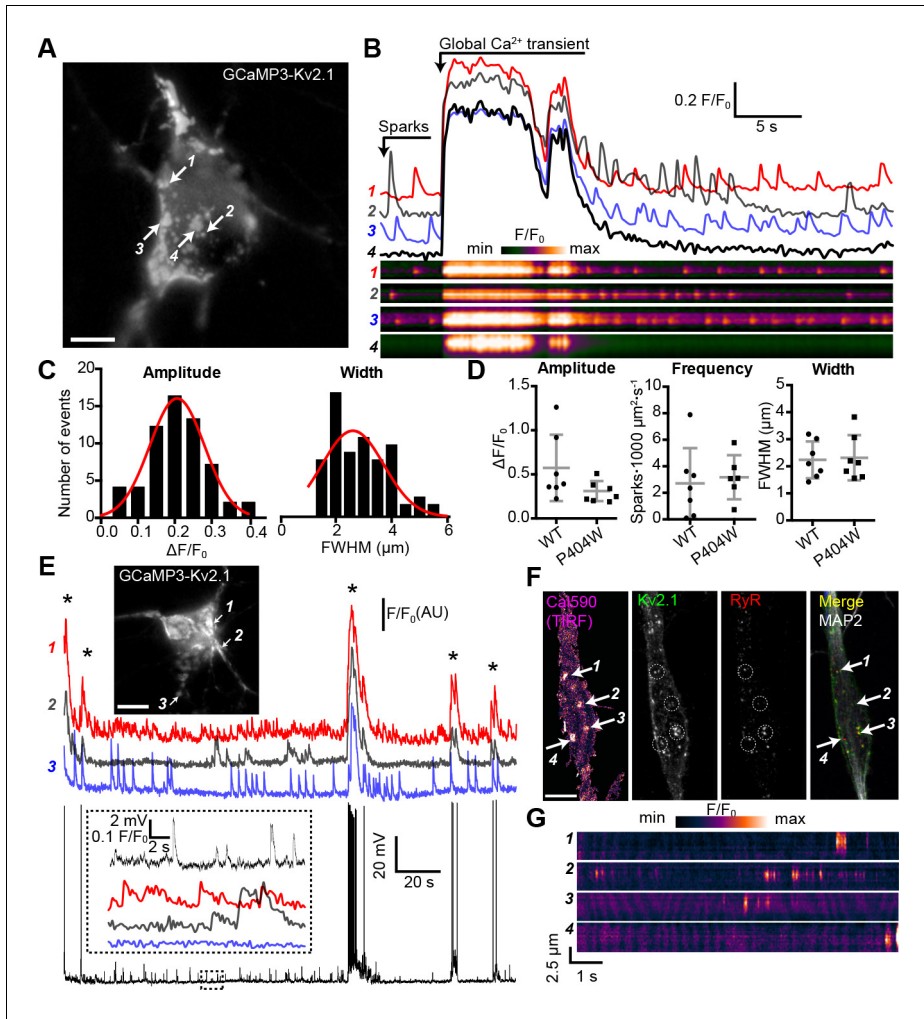

**Figure 4.** Spontaneous $Ca^{2+}$ signals are generated at Kv2.1-associated EPJs. (**A**) Widefield image of a rat CHN transfected with GCaMP3-Kv2.1 (also see *Video 1*). Arrows indicate selected Kv2.1 clusters whose fluorescent intensity profiles are plotted in panel B (scale bar: 10 μm). (**B**) Fluorescence intensity traces (upper panels) and kymographs (lower panels) corresponding to the four ROIs indicated in panel A. Note spontaneous $Ca^{2+}$ signals occurring at ROI 2 that are not detected at the adjacent ROI 4. (**C**) Amplitude ($\Delta F/F_0$) and spatial spread (full width at half maximum, FWHM; μm) of all spatially distinct localized $Ca^{2+}$ signals recorded from the neuron in panel A over a period of 90 s. (**D**) Summary data of the amplitude, frequency and spatial spread (width) of all spatially distinct localized $Ca^{2+}$ signals recorded from CHNs expressing GCaMP3-Kv2.1 or GCaMP3-Kv2.1$_{P404W}$. Each point corresponds to a single cell. No significant differences were detected. Bars are mean ± SD (Student's t -test). (**E**) Image of a rat CHN transfected with GCaMP3-Kv2.1 from which simultaneous GCaMP3-Kv2.1 fluorescence and membrane potential values were acquired (scale bar: 10 μm). Numbered arrows correspond to ROIs whose fluorescence intensity traces are depicted below image. Membrane potential measurements are provided in the bottom trace. The inset shows and expanded view of ROI $Ca^{2+}$ traces and membrane potential values from region of the time course indicated by the dashed box in the membrane potential trace. Asterisks correspond to global $Ca^{2+}$ spikes. (**F**) Representative rat CHN loaded with Cal590 and imaged with TIRF microscopy, followed by *post-hoc* immunolabeling for Kv2.1, RyRs, and MAP2. Arrows indicate ROIs where spontaneous $Ca^{2+}$ signals were detected; dashed circles indicate approximate regions where immunolabeling for Kv2.1 and RyRs was detectable (scale bar: 10 μm). (**G**) Kymograph showing the localized $Ca^{2+}$ release events detected at ROIs depicted in F.
DOI: https://doi.org/10.7554/eLife.49953.009

The following figure supplements are available for figure 4:

**Figure supplement 1.** GCaMP3-Kv2.1$_{WT}$ and GCaMP3-Kv2.1$_{P404W}$ show comparable PM expression in HEK293T cells.
DOI: https://doi.org/10.7554/eLife.49953.010

**Figure supplement 2.** Relationship of $Ca^{2+}$ sparks to global $Ca^{2+}$ spikes.

*Figure 4 continued on next page*

*Figure 4 continued*

DOI: https://doi.org/10.7554/eLife.49953.011

addition to synchronized $Ca^{2+}$ spikes, we also observed rapid and stochastic $Ca^{2+}$ signals occurring at a subset of individual GCaMP3-Kv2.1 clusters within the soma (*Figure 4B–C*, *Video 1*). These $Ca^{2+}$ signals were confined to individual clusters such that the fluorescence of adjacent GCaMP3-Kv2.1 clusters < 1 μm from the active clusters remained stable (*Figure 4B*, compare regions of interest 2 and 4). We found that $Ca^{2+}$ signal amplitude, frequency, and width were insensitive to the $K^+$ conductance of the GCaMP3-Kv2.1 reporter, as $Ca^{2+}$ signals detected by a $K^+$-impermeable variant of this construct (GCaMP3-Kv2.1$_{P404W}$) showed no difference in any of these parameters relative to GCaMP3-Kv2.1 (*Figure 4D*).

Next, we assessed the relationship between GCaMP3-Kv2.1 reported $Ca^{2+}$ signals and membrane potential ($V_m$). We performed current clamp experiments to monitor the $V_m$ and $Ca^{2+}$ signals simultaneously, using the whole-cell perforated patch clamp configuration. Spontaneous action potentials were associated with $Ca^{2+}$ spikes, suggesting that these synchronized, large-amplitude $Ca^{2+}$ transients reflected $Ca^{2+}$ entry through voltage-gated $Ca^{2+}$ channels as well as $Ca^{2+}$ release through RyRs (*Figure 4E*). However, unlike global $Ca^{2+}$ spikes, the localized $Ca^{2+}$ signals displayed no clear relationship with action potentials or other spontaneous $V_m$ fluctuations, similar to previous observations of localized $Ca^{2+}$ release events in CA1 pyramidal neurons (*Berrout and Isokawa, 2009*; *Manita and Ross, 2009*). We also found that the localized $Ca^{2+}$ signals could persist in the presence of TTX (*Figure 4—figure supplement 2A*), and that in some neurons, spark frequency appeared to be elevated immediately following a global $Ca^{2+}$ spike (*Figure 4—figure supplement 2B–C*). Together, these observations suggest that the localized $Ca^{2+}$ signals arose independently of large, uniform fluctuations in the $V_m$ such as those that occur with action potentials.

As heterologous expression of Kv2.1 in CHNs is known to result in large Kv2.1 'macroclusters' that recruit RyRs (*Antonucci et al., 2001*), we next determined whether somatic $Ca^{2+}$ signals occurred at native Kv2.1-associated EPJs. For these experiments, we used non-transfected CHNs loaded with the $Ca^{2+}$ dye Cal-590 AM and recorded $Ca^{2+}$ signals using TIRF microscopy. Using this approach, it was possible to detect spontaneous, localized $Ca^{2+}$ release events in the soma that were qualitatively similar to those recorded with GCaMP3-Kv2.1 (*Figure 4F–G*, *Video 2*), although with faster kinetics in fluorescence intensity changes relative to GCaMP3-Kv2.1, likely reflecting differences in the $Ca^{2+}$ binding properties of these probes. *Post-hoc* immunolabeling of these CHNs for Kv2.1, RyRs, and the neuron-specific cytoskeletal protein MAP2 indicated that the observed localized $Ca^{2+}$ signals occurred primarily within the soma at sites of colocalized Kv2.1 and RyR clusters (*Figure 4F*).

These observations suggested that the $Ca^{2+}$ signals observed at neuronal Kv2.1-associated EPJs reflected RyR-generated $Ca^{2+}$ sparks. To further assess this possibility, we imaged GCaMP3-Kv2.1-expressing CHNs treated with compounds that modulate LTCC- and RyR-mediated CICR. We found that caffeine, which sensitizes RyRs to cytosolic $Ca^{2+}$, enhanced the frequency of localized $Ca^{2+}$ signals (*Figure 5A,B*, *Video 3*). In contrast, depletion of ER $Ca^{2+}$ stores with the sarco-/endo-plasmic reticulum $Ca^{2+}$ ATPase (SERCA) inhibitor thapsigargin led to an elimination of local $Ca^{2+}$ signals (*Figure 5A–B*). The functional coupling of dendritic LTCCs and RyRs in hippocampal neurons has previously been demonstrated by the impact of dihydropyridine (DHP) compounds on dendritic $Ca^{2+}$ sparks: the LTCC agonist Bay K8644 increased $Ca^{2+}$ spark frequency, whereas the LTCC inhibitor nimodipine blocked $Ca^{2+}$ sparks (*Manita and Ross*,

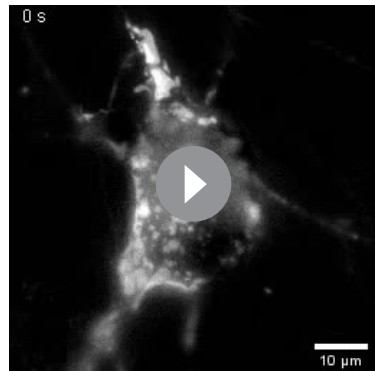

**Video 1.** Spontaneous somatic $Ca^{2+}$ signals detected at GCaMP3-Kv2.1 clusters in rat CHNs. Stack of widefield images of a rat CHN transfected with GCaMP3-Kv2.1 and imaged at 10 Hz.
DOI: https://doi.org/10.7554/eLife.49953.012

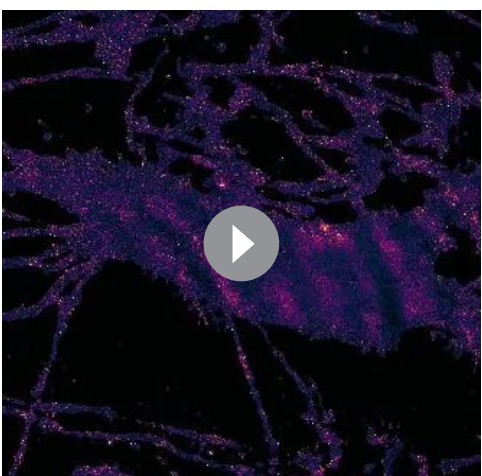

**Video 2.** Spontaneous somatic Ca²⁺ signals detected by TIRF microscopy in rat CHNs loaded with Cal-590 AM. Stack of TIRF images of rat CHNs loaded with Cal-590 AM and imaged at 30 Hz. Regular wave-like signals are a TIRF imaging artifact. Images have been normalized to the first image without detectable Ca²⁺ signals (*i.e.*, F/F$_{min}$).
DOI: https://doi.org/10.7554/eLife.49953.013

2009). Here, we obtained similar evidence of the involvement of LTCCs in the generation of somatic GCaMP3-Kv2.1 reported Ca²⁺ signals. The frequency of these Ca²⁺ sparks was enhanced by activation of LTCCs with Bay K8644 (*Figure 5A,B,D*, *Videos 4* and *5*), while they were rapidly inhibited by blockade of LTCCs with nimodipine (*Figure 5A–B*). We also performed *post-hoc* immunolabeling of these imaged CHNs to determine whether the specific GCaMP3-Kv2.1 clusters which exhibited localized Ca²⁺ signals were associated with RyRs. Using this approach, we determined that the subset of GCaMP3-Kv2.1 clusters that colocalized with RyRs corresponded to the clusters that produced localized Ca²⁺ signals, either spontaneously or in response to the pharmacological modulators caffeine (*Figure 5C*) and Bay K8644 (*Figure 5D*). We also quantified the relationship between the size of *post-hoc* immunolabeled RyR clusters and spark frequency and amplitude. Similar to previous observations in vascular smooth muscle (*Pritchard et al., 2018*) and cardiac muscle cells (*Galice et al., 2018*), we found that neuronal Ca²⁺ spark frequency but not amplitude correlated with RyR cluster size, and that application of the LTCC agonist Bay K8644 steepened this relationship (*Figure 5E*). Taken together, these observations demonstrate that Kv2.1-associated EPJs are sites of spontaneous CICR events mediated by LTCCs and RyRs.

## Kv2.1 augments LTCC and RyR2-mediated CICR reconstituted in HEK293T cells

We next asked how Kv2.1-induced clustering of LTCCs would impact RyR-mediated Ca²⁺ release in HEK293T cells. For these experiments, we expressed Kv2.1 along with Cav1.2, the LTCC auxiliary subunits α$_2$δ$_1$ and β3, RyR2, and the STAC1 adaptor protein, an approach similar to that previously used to recapitulate Cav1.1- and RyR1-mediated Ca²⁺ release in HEK293T cells (*Perni et al., 2017*). We found that in the presence of these auxiliary subunits, Kv2.1, Cav1.2, and RyR2 could spatially associate in HEK293T cells (*Figure 6A*). Thus, the spatial association of Kv2.1, Cav1.2, and RyRs seen in neurons could be recapitulated in HEK293T cells. To detect Ca²⁺ release events, we performed TIRF microscopy of cells loaded with the Ca²⁺-sensitive dye Cal-590 AM. Although it was not possible to establish whether a cell expressed all transfected constructs, we observed spontaneous Ca²⁺ release events in a subset of cells (*Figure 6B,E*) that were not seen in untransfected HEK293T cells and focused our analysis on cells that exhibited this phenotype. These spontaneous Ca²⁺ release events were rapidly blocked by the RyR inhibitor tetracaine (*Figure 6G*, *Video 6*), suggesting that they reflected CICR mediated by RyRs. Expressing Kv2.1 in these cells resulted in enhanced spark frequency and amplitude (*Figure 6C–D,F,J*). Similar results were obtained using Cav1.3 in place of Cav1.2 (*Figure 6—figure supplement 1*).

To better understand the mechanism underlying the influence of Kv2.1 on these reconstituted Ca²⁺ sparks, we next compared how they were affected by the nonconducting Kv2.1$_{P404W}$ and the non-clustering Kv2.1$_{S586A}$ point mutants (*Figure 6I*). By using these Kv2.1 isoforms, we determined that there was an interplay between both Kv2.1 K⁺ conductance and clustering on Ca²⁺ sparks reconstituted in HEK293T cells. Expression of Kv2.1 channels capable of clustered EPJ formation (i.e., Kv2.1$_{WT}$ and Kv2.1$_{P404W}$) increased spark frequency, whereas non-clustering Kv2.1$_{S586A}$ did not (*Figure 6J*). Interestingly, we found that spark amplitude was enhanced by K⁺-conducting Kv2.1$_{WT}$ but not Kv2.1$_{P404W}$ (*Figure 6J*), suggesting that while Kv2.1-mediated clustering alone was sufficient to impact spark frequency, K⁺ conductance was required to impact the amplitude of reconstituted

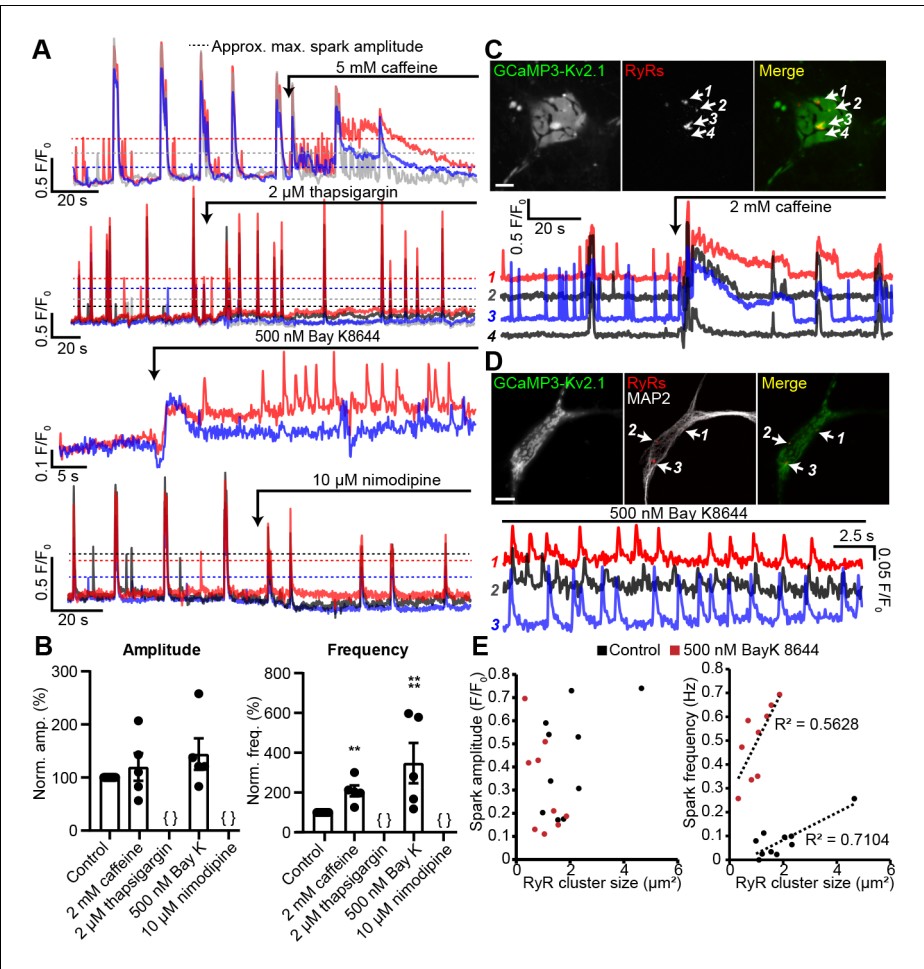

**Figure 5.** Spontaneous $Ca^{2+}$ signals at Kv2.1-associated EPJs are produced by RyR- and LTCC-mediated CICR. (**A**) Representative GCaMP3-Kv2.1 fluorescence traces from CHNs treated with pharmacological probes of CICR. Different colors indicate spatially distinct ROIs within the same neuron. Dashed lines indicate approximate maximum amplitude for localized $Ca^{2+}$ signals as opposed to the larger amplitude, synchronized global $Ca^{2+}$ transients that exceed the dashed lines. (**B**) Summary data of the amplitude and frequency of all sparks recorded from CHNs treated with pharmacological probes of CICR. Each point corresponds to a single cell (**p=0.0013 vs. control; ****p<0.0001 vs. control; {}: no $Ca^{2+}$ sparks detected; one-way ANOVA followed by Dunnett's test). (**C**) Image of rat CHN transfected with GCaMP3-Kv2.1 and treated with caffeine, followed by *post-hoc* immunolabeling for RyRs (scale bar: 10 µm). Numbered arrows indicate ROIs where localized $Ca^{2+}$ signals were detected (ROIs 1–3) or not detected (ROI 4). ROI fluorescence traces are shown in lower panel; note lack of spontaneous $Ca^{2+}$ signals at ROI 4 despite its proximity to ROI 3, which displays prominent spontaneous $Ca^{2+}$ release. (**D**) As in panel A, except CHN was treated with 500 nM Bay K8644 to induce spontaneous $Ca^{2+}$ signals (scale bar: 10 µm). (**E**) Plot of individual RyR cluster size (determined from *post-hoc* immunolabeling) versus its spark amplitude (left panel) or frequency (right panel) reported by GCaMP3-Kv2.1 fluorescence in control (black symbols) or Bay K8644-treated (red symbols) cells. Each point corresponds to an individual RyR cluster (*n* = data from 4 cells [control] or 5 cells [Bay K8644]).

DOI: https://doi.org/10.7554/eLife.49953.014

$Ca^{2+}$ sparks in HEK293T cells. We hypothesize that the high input resistance of HEK293T cells relative to CHNs, the latter of which possess numerous endogenous ionic conductances (including native Kv2.1 channels), enabled $K^+$ conductance through Kv2.1$_{WT}$ and Kv2.1$_{S586A}$ to promote $V_m$ hyperpolarization in HEK293T cells, maintaining a greater electrochemical driving force for extracellular $Ca^{2+}$ and also promoting recovery of Cav1.2 channels from voltage-dependent inactivation. In conclusion, these observations indicate that Kv2.1-mediated clustering promotes the functional coupling of Cav1.2 and RyRs.

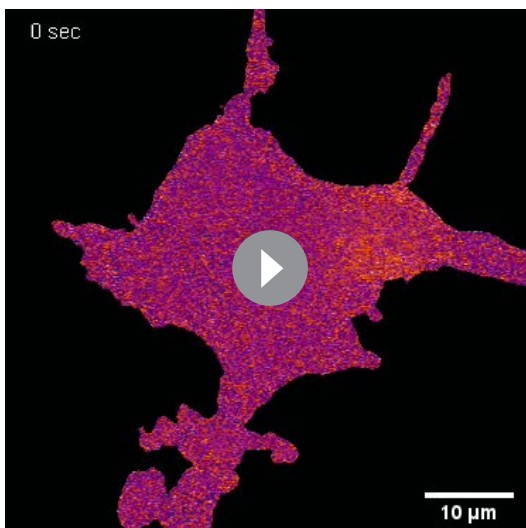

**Video 3.** Caffeine increases the frequency of somatic Ca$^{2+}$ sparks in CHNs. Images of a rat CHN transfected with GCaMP3-Kv2.1 acquired at 5 Hz. Neuron is treated with 5 mM caffeine at t = 84 s; the increased Ca$^{2+}$ spark frequency is apparent from t = 87 s-101s. Images have been normalized to the first image without detectable Ca$^{2+}$ signals (*i.e.*, F/F$_{min}$).
DOI: https://doi.org/10.7554/eLife.49953.015

## Kv2.1 reduces the voltage threshold for Cav1.2 opening and enhances LTCC activity

Having demonstrated a spatial and functional association of Kv2.1, LTCCs, and RyRs in hippocampal neurons that could be reconstituted in HEK293T cells, we next investigated whether clustering by Kv2.1 influenced the Cav1.2-mediated LTCC activity. As physical interactions between adjacent LTCCs promote enhanced LTCC activity (reducing the membrane voltage threshold for channel opening and elevating channel open probability) (*Navedo et al., 2005*; *Dixon et al., 2012*; *Moreno et al., 2016*), we reasoned that this functional property of Cav1.2 might be enhanced by Kv2.1-induced clustering. To test this possibility, we obtained whole-cell patch-clamp recordings from HEK293T cells transfected with Cav1.2 and the clustered but non-K$^+$ conducting Kv2.1$_{P404W}$ point mutant, which allowed us to measure Ca$^{2+}$ currents ($I_{Ca}$) in the absence of the very large outward K$^+$ currents produced by Kv2.1$_{WT}$. Consistent with an influence of Cav1.2 spatial organization on its activity, we found that expression of Cav1.2 with Kv2.1$_{P404W}$ enhanced peak $I_{Ca}$ as compared to cells expressing Cav1.2 alone (*Figure 7A–B*). Analysis of the conductance-voltage (*G-V*) relationship also showed an influence of Kv2.1 on the $V_m$ threshold for Cav1.2 opening, with currents produced by Cav1.2 activating at more negative voltages in the presence of Kv2.1$_{P404W}$ than those produced by Cav1.2 alone, with no effect on steady-state inactivation (*Figure 7C*). However, we did observe a greater reduction in the fraction of peak current remaining after 250 ms of depolarization ($r_{250}$), (*Figure 7D*), suggesting elevated Ca$^{2+}$-dependent inactivation (CDI) of Cav1.2 in the presence of Kv2.1. Cells coexpressing STAC1 with Cav1.2 and Kv2.1$_{P404W}$ also exhibited an increase in whole-cell $I_{Ca}$ and a hyperpolarized

**Video 4.** Bay K8644 increases the frequency of somatic Ca$^{2+}$ sparks in CHNs. Rat CHN transfected with GCaMP3-Kv2.1 and imaged in the presence of 500 nM Bay K8644. Images were acquired at 11.3 Hz and have been normalized to the first image without detectable Ca$^{2+}$ signals (*i.e.*, F/F$_{min}$).
DOI: https://doi.org/10.7554/eLife.49953.016

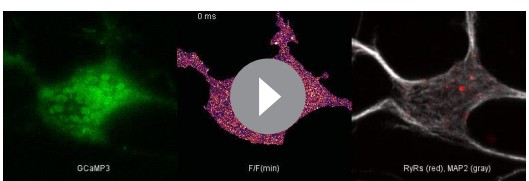

**Video 5.** Video depicting increase in Ca$^{2+}$ spark frequency upon addition of Bay K8644. Rat CHN transfected with GCaMP3-Kv2.1 and imaged at 20 Hz. 500 nM Bay K 8644 is added starting at approximately t = 25 s. Non-normalized GCaMP3-Kv2.1 images are shown on the left, images normalized to the first image without detectable Ca$^{2+}$ signals (*i.e.*, F/F$_{min}$) are shown in the center, and the same cell following fixation and immunolabeling for RyRs (red) and MAP2 (gray) is shown on the right.
DOI: https://doi.org/10.7554/eLife.49953.017

shift in Cav1.2 opening, similar to results obtained without STAC1 (*Figure 7—figure supplement 1A–C*). However, in the presence of STAC1, which substantially reduces CDI in Cav1.2 (*Campiglio et al., 2018*), Cav1.2 $r_{250}$ values were comparable between control cells and cells expressing Kv2.1 (*Figure 7—figure supplement 1D*). Measurement of $Ca^{2+}$-induced fluorescence increases in cells loaded with the $Ca^{2+}$-sensitive dye Rhod-2 *via* the patch pipette also revealed an enhancing effect of Kv2.1$_{P404W}$ on Cav1.2-mediated $Ca^{2+}$ influx (*Figure 7E*). Similarly, HEK293T cells loaded with the $Ca^{2+}$ dye Fluo-4 and expressing Cav1.2 and either Kv2.1$_{WT}$ or Kv2.1$_{P404W}$ displayed greater $K^+$-depolarization induced $Ca^{2+}$ influx than control cells (*Figure 7F–G*), further supporting that $K^+$-conducting as well as -nonconducting isoforms of Kv2.1 augment Cav1.2 activity.

Ion channel activity can be described by the product of the number of channels present in the PM ($n$), the channel's unitary conductance ($i$), and the open probability of these channels ($P_o$), such that the whole-cell current $I$ can be described by the relationship $I = nP_o i$. Thus, the enhancement of Cav1.2 activity observed in the presence of Kv2.1 could be caused by an effect on any one or more of these parameters. To better understand the underlying mechanism, we acquired gating and ionic tail currents from the same cell. Depolarization-induced voltage sensor movement in activating voltage-gated channels produces a gating current ($Q_{on}$) that is proportional to the number of channels present in the PM ($n$). Repolarization-induced ionic tail currents ($I_{tail}$) reveal overall channel activity ($I$). Changes in one or both can be used to infer whether it is '$n$' versus some combination of '$P_o$' and/or '$i$' that yield changes in total channel activity. We used nitrendipine, a DHP LTCC gating inhibitor, to pharmacologically isolate Cav1.2 $Q_{on}$ when the $V_m$ was stepped to the $I_{Ca}$ reversal potential, and to measure $I_{tail}$ elicited by returning to the $-70$ mV holding potential (*Figure 7H*). Nitrendipine-sensitive $Q_{on}$ values produced by Cav1.2 alone were comparable to those measured in the presence of Kv2.1, indicating that the increased $I_{Ca}$ in cells coexpressing Kv2.1 was not associated with an increase in the number of PM Cav1.2 channels (*Figure 7I*). However, the nitrendipine-sensitive $I_{tail}$ was significantly greater in the presence of Kv2.1, demonstrating that the open probability and/or conductance of Cav1.2 was increased when coexpressed with Kv2.1. As comparable $Q_{on}$ values (*i.e.*, Cav1.2 voltage sensor movement) produced a larger $I_{tail}$ in the presence of Kv2.1, these data taken together with the altered $G$-$V$ curve shown in *Figure 7C* suggest that the Kv2.1-dependent increase in $I_{Ca}$ apparently came from enhanced Cav1.2 voltage sensor coupling to channel opening.

We next tested how Kv2.1 impacted the spontaneous opening of Cav1.2 at hyperpolarized $V_m$ values. We used an optical approach to measure single Cav1.2 channel activity by recording Cav1.2-mediated $Ca^{2+}$ sparklets, local elevations in intracellular $Ca^{2+}$ produced by the opening of a single or small cluster of Cav1.2 channels (*Cheng and Lederer, 2008*). In addition to providing the single-channel activity of all active Cav1.2 channels present in the TIRF footprint, this approach had the additional benefit of revealing where in the PM the active channels were localized. We recorded $Ca^{2+}$ sparklets at a holding potential of $-70$ mV in the presence of 20 mM external $Ca^{2+}$ to increase the driving force for $Ca^{2+}$ influx. In control cells expressing Cav1.2 alone, we observed occasional sparklets whose activity was enhanced by the application of the LTCC agonist Bay K8644 (*Figure 8A–B,H–I*, *Video 7*). In contrast, control cells coexpressing Cav1.2 and DsRed-Kv2.1$_{P404W}$ displayed significantly more sparklets than control cells expressing Cav1.2 alone and a higher level of basal activity as measured by $nP_s$, where $n$ is the number of quantal levels of Cav1.2 opening and $P_s$ is the probability of sparklet occurrence (*Figure 8C–E,I,J,L*, *Video 8*). This observation is consistent with our ionic tail current data indicating that Cav1.2 open probability was enhanced in the presence of Kv2.1. Interestingly, Bay K8644 treatment did not result in any further enhancement in $nP_s$ in cells expressing DsRed-Kv2.1$_{P404W}$ (*Figure 8J*), suggesting that DsRed-Kv2.1$_{P404W}$ coexpression may result in near-maximal activation of Cav1.2-mediated sparklets at this hyperpolarized membrane potential. Many sparklets occurred near clusters of Kv2.1 channels, and we found that the nearest-neighbor distance (NND) of individual sparklet sites was significantly reduced in the presence of Kv2.1 (*Figure 8K*). Taken together, these findings demonstrate that Kv2.1 enhanced the open probability of Cav1.2 channels and increased the proximity of spontaneously active Cav1.2 channels to each other.

Given the impact of Kv2.1 coexpression on LTCC activity in heterologous HEK293T cells, we next asked whether LTCC currents were altered in CHNs lacking Kv2.1. For these experiments, we chose to record from CHNs as opposed to acutely dissociated neurons. Although the absence of extensive processes in acutely dissociated neurons enables much better control of the $V_m$ than in arborized neurons, we reasoned based on the loss of Kv2.1 clustering upon dissociation in other cell types

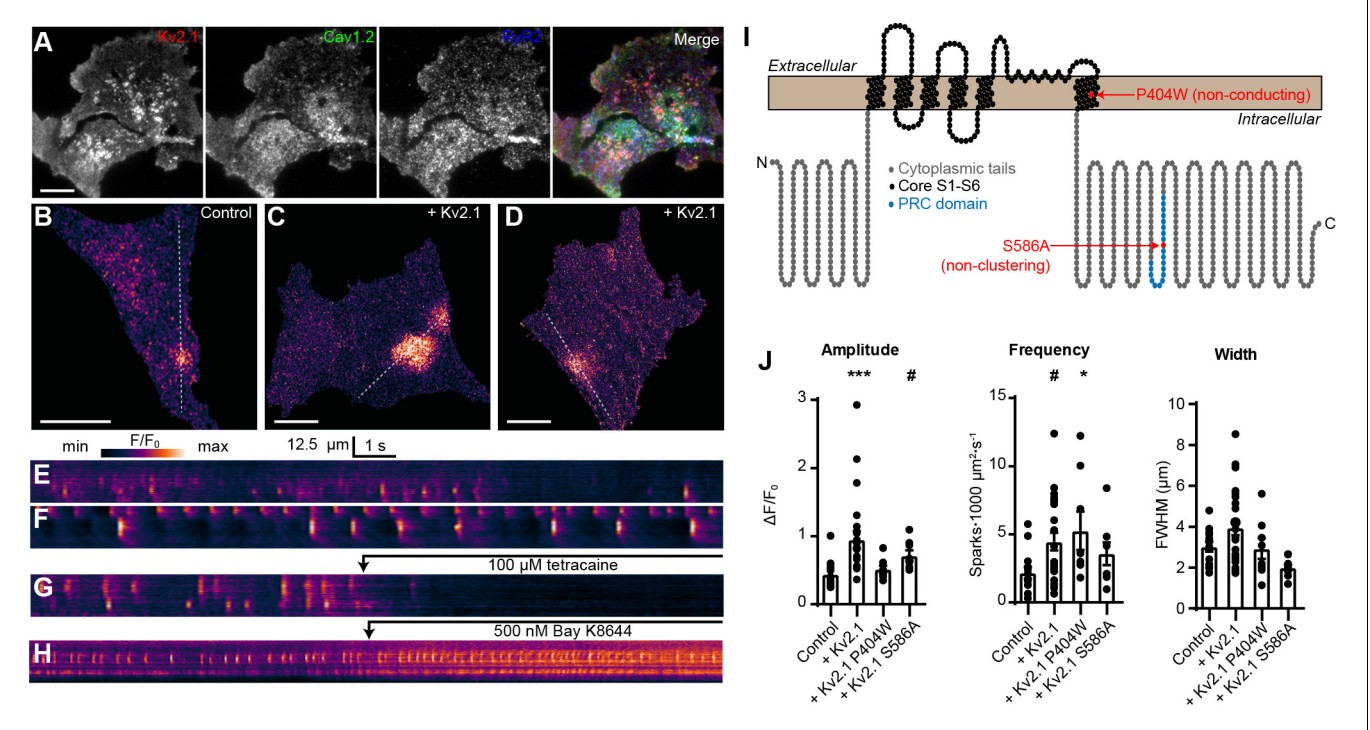

**Figure 6.** Kv2.1 expression increases the frequency of LTCC- and RyR-mediated sparks reconstituted in HEK293T cells. (**A**) TIRF images of a HEK293T cell cotransfected with DsRed-Kv2.1 (red), Cav1.2 (green), YFP-RyR2 (blue), and auxiliary subunits Cavβ3, Cavα2δ1, and STAC1 (not shown) (scale bar: 10 μm). (**B**) TIRF image of a HEK293T cell expressing Cav1.2, RyR2, STAC1, and the LTCC auxiliary subunits β3 and α2δ1, and loaded with Cal-590 AM. (**C–D**) TIRF images of HEK293T cells additionally coexpressing Kv2.1. Dashed line indicates ROI depicted in corresponding kymographs (scale bar in panels B-D: 10 μm). (**E–G**) Kymograph showing the localized Ca$^{2+}$ release events detected in the ROI on the cell in panels B-D, respectively. In (**G**), 100 μM tetracaine was added at the indicated time point. (**H**) Kymograph showing the localized Ca$^{2+}$ release events detected in a cell treated with 500 nM Bay K8644 at the indicated time point. (**I**) Illustration of the membrane topology of a single Kv2.1 α subunit depicting the locations of the P404W and S586A point mutations. (**J**) Summary data of the amplitude, frequency and spatial spread (width) of all sparks recorded from HEK293T cells expressing Cav1.2, RyR2, and auxiliary subunits, without (control) or with addition of the indicated Kv2.1 isoforms. Each point corresponds to a single cell (amplitude: ***p=0.0001, #p=0.051; frequency: #p=0.055, *p=0.047; one-way ANOVA followed by Dunnett's *post-hoc* test vs. control).

DOI: https://doi.org/10.7554/eLife.49953.018

The following figure supplement is available for figure 6:

**Figure supplement 1.** Kv2.1 increases the frequency of Cav1.3s and RyR-mediated sparks reconstituted in HEK293T cells.

DOI: https://doi.org/10.7554/eLife.49953.019

expressing clustered Kv2.1 (PC12, MDCK, and HEK293 cells; J.S. Trimmer, unpublished observations), and that endogenous Kv2.1 clusters in CHNs are sensitive to changes in intracellular Ca$^{2+}$ and metabolism (*Misonou et al., 2005b*), that acute dissociation would disrupt the clustered localization of Kv2.1, potentially concealing LTCC regulation by Kv2.1 clustering. To improve somatic voltage clamp, we used recording solutions lacking Na$^+$ and containing Cs$^+$ and Ba$^{2+}$ (which block K$^+$ channels; Ba$^{2+}$ also permeates voltage-gated Ca$^{2+}$ channels) to increase membrane impedance. We focused our analyses of electrophysiological recordings on repolarization-induced tail currents after activation of channels by a depolarizing prepulse, rather than measurement of currents induced by depolarizing voltage steps that can be distorted due to space clamp limitations (e.g., see *Milescu et al., 2010*). Similar to the impact of Kv2.1 on LTCCs in HEK293T cells, whole-cell Ba$^{2+}$ currents ($I_{Ba}$) at +10 mV (*Figure 9A*), as well as LTCC tail currents (*Figure 9B,C*) were larger in CHNs from WT mice than those measured in Kv2.1 KO CHNs (*Figure 9A–C*). To isolate the LTCC component of $I_{Ba}$, we applied the LTCC gating inhibitor nimodipine (10 μM), and found that the reduced $I_{Ba}$ observed in Kv2.1 KO CHNs (*Figure 9A–C*) was primarily due to a reduction in the nimodipine-sensitive component of the current (*Figure 9A,B,E*), with no apparent difference in the nimodipine-resistant current (*Figure 9A,B,D*). We also examined nimodipine-sensitive gating and ionic tail

currents when the $V_m$ was stepped to the $I_{Ba}$ reversal potential and found that while $Q_{on}$ was not significantly different between control and Kv2.1 KO CHNs, peak $I_{tail}$ was reduced in Kv2.1 KO CHNs (*Figure 9F–G*). The data in *Figures 7* and *8* (from exogenously expressed channels in HEK293T cells) and *Figure 9* (from endogenously expressed channels in CHNs) show that Kv2.1 enhances neuronal LTCC activity and suggest that the underlying mechanism in both experimental systems involves enhanced coupling efficiency between LTCC voltage-sensor movement and channel opening due to Kv2.1-mediated clustering.

## Kv2.1 promotes spatial coupling of LTCCs and RyRs

Given that Kv2.1-mediated clustering impacts the spatial distribution of Cav1.2 in coexpressing HEK293T cells, we next examined whether loss of Kv2.1 was associated with changes in the expression and localization of Cav1.2. We first performed immunolabeling of hippocampal neurons in brain sections from adult control and Kv2.1 KO mouse littermates. We have previously determined that the anatomic structure of mouse brains lacking Kv2.1 is comparable to controls, and there do not appear to be compensatory changes in the expression of other Kv channels tested (*Speca et al., 2014*). Here, we confirmed that immunolabeling for somatodendritic Kv2.2 and dendritic Kv4.2 channels was similar in WT and Kv2.1 KO hippocampus (*Figure 10A–C*). However, Cav1.2 labeling was increased in pyramidal neurons in area CA1 in Kv2.1 KO brain sections, both within the cell bodies and in the apical dendrites (*Figure 10C*). These results suggest that in adult mice lacking functional Kv2.1 channels, Cav1.2 expression may be elevated, potentially as a compensatory mechanism to overcome reduced Cav1.2 channel function.

To obtain more detailed individual cell information, we next investigated how the loss of endogenous Kv2.1 influenced the localization and function of LTCCs and RyRs in WT and Kv2.1 KO CHNs. To determine whether Kv2.1 channels regulate the localization of somatodendritic Cav1.2 and/or RyRs, we first analyzed the size and morphology of immunolabeled Cav1.2 and RyR clusters in WT and Kv2.1 KO mouse CHNs (*Figure 11A–B*). We found that compared to WT CHNs, Kv2.1 KO CHNs had reduced colocalization between Cav1.2 clusters and RyR, decreased size of RyR clusters, and increased distance between Cav1.2 clusters (*Figure 11C–F*). However, unlike the increased Cav1.2 immunolabeling found in adult Kv2.1 KO mouse brain sections, we found that the number of Cav1.2 clusters per $\mu m^2$ of somatic membrane did not differ between WT and Kv2.1 KO CHNs. These observations suggests that while compensatory changes in Cav1.2 expression did not occur in cultured Kv2.1 KO CHNs after approximately two weeks in vitro as it did in adult brain neurons in vivo, the presence of Kv2.1 promoted the spatial coupling of Cav1.2 to RyRs, consistent with our results in HEK293T cells.

Finally, to evaluate how impaired Cav1.2 and RyR spatial coupling in Kv2.1 KO CHNs affected spontaneous CICR events or sparks, we imaged Cal-590-loaded cells using TIRF microscopy. Similar to rat CHNs, we observed spontaneous sparks in WT mouse CHNs that were associated with Kv2.1, Cav1.2, and RyR clusters identified by *post-hoc* immunolabeling (*Figure 11G*). Consistent with the reduced colocalization of Cav1.2 and RyRs in Kv2.1 KO CHNs, we found that loss of Kv2.1 was associated with a significant reduction in spark frequency relative to WT control CHNs (*Figure 11H*). Taken together, these findings demonstrate that Kv2.1 channels promote the spatial and functional association of endogenous Cav1.2 and RyRs in neurons, as well as the corresponding exogenous channels in HEK293T cells.

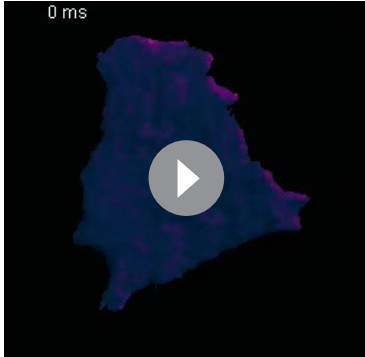

**Video 6.** Tetracaine blocks $Ca^{2+}$ sparks reconstituted in HEK293T cells. Stack of TIRF images acquired at 20 Hz of a single HEK293T cell transfected with RyR2, Cav1.2, and auxiliary subunits and loaded with Cal-590 AM. 100 $\mu M$ tetracaine was added at t = 7000 ms. Regular wavelike signals are a TIRF imaging artifact. Images have been normalized to the first image without detectable $Ca^{2+}$ signals (*i.e.*, $F/F_{min}$).
DOI: https://doi.org/10.7554/eLife.49953.020

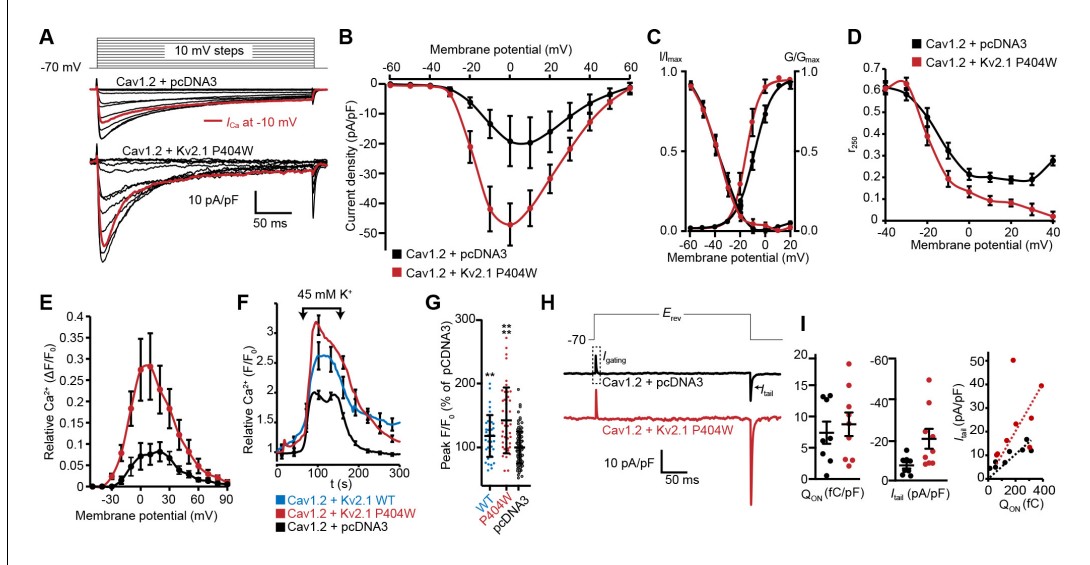

**Figure 7.** Cav1.2 channel activity is increased by coexpression with Kv2.1$_{P404W}$. (**A**) Representative Ca$^{2+}$ current trace families recorded from HEK293T cells transfected with Cav1.2-GFP and auxiliary subunits Cav$\beta$3 and Cav$\alpha_2\delta_1$, without (+ pcDNA3 empty vector) with cotransfection of DsRed-Kv2.1$_{P404W}$. For panels B-D, H, and I, data are from cells without (+ pcDNA3 empty vector, in black) or with coexpression of Kv2.1$_{P404W}$ (in red). (**B**) Normalized current-voltage (I–V) relationship of whole-cell $I_{Ca}$ recorded from $n = 17$ (Cav1.2 + pcDNA3) and $n = 10$ (Cav1.2 + Kv2.1$_{P404W}$) cells. (**C**) Voltage-dependence of whole-cell Cav1.2 conductance G/G$_{max}$ and steady-state inactivation I/I$_{max}$. For the conductance-voltage relationships, the half-maximal activation voltage $V_{1/2}$=-8.9±0.8 [pcDNA3] vs. $-13.9 \pm 1.6$ [+Kv2.1$_{P404W}$] mV, p=0.0045; slope factor $k = 6.9 \pm 0.3$ [pcDNA3] vs. $4.5 \pm 0.7$ [+Kv2.1$_{P404W}$], p=0.0025; Student's $t$-test. (**D**) Comparison of $r_{250}$ values (fraction of peak current remaining after 250 ms of depolarization) at the indicated potentials. (**E**) Average Rhod-2 fluorescence intensity measurements obtained from cells held at different membrane potentials during voltage clamp experiments ($n = 4$ cells per condition). (**F**) Average fluorescence intensity measurements from Fluo4-loaded HEK293T cells transfected with Cav1.2-RFP, auxiliary subunits Cav$\beta$3 and Cav$\alpha$2$\delta$, without (+ pcDNA3 empty vector, in black) or with cotransfection of Kv2.1$_{WT}$ (in blue) or Kv2.1$_{P404W}$ (in red). Ca$^{2+}$ influx was stimulated by depolarization with high extracellular K$^+$ (45 mM) as indicated on the graph. (**G**) Average peak fluorescence values obtained during high-K$^+$ depolarization of HEK293T cells expressing Cav1.2 and Kv2.1$_{WT}$ or Kv2.1$_{P404W}$ as in F. Each point represents a single cell. Bars are mean ± SD (**p<0.0001, *p=0.0047 versus control; Student's $t$-test). (**H**) Representative nitrendipine-sensitive Cav1.2 gating and tail currents recorded from control (pcDNA3) cells and cells coexpressing Kv2.1$_{P404W}$. (**I**) Quantification of nitrendipine-sensitive Cav1.2 Q$_{on}$ (left, p=0.3931, Student's $t$-test), $I_{tail}$ (center, *p=0.0195, Student's $t$-test), and Q$_{on\ vs.}I_{tail}$ (right). Each point corresponds to a single cell.

DOI: https://doi.org/10.7554/eLife.49953.021

The following figure supplement is available for figure 7:

**Figure supplement 1.** Cav1.2 channel activity is increased in cells coexpressing STAC1 upon coexpression with Kv2.1$_{P404W}$.

DOI: https://doi.org/10.7554/eLife.49953.022

## Discussion

The findings in this study support a new model for the formation of Ca$^{2+}$ signaling microdomains at EPJs and the local control of Ca$^{2+}$ release from these structures. In this model, neuronal EPJs are Ca$^{2+}$ signaling microdomains in which Cav1.2 and RyRs are brought into close proximity by Kv2.1-mediated clustering, forming a specialized somatic complex for the generation of localized Ca$^{2+}$ signals by these Ca$^{2+}$ channels (*Figure 11I*). We propose that a nonconducting function of PM Kv2.1 channels is to not only anchor the ER to the PM *via* a direct interaction with ER VAP proteins (*Johnson et al., 2018*; *Kirmiz et al., 2018a*), but also to promote the organization of Cav1.2 channels into clusters in direct apposition to nearby ER-localized RyRs. Our data indicate that Kv2.1-mediated clustering also increases the activity of Cav1.2. The enhanced spontaneous openings of Cav1.2 channels at negative potentials is evidenced by the increased frequency of sparklets, which allow a small amount of Ca$^{2+}$ to enter the cell at EPJs, activating nearby RyRs by the mechanism of CICR. The resulting Ca$^{2+}$ sparks occur independently of action potentials. Thus, our model proposes the molecular architecture of a protein complex (*Figure 11I*) underlying the localized somatodendritic Ca$^{2+}$ signals previously observed in brain neurons (*Berrout and Isokawa, 2009*; *Manita and Ross, 2009*), and suggests a mechanism whereby Kv2.1 modulates these Ca$^{2+}$ signals by

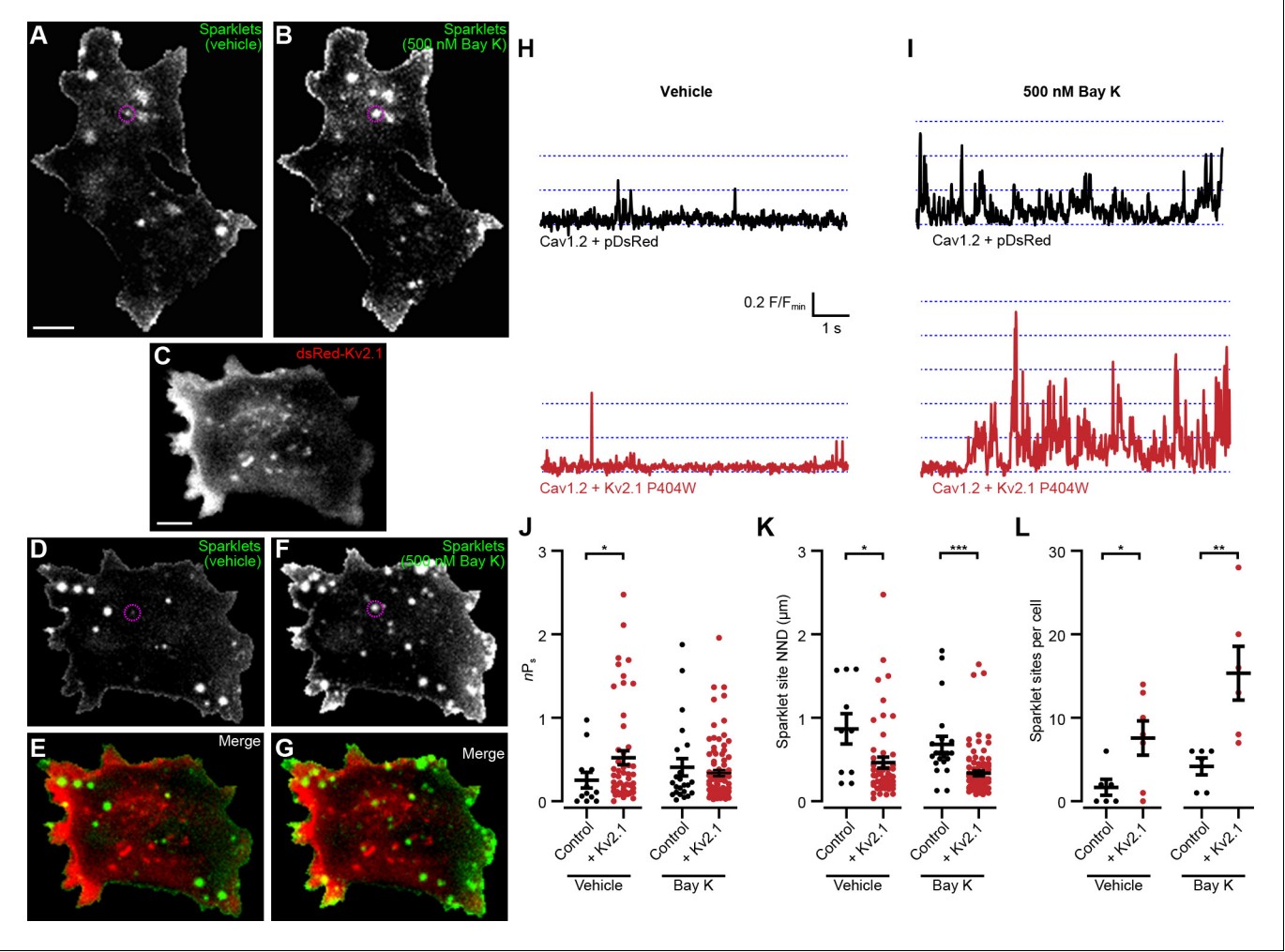

**Figure 8.** Kv2.1_{P404W} increases Cav1.2 single channel activity. (**A, B**) Maximum *z*-projections of TIRF images of Cav1.2-mediated Ca$^{2+}$ sparklets in a representative HEK293T cell transfected with Cav1.2 and auxiliary subunits and loaded with Fluo-5F *via* the patch pipette, before (**A**) and after (**B**) treatment with 500 nM Bay K8644 (scale bar: 5 μm). (**C**) Maximum *z*-projection of TIRF images of DsRed-Kv2.1 in a representative HEK293T cell cotransfected with Cav1.2 and auxiliary subunits (scale bar: 5 μm). (**D, F**) Maximum *z*-projections of TIRF images of sparklets in a representative HEK293T cell transfected with DsRed-Kv2.1, Cav1.2, and auxiliary subunits and loaded with Fluo-5F *via* the patch pipette, before (**D**) and after (**F**) treatment with 500 nM Bay K8644. (**E, G**) Merged images of panels C and D (**E**), or panels C and F (**G**). (**H**) Fluorescence intensity profiles of representative sparklets recorded in 20 mM external Ca$^{2+}$ in a control cell (upper panel, ROI depicted in A) or in a cell additionally expressing Kv2.1 (lower panel, ROI depicted in D). (**I**) Fluorescence intensity profiles of representative sparklets recorded in 20 mM external Ca$^{2+}$ and treated with Bay K8644 in a control cell (upper panel, ROI depicted in B) or in a cell additionally expressing Kv2.1 (lower panel, ROI depicted in F). (**J**) Summary data of sparklet site $nP_s$ measured from *n* = 6 cells expressing Cav1.2 alone and *n* = 7 cells coexpressing Cav1.2 and Kv2.1. Each point represents a single sparklet site (vehicle: *p=0.0367; Bay K: p=0.9224; two-tailed Mann-Whitney test). (**K**) Summary data of sparklet site nearest neighbor distance (NND) measured from *n* = 6 cells expressing Cav1.2 alone and *n* = 7 cells coexpressing Cav1.2 and Kv2.1. Each point represents a single sparklet site (vehicle: *p=0.0214; Bay K: p<0.0001; two-tailed Mann-Whitney test). (**L**) Summary data of the number of sparklet sites in *n* = 6 cells expressing Cav1.2 alone and *n* = 7 cells coexpressing Cav1.2 and Kv2.1. Each point represents a single cell (vehicle: *p=0.0318; Bay K: **p=0.0079; two-tailed *t*-test).
DOI: https://doi.org/10.7554/eLife.49953.023

simultaneously promoting the spatial association of Cav1.2 channels with RyRs and increasing their activity to trigger CICR.

## Kv2 channels dynamically cluster LTCCs

A key finding in this study is that endogenous LTCCs colocalize with clustered Kv2.1 in brain neurons, a finding supported by our crosslinking-based proteomic analyses showing that they exist in close spatial proximity. Moreover, colocalization of LTCCs with Kv2.1 could be reconstituted in

heterologous cells, a property that required Kv2.1's ability to cluster at EPJs but was separable from its voltage-gated K$^+$ channel function. The Kv2.1-mediated association of Cav1.2 with EPJs appears to be dynamically regulated and sensitive to neuronal activity, as acute dispersal of Kv2.1 clusters in CHNs by glutamate stimulation reduced Cav1.2 association with RyRs, whereas suppression of neuronal activity with TTX (which increases Kv2.1 phosphorylation and clustering) enhanced spatial coupling of Cav1.2 and RyRs. In addition, Kv2.1 expression in heterologous cells simultaneously enhanced the size of LTCC clusters and recruited LTCCs to Kv2.1-mediated EPJs where they more functionally coupled to RyRs to generate sparks. Consistent with this, we found that the spatial and functional coupling of somatic Cav1.2 channels to RyRs was reduced in Kv2.1 KO CHNs. Together, these findings indicate that LTCCs are recruited to Kv2.1-associated EPJs, a property we found was not shared by the T-type Ca$^{2+}$ channel Cav3.1. Moreover, the co-purification of several Cavβ auxiliary subunits, which associate with LTCCs but not T-Type Ca$^{2+}$ channels such as Cav3.1 (*Fang and Colecraft, 2011*), by IP of Kv2.1 from crosslinked brain samples further suggests a specific spatial interaction of LTCCs with Kv2.1. While like other plasma membrane proteins LTCCs can also exhibit stochastic clustering (*Sato et al., 2019*), numerous proteins have been identified that promote clustering of LTCCs in dendritic spines, including AKAP15 (*Marshall et al., 2011*) and PDZ domain-containing proteins (*Zhang et al., 2005*). The absence of these known LTCC clustering proteins from our proteomic analyses of proteins in close spatial proximity to Kv2.1, and our observation that expression of Kv2.1 increases Cav1.2 clustering in heterologous HEK293T cells, suggests that the proteins mediating Cav1.2 clustering in dendritic spines and at somatic EPJs may be distinct. We note that while our studies support that Kv2.1 coexpression leads to enhanced clustering of PM Cav1.2, our data do not allow us to distinguish whether this occurs through clustering of Cav1.2 channels already in the PM, or through other mechanisms, such as enhanced fusion of Cav1.2-containing endocytic vesicles that support enhanced clustering of Cav1.2 upon its reappearance in the PM, and that also leads to enhanced Cav1.2 clustering and cooperative gating (*Ghosh et al., 2018*).

Although the molecular mechanism of Kv2.1 recruitment to EPJs is now established, and occurs *via* its phosphorylation-dependent interaction with VAPs (*Johnson et al., 2018*; *Kirmiz et al., 2018a*), the precise molecular mechanism that underlies how LTCCs and RyRs are recruited to these sites is not yet clear. However, our data show that PM Cav1.2 organization was not impacted by coexpression of the clustering- and EPJ formation-deficient Kv2.1$_{S586A}$ mutant as it was by Kv2.1$_{WT}$ and the nonconducting Kv2.1$_{P404W}$ point mutant. Additionally, Kv2.1$_{S586A}$ was unable to enhance Cav1.2- and RyR-mediated sparks reconstituted in HEK293T cells, unlike these clustering-competent Kv2.1 isoforms. These findings support that Kv2.1 clustering and induction of EPJs is necessary for its spatial association with LTCCs.

It has been reported that LTCCs can also be recruited to the EPJs formed in HEK293T cells upon heterologous expression of junctophilin-2 (*Perni et al., 2017*), an ER-localized protein critical for bridging the PM to the ER in myocytes (*Jiang et al., 2016*). This is consistent with a model whereby tethering of LTCCs at or near Kv2-associated EPJs could be mediated by an intermediary recruited to Kv2.1-mediated EPJs, perhaps even one of the proteins identified in our proteomics analyses

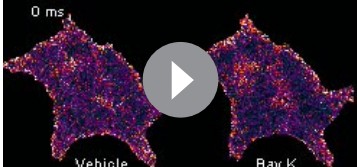

**Video 7.** Ca$^{2+}$ sparklets in a control HEK293T cell expressing Cav1.2. Stack of TIRF images acquired at approximately 100 Hz of a single HEK293T cell transfected with Cav1.2, PKCα, and auxiliary subunits and loaded with Fluo-5F *via* the patch pipette, before (left) and after (right) application of 500 nM Bay K8644. Each pixel has been normalized to its minimum pixel intensity (*i.e.*, F/F$_{min}$).
DOI: https://doi.org/10.7554/eLife.49953.024

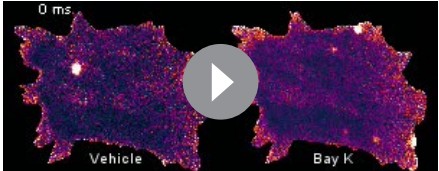

**Video 8.** Ca$^{2+}$ sparklets in a HEK293T cell coexpressing Cav1.2 and Kv2.1. Stack of TIRF images acquired at approximately 33 Hz of a single HEK293T cell transfected with Cav1.2, DsRed-Kv2.1$_{P404W}$, PKCα, and auxiliary subunits and loaded with Fluo-5F *via* the patch pipette, before (left) and after (right) application of 500 nM Bay K8644. Each pixel has been normalized to its minimum pixel intensity (*i.e.*, F/F$_{min}$).
DOI: https://doi.org/10.7554/eLife.49953.025

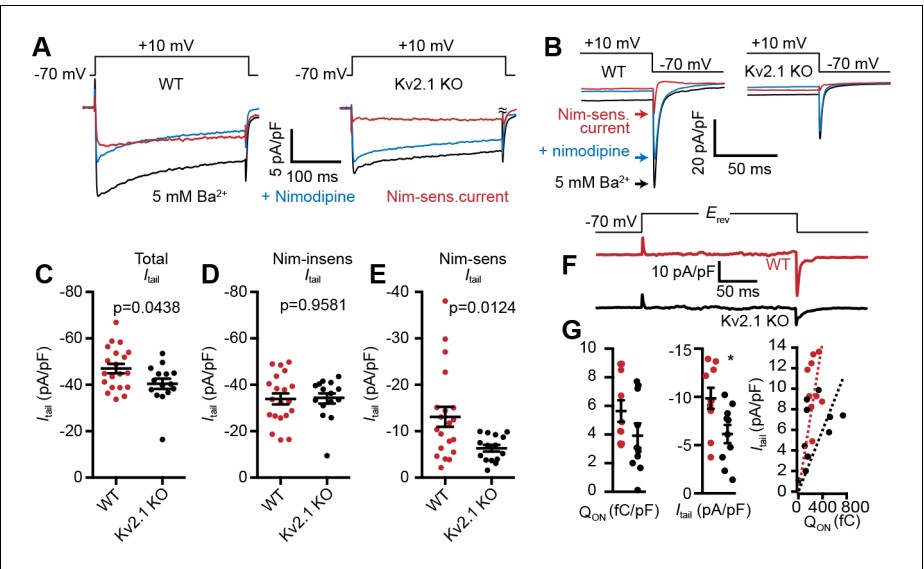

**Figure 9.** LTCC activity is reduced in Kv2.1 KO hippocampal neurons. (**A**) Representative $Ba^{2+}$ current traces recorded from WT (left) and Kv2.1 KO CHNs (right) recorded at +10 mV in vehicle or in the presence of the LTCC inhibitor nimodipine (10 μM). (**B**) Representative raw tail current records from a WT (left) and Kv2.1 KO (right) CHN induced by a step to −70 mV from a 10 mV prepulse, recorded in vehicle or in the presence of 10 μM nimodipine. C-F. Comparison of WT (red) and Kv2.1 KO (black) CHNs. (**C**) Maximum tail current amplitudes measured at −70 mV from a 10 mV prepulse. Each point represents one cell. (**D**) As in C but recorded in the presence of 10 μM nimodipine. (**E**) Maximum nimodipine-sensitive tail current amplitudes obtained from each cell by subtracting maximum tail current amplitudes measured in vehicle from those measured in the presence of nimodipine. (**F**) Representative nimodipine-sensitive LTCC gating and tail currents recorded from WT and Kv2.1 KO CHNs. (**G**) Quantification of nimodipine-sensitive LTCC $Q_{on}$ (left), $I_{tail}$ (center), and $Q_{on}$ $_{vs.}I_{tail}$ (right) recorded from WT and Kv2.1 KO CHNs. Each point corresponds to a single cell (*p=0.019, Student's t-test).

DOI: https://doi.org/10.7554/eLife.49953.026

(although note a different study did not observe this effect of junctophilin-2 coexpression [**Dixon et al., 2015**]). We note that these proteomics analyses have the potential to identify any proteins with lysine residues in close spatial proximity ($\approx$ 12 Å) to those in Kv2.1, making them amenable to being crosslinked to Kv2.1 by DSP, and do not require their direct association. Moreover, the crosslinking reaction could potentially yield 'daisy-chained' protein linkages of spatially adjacent proteins. While any such crosslinked protein chain would need to ultimately connect back to Kv2.1 to be immunopurified, every protein present in the purified sample need not be in close spatial proximity to Kv2.1 itself. Our observation that immunolabeling of endogenous Cav1.2 and Cav1.3 channels often appeared adjacent to rather than co-occurring with Kv2.1 (e.g., see **Figure 1H**) may also indicate that there is an indirect interaction between Kv2.1 and LTCCs. However, it remains possible that PM Kv2s and LTCCs associate through a direct intermolecular interaction. Any domains on Kv2.1 contributing to the spatial association with LTCCs, whether it occurs *via* direct or indirect interaction, would likely be conserved in Kv2.2, as we found that both Kv2 channel paralogs similarly impacted LTCC cluster size and localization. It is unlikely that RyRs are directly recruited to EPJs by Kv2 channels, as RyR clusters persist in CHNs exposed to treatments that disperse Kv2.1 clusters (**Misonou et al., 2005a**) and while reduced in size in CA1 pyramidal neurons in the double Kv2.1/Kv2.2 knockout (**Kirmiz et al., 2018a**), in general RyR clusters persist in neurons in the brains of mice lacking Kv2 channels (**Mandikian et al., 2014**; **Kirmiz et al., 2018a**). Further experiments are needed to determine the molecular mechanisms and direct protein-protein interactions that result in the spatial association of these proteins at neuronal EPJs.

## Kv2.1-dependent potentiation of Cav1.2 currents

Given their prominent physiological role, the regulation of LTCCs is extensive and multimodal (**Lipscombe et al., 2013**; **Hofmann et al., 2014**; **Neely and Hidalgo, 2014**). The mechanisms

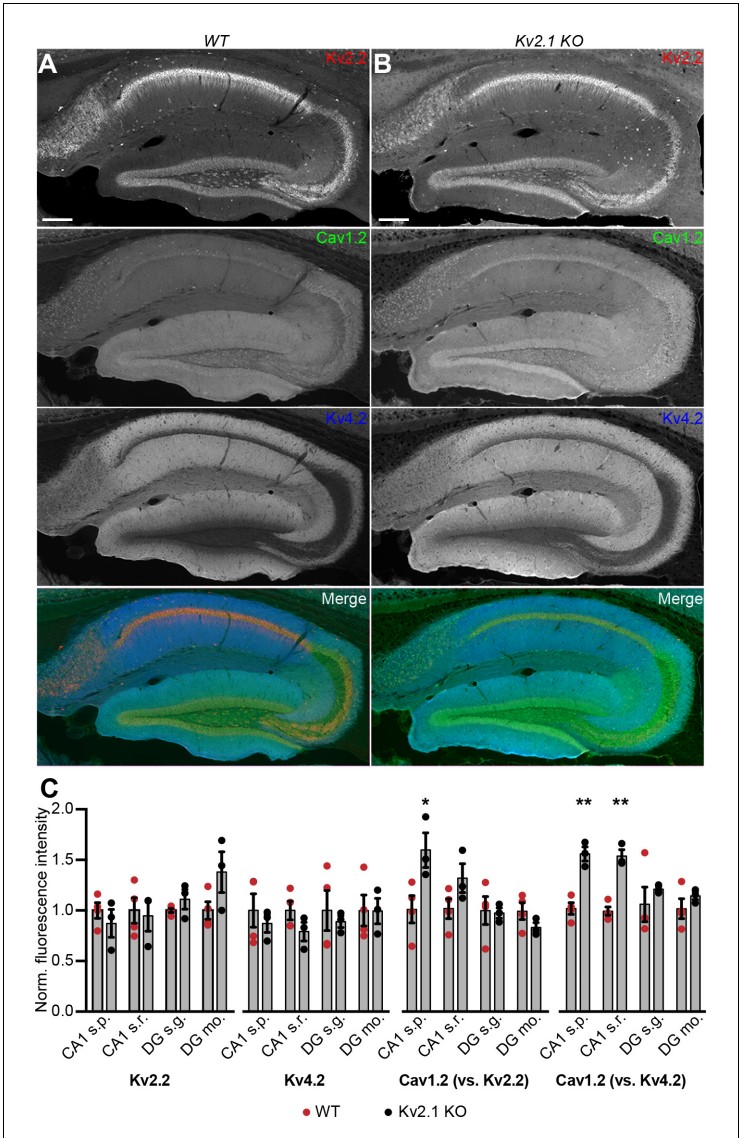

**Figure 10.** Increased immunolabeling for Cav1.2 in Kv2.1 KO brain sections. (**A**) Column shows exemplar images of the hippocampus acquired from brain sections of adult WT mice immunolabeled for Kv2.2 (red), Cav1.2 (green) and Kv4.2 (blue) (scale bar: 200 μm). (**B**) As in A but acquired from Kv2.1 KO mice. (**C**) Summary graphs of normalized mean fluorescence intensity of Kv2.2, Kv4.2, and Cav1.2 immunolabeling from ROIs from various laminae within CA1 (s.p.: *stratum pyramidale*; s.r.: *stratum radiatum*) and DG (s.g.: stratum granulosum; mo: molecular layer) in brain sections from adult WT (red) and Kv2.1 KO (black) mice. Each point corresponds to an individual mouse (Cav1.2 vs. Kv2.2: *p=0.0408; Cav1.2 vs. Kv4.2: **p=0.0018, ***p=0.0007).
DOI: https://doi.org/10.7554/eLife.49953.027

involved in the modulation of LTCC function include post-translational modification (e.g., phosphorylation), as well as changes in the expression of the subunits (principal α1, and auxiliary Cavβ and α2δ) that together comprise the quaternary structure of an LTCC (*Catterall, 2011*; *Zamponi et al., 2015*). We have recently demonstrated a novel mechanism for regulating Cav1.2- (and Cav1.3-) containing LTCCs, whereby LTCCs function differently when clustered due to their clustering-dependent cooperative gating (*Dixon et al., 2012*; *Dixon et al., 2015*; *Moreno et al., 2016*). Thus, LTCC activity is sensitive to its spatial organization in the PM, influenced by its proximity to adjacent LTCCs (*Navedo et al., 2005*; *Navedo et al., 2010*; *Dixon et al., 2012*; *Moreno et al., 2016*) and also to its localization to specific neuronal compartments (*Hall et al., 2013*; *Tseng et al., 2017*). In neurons,

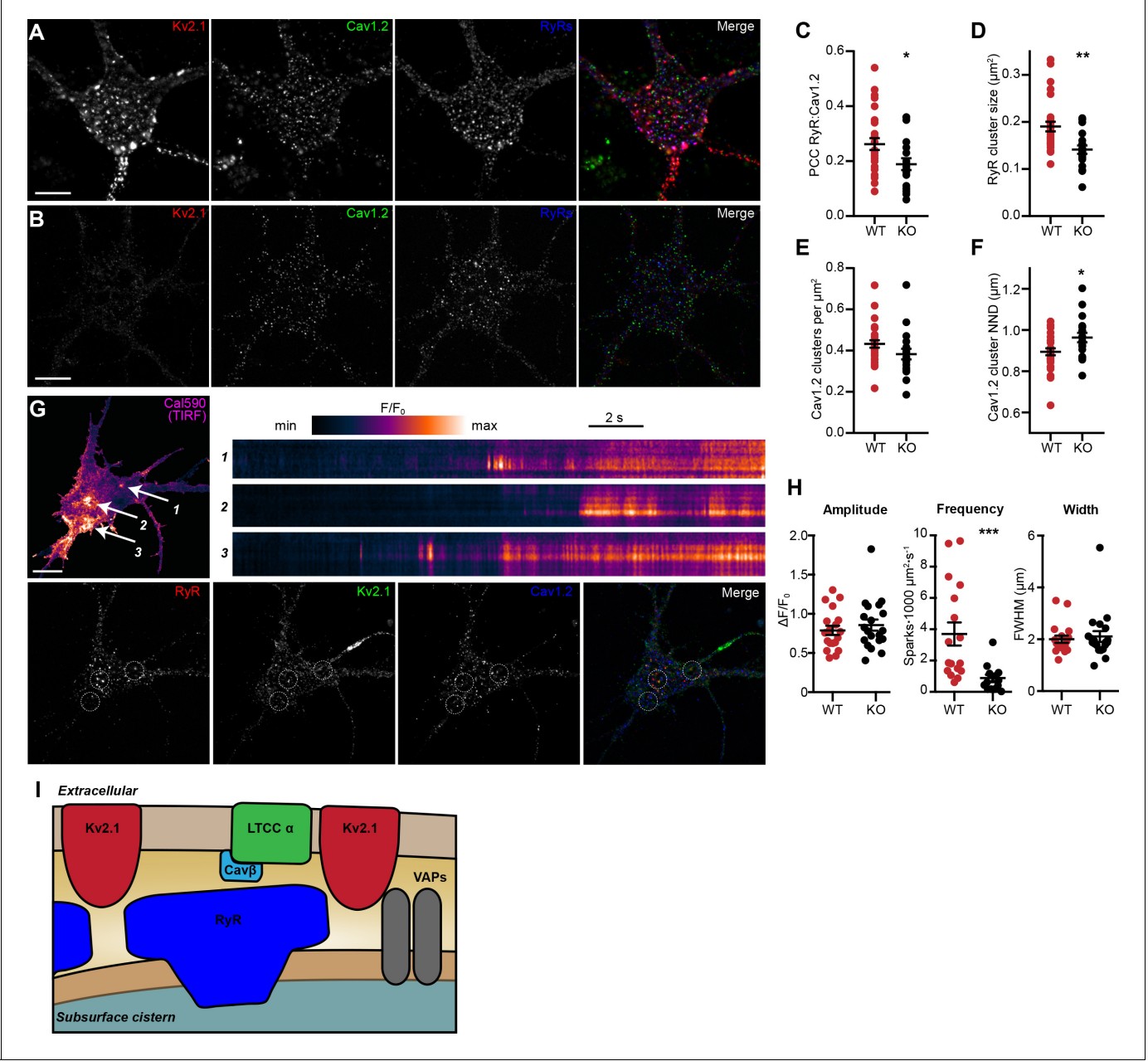

**Figure 11.** Reduced association of Cav1.2 and RyRs and decreased spark frequency in Kv2.1 KO CHNs. (A) A single optical section image of a WT mouse CHN immunolabeled for Kv2.1, Cav1.2, and RyRs (scale bar: 10 μm). (B) As in D but acquired from a Kv2.1 KO mouse CHN. (C–F) Morphology and spatial distribution of the indicated parameters determined from WT and Kv2.1 KO CHNs (each point represents one cell; Student's *t*-test). (C) *p=0.02255. (D) **p=0.0014. (E) p=0.1126. (F) *p=0.0173. (G) Representative WT mouse CHN loaded with Cal590 and imaged with TIRF microscopy, followed by *post-hoc* immunolabeling for RyRs, Kv2.1, and Cav1.2. Arrows indicate ROIs where spontaneous $Ca^{2+}$ signals were detected; dashed circles indicate approximate regions where immunolabeling for Kv2.1, Cav1.2, and RyRs was detectable. Kymograph showing the localized $Ca^{2+}$ release events detected at ROIs are depicted to the right. (H) Summary data of the amplitude, frequency and spatial spread (width) of all sparks recorded from WT and Kv2.1 KO mouse CHNs. Each point corresponds to a single cell (***p=0.0042; Student's *t*-test). (I) Diagram illustrates a model for the molecular architecture of Kv2.1-associated EPJs.

DOI: https://doi.org/10.7554/eLife.49953.028

such regulation likely acts to ensure that Cav1.2 is most active when properly targeted to specific subcellular domains and less active when outside these regions. Here, we show that the subcellular localization and activity of somatic Cav1.2 channels are influenced by Kv2.1, which increases both Cav1.2 clustering and its opening at polarized $V_m$ values. At least two other proteins, α-actinin (*Hall et al., 2013*) and densin-180 (*Wang et al., 2017*), exert a similar dual regulation on neuronal Cav1.2, by promoting its localization to dendritic spines and enhancing its activity at these sites. Neither of these proteins was identified in our proteomic analyses of proteins in close spatial proximity to Kv2.1, further suggesting that Cav1.2 complexes in dendritic spines and at somatic EPJs may be distinct. The reduced whole-cell LTCC currents and impaired association of somatic Cav1.2 with RyRs in Kv2.1 KO CHNs suggests that Kv2.1 serves this dual targeting/modulation function for LTCCs within the soma and proximal dendrites.

In both CHNs and HEK293T cells, currents resulting from the opening of endogenous and exogenous Cav1.2 channels, respectively, are increased in the presence of Kv2.1. In HEK293T cells, Cav1.2 channels coexpressed with clustered Kv2.1 are activated at more polarized $V_m$ values relative to those produced by Cav1.2 alone. Moreover, coexpression of Cav1.2 with nonconducting Kv2.1 increases the frequency of spontaneous Cav1.2 channel openings in HEK293T cells as reflected in an increased frequency of $Ca^{2+}$ sparklets. The Kv2.1-dependent increase in whole-cell Cav1.2 current amplitude in both HEK293T cells and CHNs occurs without an apparent change in the number of Cav1.2 channels present on the PM, as total Cav1.2 on-gating charges were unaltered by coexpression with Kv2.1. Instead, it appears that coupling of Cav1.2 voltage sensor movement to channel opening is enhanced in the presence of Kv2.1. What is the molecular mechanism underlying this effect on Cav1.2 channel opening? We suggest three possibilities. First, the increase in $I_{Ca}$ and leftward shift in the voltage-dependence of activation that we observed upon coexpression of Kv2.1 in HEK293T cells are similar to those observed during optogenetic induction of Cav1.2 channel oligomerization (*Navedo et al., 2010*; *Dixon et al., 2012*). Thus, one possible mechanism is that Kv2.1-induced clustering at EPJs increases the probability of physical interactions between Cav1.2 channels, which promotes their cooperative gating. A second possibility is that Kv2.1 functions as an auxiliary voltage sensor for Cav1.2 channels, perhaps through a direct intermolecular interaction of the two channels. However, the apparent localization of many Cav1.2 clusters adjacent to rather than directly overlapping with Kv2.1 clusters in CHNs (e.g., see *Figure 1B,H*) suggests that although these proteins associate in close spatial proximity, there may not be a direct interaction between individual Kv2.1 and Cav1.2 channels.

A third potential explanation for the Kv2.1-mediated increase in Cav1.2 channel activity is that Cav1.2 is modulated by signaling molecules that it encounters when recruited to EPJs by Kv2.1. It is well established that phosphorylation of Cav1.2 is a major mechanism to regulate its activity. Phosphorylation by protein kinase A (PKA) increases $Ca^{2+}$ influx through Cav1.2, enhancing CICR (*Dittmer et al., 2019*). Another candidate which might impact Cav1.2 at EPJs is $Ca^{2+}$/calmodulin-dependent protein kinase II (CaMKII), which has also been shown to interact with Kv2.1 (*McCord and Aizenman, 2013*). Enhanced Cav1.2 opening at polarized $V_m$ values and increased open probability are produced by both PKA- (*Tsien et al., 1986*; *Bers and Perez-Reyes, 1999*) and CaMKII- (*Erxleben et al., 2006*; *Blaich et al., 2010*) dependent phosphorylation of Cav1.2. Moreover, given the well-established association of RyRs with PKA and CaMKII (*Zalk et al., 2007*), it is conceivable that RyRs, Cav1.2, and Kv2.1 are substrates of these protein kinases at somatic EPJs. A recent study showed that in dendritic EPJs adjacent to spines, Cav1.2 is inhibited through a direct interaction with the ER-localized protein stromal interaction molecule 1 (STIM1) in a negative feedback response to Cav1.2- and RyR-mediated CICR (*Dittmer et al., 2019*). As such the Kv2.1-mediated localization of Cav1.2 at EPJs may bring it in close proximity to numerous regulatory molecules, at least a subset of which should also be expressed in HEK293T cells as these also exhibit prominent effects of Kv2.1 clustering on Cav1.2 activity.

## Properties of $Ca^{2+}$ sparks at Kv2.1-associated EPJs

The results presented here indicate that $Ca^{2+}$ sparks occurring at Kv2.1-associated EPJs were triggered primarily by $Ca^{2+}$ influx through LTCCs initiating the opening of juxtaposed RyRs. Accordingly, $Ca^{2+}$ spark frequency increased when neurons were exposed to Cav1.2 channel agonists and decreased by blockade of LTCCs. Loss of Kv2.1 expression was also associated with a decrease in

$Ca^{2+}$ spark frequency, likely because of decreased spatial association of Cav1.2 and RyRs, decreased RyR cluster size, and decreased LTCC currents.

As Kv2.1 clusters intrinsically represent EPJs by nature of their formation *via* an interaction with ER-resident VAPs (*Johnson et al., 2018*; *Kirmiz et al., 2018a*; *Kirmiz et al., 2018b*), our observation that localized $Ca^{2+}$ signals occurred only at a subset of GCaMP3-Kv2.1 clusters suggests that only a subset of Kv2.1-associated EPJs possess the molecular machinery required to generate these $Ca^{2+}$ signals. As identified in electron micrographs, EPJs represent a major class of membrane contact sites in brain neurons, where > 10% of the somatic PM may be engaged in EPJs (*Wu et al., 2017*). In addition to those formed by the Kv2.1:VAP association (*Johnson et al., 2018*; *Kirmiz et al., 2018a*; *Kirmiz et al., 2018b*), EPJs can be organized by a set of ER membrane proteins that bind PM phospholipids (*Henne et al., 2015*; *Gallo et al., 2016*). Experiments in heterologous cells exogenously expressing these ER-PM junction components show these ER tethers can also participate in ER-PM junctions formed by Kv2.1-VAP association (*Kirmiz et al., 2018b*). However, the relationship of the Kv2.1, LTCC and RyR-containing EPJs described here to those formed by these ER tethers in brain neurons and other cells that endogenously express these proteins is not known.

Our findings indicate that Kv2.1-mediated somatodendritic EPJs provide a molecular platform to elevate local $Ca^{2+}$ at individual EPJs without an increase in global $Ca^{2+}$, but that can also contribute to global, action potential-induced increases in cytoplasmic $Ca^{2+}$. These results reinforce previous observations (*Berrout and Isokawa, 2009*; *Manita and Ross, 2009*; *Miyazaki et al., 2012*; *Miyazaki and Ross, 2013*) that hippocampal neurons possess the molecular machinery to produce spontaneous local elevations in somatodendritic $Ca^{2+}$ that could potentially impact a wide variety of signaling pathways. That sparks can occur independently in neighboring Kv2.1-containing EPJs suggests a mechanism for compartmentalized $Ca^{2+}$ signaling in the aspiny regions of neurons (somata, proximal dendrites, axon initial segment) in which Kv2.1 clusters are located. One specific role identified for $Ca^{2+}$ signals produced by somatic RyR receptors at EPJs is in cartwheel cells (inhibitory interneurons found in the dorsal cochlear nucleus), where they trigger rapid gating of BK $Ca^{2+}$-activated $K^+$ channels to control electrical excitability (*Irie and Trussell, 2017*). While this mode of BK channel activation has not been observed in CA1 pyramidal neurons (*Ross, 2012*), somatic LTCC- and RyR-mediated $Ca^{2+}$ release has recently been demonstrated to activate KCa3.1 channels in hippocampal neurons, reducing spike frequency (*Sahu et al., 2019*). Sparks at Kv2.1-associated EPJs might also influence electrical activity in pyramidal cells through $Ca^{2+}$-sensitive enzymes that modify ion channel function, such as protein kinases and phosphatases that influence their phosphorylation state (*Misonou et al., 2004*). In addition, a role for somatic $Ca^{2+}$ sparks has been identified in DRG neurons, where they promote non-synaptic exocytosis of ATP-loaded secretory vesicles (*Ouyang et al., 2005*). Whether $Ca^{2+}$ entry mediated by LTCCs and RyRs at Kv2.1-associated EPJs impacts secretory vesicle exocytosis in brain neurons or other cell types will need to be investigated in future studies.

## Potential impact on downstream signaling pathways

Somatodendritic LTCCs are preferentially coupled to activation of signaling pathways resulting in changes in gene expression (*Wheeler et al., 2012*; *Wild et al., 2019*). In sympathetic neurons, local $Ca^{2+}$ influx through LTCCs rather than bulk elevation of intracellular $Ca^{2+}$ efficiently activates the transcription factor cAMP response element–binding protein (CREB) (*Wheeler et al., 2008*) through a mechanism that involves a signaling complex containing components of a PM-to-nucleus $Ca^{2+}$ shuttle (*Ma et al., 2012*; *Ma et al., 2014*; *Cohen et al., 2015*). Moreover, somatic LTCCs play a unique role in the $Ca^{2+}$ influx that leads to activation of the NFAT transcription factor (*Wild et al., 2019*). The results presented here suggest that Kv2.1-mediated organization and regulation of somatic LTCCs provides a molecular mechanism to control local $Ca^{2+}$ influx and serve as an organizer of $Ca^{2+}$ signaling microdomains. Previous work from us (*Misonou et al., 2004*; *Misonou et al., 2005b*) and others (*Mulholland et al., 2008*; *Aras et al., 2009*) has shown that acute ischemic or depolarizing events lead to $Ca^{2+}$-dependent dispersal of Kv2.1 clusters and hyperpolarize its $V_m$ activation threshold, potentially as a homeostatic mechanism to reduce neuronal activity and $Ca^{2+}$ overload that can lead to excitotoxicity. In our experiments here, we determined that Kv2.1-mediated clustering was associated with enhanced functional coupling of Cav1.2 and RyRs, as well as increased activation of Cav1.2 at polarized $V_m$ values. Therefore, the $Ca^{2+}$-dependent dispersal of Kv2.1 clusters and the resulting dissociation of Cav1.2 and RyRs may represent a negative feedback loop to limit excessive increases in cytoplasmic $Ca^{2+}$. By decreasing LTCC- and RyR-mediated CICR,

dispersal of Kv2.1 clusters may help to curb excessive accumulation of intracellular $Ca^{2+}$, which inappropriately activates signaling pathways contributing to neuronal damage or death (*Dirnagl et al., 1999*). Activity-dependent declustering of Kv2.1 may also help to reduce currents conducted by LTCCs, both through increased activation of hyperpolarizing Kv2.1 currents at polarized $V_m$ (opposing activation of voltage-gated $Ca^{2+}$ channels) and also through limiting Cav1.2 activity by altering its spatial organization in the PM. Our findings may also contribute to an understanding of the pathogenic mechanisms underlying mutations in Kv2.1 predicted to selectively disrupt the PRC domain required for Kv2.1 clustering (*de Kovel et al., 2017*).

Overall, the findings presented here identify a molecular structure underlying the spontaneous somatodendritic $Ca^{2+}$ signals previously observed in hippocampal pyramidal neurons. While our live cell experiments were primarily confined to CHNs cultured for 9–15 days in vitro, our data indicate that the spatial association of Kv2.1, Cav1.2, and RyRs is preserved in intact adult mouse and rat brains and can be recapitulated in heterologous cells. Moreover, somatodendritic $Ca^{2+}$ sparks have been observed in acute hippocampal slices obtained from rats aged P3-P80 (*Miyazaki et al., 2012*), suggesting that these $Ca^{2+}$ release events serve functional roles that emerge early in pyramidal neuron development and continue beyond this period. Although it is unclear whether spontaneous $Ca^{2+}$ sparks serve a specific function at their site of generation, or if they instead reflect stochastic events whose primary impact lies in their group behavior (*i.e.*, through modulation of bulk cytosolic $Ca^{2+}$), the results described here have relevance to obtaining a better understanding of their generation as well as their downstream effects.

# Materials and methods

## Key resources table

| Reagent type (species) or resource | Designation | Source or reference | Identifiers | Additional information |
|---|---|---|---|---|
| Cell line (Human) | HEK293T | ATCC Cat # CRL-3216 | RRID: CVCL_0063 | |
| Strain (*R. norvegicus*) | Sprague Dawley | Charles River | | |
| Strain (*M. musculus*) | C57/BL6J mice | The Jackson Laboratory | RRID: IMSR_JAX:000664 | |
| Strain (*M. musculus*) | Kcnb1[-/-] mice | PMID: 17767909; PMID: 24494598 | RRID: MGI:3806050 | maintained on the C57BL/6J background |
| Antibody | numerous | | | See *Table 2* |
| Recombinant DNA reagent | GCaMP3-Kv2.1 | This paper | | |
| Recombinant DNA reagent | GCaMP3-Kv2.1[P404W] | This paper | | |
| Recombinant DNA reagent | DsRed-Kv2.1 | PMID: 30012696 | | |
| Recombinant DNA reagent | DsRed-Kv2.1[P404W] | PMID: 30012696 | | |
| Recombinant DNA reagent | DsRed-Kv2.2 | PMID: 30012696 | | |
| Recombinant DNA reagent | Kv2.1[S586A] | PMID: 10719893 | | |
| Recombinant DNA reagent | Kv1.5 | PMID: 8636142 | | |
| Recombinant DNA reagent | BFP-Sec61β | Addgene | Plasmid #49154 | |
| Recombinant DNA reagent | Cav1.2-eGFP | PMID: 25714924 | | |
| Recombinant DNA reagent | Cav1.2-tagRFP | PMID: 25714924 | | |

*Continued on next page*

*Continued*

| Reagent type (species) or resource | Designation | Source or reference | Identifiers | Additional information |
|---|---|---|---|---|
| Recombinant DNA reagent | Cav1.3$_S$-GFP | PMID: 27187148 | | |
| Recombinant DNA reagent | Cav1.2 | Addgene | Plasmid #26572 | |
| Recombinant DNA reagent | Cav1.2-HA | PMID: 15090038 | | Gift from Dr. Valentina Di Biase |
| Recombinant DNA reagent | Cavα2δ1 | Addgene | Plasmid #26575 | |
| Recombinant DNA reagent | Cavβ3 | Addgene | Plasmid #26574 | |
| Recombinant DNA reagent | YFP-RyR2 | PMID: 17452324 PMID: 20427316 | | Gift from Dr. S.R. Wayne Chen |
| Recombinant DNA reagent | STAC1 | DNASU | Plasmid # HsCD00445396 | |
| Chemical compound | Cal-590 AM | AAT Bioquest | Cat# 20510 | |
| Chemical compound | Rhod-2 | AAT Bioquest | Cat# 21068 | |
| Chemical compound | Fluo-4 AM | Invitrogen | Cat# F14201 | |
| Chemical compound | Fluo-5F | Invitrogen | Cat# F14221 | |
| Chemical compound | Caffeine | Sigma | Cat# C0750 | |
| Chemical compound | Thapsigargin | Millipore | Cat# 586005 | |
| Chemical compound | Nimodipine | Alomone | Cat# N-150 | |
| Chemical compound | Nitrendipine | Alomone | Cat# N-155 | |
| Chemical compound | Bay K8644 | Alomone | Cat# B-350 | |
| Chemical compound | Tetracaine | Sigma | Cat# T7508 | |
| Chemical compound | Tetrodotoxin | Alomone | Cat #T-550 | |
| Chemical compound | Amphotericin B | Millipore | Cat# 171375 | |
| Software | Photoshop | Adobe Systems | RRID:SCR_014199 | |
| Software | Axiovision | Carl Zeiss MicroImaging | RRID:SCR_002677 | |
| Software | pClamp | Molecular Devices | RRID:SCR_011323 | |
| Software | TILLvisION | TILL Photonics | | |
| Software | Fiji | PMID: 22743772 | RRID:SCR_002285 | |
| Software | Prism | GraphPad Software | RRID:SCR_002798 | |

## Animals

All procedures involving rats and mice were approved by the University of California, Davis Institutional Animal Care and Use Committee and performed in accordance with the NIH Guide for the Care and Use of Laboratory Animals. Animals were maintained under standard light-dark cycles and

allowed to feed and drink ad libitum. Sprague-Dawley rats were used for immunolabeling experiments and as a source of hippocampal neurons for primary culture. Kv2.1 KO mice (RRID:IMSR_MGI: 3806050) (*Jacobson et al., 2007*; *Speca et al., 2014*) were generated from breeding of *Kcnb1*[+/-] mice that had been backcrossed on the C57/BL6J background (RRID:IMSR_JAX:000664). Littermates were used when available. Adult male (mice and rats) and female (rats) were used in immunohistochemistry experiments; adult male and female mice were used in proteomics; P0-P1 mouse littermates were used as a source of hippocampal neurons for primary culture. Experiments using CHNs were performed using neuronal cultures obtained from pooling neurons from animals of both sexes (rats and mice) and also cultures in which individual pups were grouped by sex after visual inspection (mice).

## Hippocampal neuron cultures

Neuronal cultures were prepared and maintained as previously described (*Kirmiz et al., 2018a*; *Kirmiz et al., 2018b*). Hippocampi were dissected from either postnatal day 0–1 pups (mice) following genotyping or embryonic day 18 embryos (rat) and dissociated enzymatically for 20 min at 37°C in HBSS supplemented with 0.25% (w/v) trypsin (Worthington Cat# LS003707), followed by mechanical dissociation *via* trituration with fire-polished glass Pasteur pipettes. Dissociated cells were suspended in plating medium containing Neurobasal (ThermoFisher Cat# 21103049) supplemented with 10% fetal bovine serum (FBS, Invitrogen Cat# 16140071), 2% B27 supplement (Invitrogen Cat# 17504044), 2% GlutaMAX (ThermoFisher Cat# 35050061), and 0.001% gentamycin (Gibco Cat# 15710064) and plated at 60,000 cells per dish in glass bottom dishes (MatTek Cat# P35G-1.5–14 C) or on microscope cover glasses (Karl Hecht Assistent Ref# 92099005050) coated with poly-L-lysine (Sigma Cat# P2636). After 5 days in vitro (DIV), cytosine-D-arabinofuranoside (Millipore Cat# 251010) was added to inhibit non-neuronal cell growth. Neurons were transiently transfected at DIV 7–10 using Lipofectamine 2000 (Invitrogen Cat# 11668019) for 1.5 hr as previously described (*Lim et al., 2000*). Neurons were used for experiments 40–48 hr post transfection.

For acute treatment of rat CHNs with glutamate or TTX, 20–24 DIV neurons cultured on microscope cover glasses were incubated in 1 mL of a modified Krebs-Ringer buffer (KRB) containing (in mM): 146 NaCl, 4.7 KCl, 2.5 CaCl$_2$, 0.6 MgSO$_4$, 1.6 NaHCO$_3$, 0.15 NaH$_2$PO$_4$, 8 glucose, 20 HEPES, pH 7.4, approximately 330 mOsm for 30 min at 37°C. We then added an additional 1 mL of KRB prewarmed to 37°C, with or without 20 µM glutamate (Calbiochem Cat #3510) or 1 µM TTX (Alomone Cat #T-550) for a final concentration of 10 µM (glutamate) or 500 nM (TTX), and incubated CHNs for 10 min (glutamate) or 1 hr (TTX) at 37°C. We then proceeded immediately to fixation.

## HEK293T cell culture

HEK293T cells were obtained from ATCC (Cat# CRL-3216). The accompanying Certificate of Analysis shows species determination was performed and yielded the expected results. HEK293T cells were further validated by short terminal repeat (STR) analysis. Cells were tested for mycoplasma contamination using the MycoAlert Mycoplasma Detection Kit (Lonza Catalog#: LT07-318). HEK293T cells were maintained in Dulbecco's modified Eagle's medium (Gibco Cat# 11995065) supplemented with 10% Fetal Clone III (HyClone Cat# SH30109.03), 1% penicillin/streptomycin, and 1x GlutaMAX (ThermoFisher Cat# 35050061) in a humidified incubator at 37°C and 5% CO$_2$. Cells were transiently transfected using Lipofectamine 2000 following the manufacturer's protocol, in DMEM without supplements, then returned to regular growth medium 4 hr after transfection. 20–24 hr later, cells were passaged to obtain single cells on glass bottom dishes (MatTek Cat# P35G-1.5–14 C) or microscope cover glasses (VWR Cat# 48366–227) coated with poly-L-lysine. Cells were then used for experiments approximately 15 hr after being passaged.

## Immunolabeling of cells

Immunolabeling of CHNs and HEK293T cells was performed as described previously (*Kirmiz et al., 2018a*; *Kirmiz et al., 2018b*). CHNs were fixed in ice cold 4% (wt/vol) formaldehyde (freshly prepared from paraformaldehyde, Fisher Cat# O4042) in phosphate buffered saline (PBS, Sigma Cat #P3813) supplemented with 4% (wt/vol) sucrose (Sigma Cat# S9378), pH 7.4, for 15 min at 4°C. HEK293T cells were fixed in 3.2% formaldehyde (freshly prepared from paraformaldehyde) and 0.1% glutaraldehyde (Ted Pella, Inc, Cat# 18426) prepared in PBS pH 7.4, for 20 min at room temperature

**Table 2.** Antibody information.

| Antigen and antibody name | Immunogen | Manufacturer information | Concentration used | Figures |
|---|---|---|---|---|
| PSD-95 (K28/43) | Fusion protein aa 77–299 of human PSD-95 | Mouse IgG2a mAb, NeuroMab, RRID:AB_10807979 | Tissue culture supernatant (1:5) | *Figure 1* |
| Cav1.2 (N263/31) | Fusion protein aa 808–874 of rat Cav1.2 | Mouse IgG2b mAb, NeuroMab, RRID:AB_11001554 | Tissue culture supernatant (1:5) | *Figure 1*, *Figure 2*, *Figure 2—figure supplement 1*, *Figure 6*, *Figure 10*, *Figure 11* |
| Cav1.2 (L57/23) | Fusion protein aa 1507–1733 of rabbit Cav1.2 | Mouse IgG2a mAb, In-house (Trimmer Laboratory) RRID:AB_2802123 | Tissue culture supernatant, neat | *Figure 1*, *Figure 3* |
| Cav1.3 (ACC-005) | Synthetic peptide aa 859–875 of rat Cav1.3 | Rabbit pAb, Alomone catalog # ACC-005, RRID:AB_2039775 | Affinity purified, 10 µg/mL | *Figure 1* |
| Kv2.1 (KC) | Synthetic peptide aa 837–853 of rat Kv2.1 | Rabbit pAb, In-house (Trimmer Laboratory), RRID:AB_2315767 | Affinity purified, 1:100 | *Table 1* (immunopurifications) |
| Kv2.1 (K89/34R) | Synthetic peptide aa 837–853 of rat Kv2.1 | Recombinant mouse IgG2a mAb, In-house (Trimmer Laboratory), RRID:AB_2750677 | Tissue culture supernatant (1:5) | *Figure 1*, *Figure 2*, *Figure 3*, *Figure 3—figure supplement 2*, *Figure 4*, *Figure 11* |
| Kv2.1 (K39/25R) | Synthetic peptide aa 211–229 of human Kv2.1 | Recombinant mouse IgG2a mAb, In-house (Trimmer Laboratory), RRID:AB_2750663 | Tissue culture supernatant (1:5) | *Figure 3—figure supplement 1* |
| MAP2 (AB5622-I) | KLH-conjugated three peptides from N-and C-terminal regions of rat MAP2 | Rabbit pAb, Millipore catalog # AB5622-I, RRID: AB_2800501 | Affinity purified, 1:1000 | *Figure 1*, *Figure 4*, *Figure 5* |
| RyRs (34C) | Partially purified chicken pectoral muscle ryanodine receptor | Mouse IgG1 mAb, Developmental Studies Hybridoma RRID:AB_528457 | Concentrated tissue culture supernatant, 3 µg/ml | *Figure 1*, *Figure 2*, *Figure 4*, *Figure 5*, *Figure 6*, *Figure 11* |
| Cav3.1 (N178A/9) | Fusion protein aa 2052–2172 of mouse Cav3.1 | Mouse IgG1 mAb, NeuroMab, RRID:AB_10673097 | Tissue culture supernatant (1:5) | *Figure 3—figure supplement 2* |
| Kv1.5 (Kv1.5e) | Synthetic peptide aa 271–284 of rat Kv1.5 | Rabbit pAb, in-house (Trimmer Laboratory), RRID:AB_2722698 | Affinity purified, 5 µg /ml | *Figure 3—figure supplement 1* |
| Kv2.2 (Kv2.2C) | Fusion protein aa 717–907 of rat Kv2.2 | Rabbit pAb, in-house (Trimmer Laboratory), RRID:AB_2801484 | Affinity purified, 1:100 | *Figure 2—figure supplement 1*, *Figure 10* |
| Kv4.2 (K57/41) | Synthetic peptide aa 209–225 of human Kv4.2 | Mouse IgG1 mAb, In-house (Trimmer Laboratory), RRID:AB_2802124 | Affinity purified, 10 µg /ml | *Figure 1*, *Figure 10* |
| Anti-HA (12CA5) | Amino acids 98–106 of the human influenza virus hemagglutinin protein | Mouse IgG2b mAb, In-house (Trimmer Laboratory) RRID: AB_2532070 | Pure, 5 µg/mL | *Figure 3* |

*Table 2 continued on next page*

*Table 2 continued*

| Antigen and antibody name | Immunogen | Manufacturer information | Concentration used | Figures |
|---|---|---|---|---|
| Anti-HA (2–2.2.14-647) | HA peptide YPYDVPDYA | Mouse IgG1 mAb, Thermo Fisher Scientific catalog # 26183-A647, RRID: AB_2610626 | Affinity purified, 1 µg /ml | *Figure 3—figure supplement 1* |

DOI: https://doi.org/10.7554/eLife.49953.029

(RT), washed 3 × 5 min in PBS and quenched with 0.1% sodium borohydride (Sigma Cat# 452882) in PBS for 15 min at RT. All subsequent steps were performed at RT. Cells were then washed 3 × 5 min in PBS, followed by blocking in blotto-T (Tris-buffered saline [10 mM Tris, 150 mM NaCl, pH 7.4] supplemented with 4% (w/v) non-fat milk powder and 0.1% (v/v) Triton-X100 [Roche Cat# 10789704001]) for 1 hr. Cells were immunolabeled for 1 hr with primary antibodies diluted in blotto-T (primary antribodies and concentrations used are listed in *Table 2*). Following 3 × 5 min washes in blotto-T, cells were incubated with mouse IgG subclass- and/or species-specific Alexa-conjugated fluorescent secondary antibodies (Invitrogen) diluted in blotto-T for 45 min, then washed 3 × 5 min in PBS. Cover glasses were mounted on microscope slides with Prolong Gold mounting medium (ThermoFisher Cat # P36930) according to the manufacturer's instructions. For cell surface immunolabeling of HEK293T cells, cells were fixed and quenched in sodium borohydride as described above, followed by 3 × 10 min washes in PBS without Triton X-100, blocked for 1 hr in blotto-T without Triton X-100, then incubated for 2 hr in primary antibodies diluted in blotto-T without Triton X-100. Cells were then washed 3 × 10 min in PBS without Triton X-100, followed by fixation of surface antibody with 1% formaldehyde prepared in PBS for 15 min. Cells were then washed 3 × 5 min in PBS and processed for immunolabeling of cellular proteins as described above. For TIRF imaging of fixed cells, cover glasses were mounted in PBS onto glass depression slides.

Unless otherwise stated, optical sections were acquired with an AxioCam MRm digital camera installed on a Zeiss AxioImager M2 microscope or with an AxioCam HRm digital camera installed on a Zeiss AxioObserver Z1 microscope with a 63×/1.40 NA Plan-Apochromat oil immersion objective and an ApoTome coupled to Axiovision software (Zeiss, Oberkochen, Germany). Confocal images were acquired using a Zeiss LSM880 confocal laser scanning microscope equipped with an Airyscan detection unit and a Plan-Apochromat 63×/1.40 NA Oil DIC M27 objective. Structured illumination microscopy (N-SIM) images were acquired with a Hamamatsu ORCA-ERCCD camera on a SIM/widefield capable Nikon Eclipse Ti microscope with an EXFO X-Cite metal halide light source and a 100 × PlanApo TIRF/1.49 objective. Colocalization and morphological analyses of Cav1.2 and RyRs in CHNs was performed using Fiji (NIH). For the colocalization analyses, an ROI was drawn around the soma of a neuron and PCC values were collected using the Coloc2 plugin. All intensity measurements were collected using Fiji. All intensity measurements reported in line scans are normalized to the maximum intensity measurement. Measurements of cluster sizes were performed essentially as previously described (*Kirmiz et al., 2018a*; *Kirmiz et al., 2018b*). Images were subjected to rolling ball background subtraction and subsequently converted into a binary mask by thresholding. Cluster sizes were measured using the 'analyze particles' feature of Fiji; nearest neighbor distances were calculated from cluster centroid values using the nearest neighbor distance plugin in Fiji. The spatial distributions of Kv2.1 and Cav1.2 puncta were analyzed using the Interaction Analysis function that is part of the MosaicSuite plugin for Fiji. For presentation, images were exported as TIFFs and linearly scaled for min/max intensity and flattened as RGB TIFFs in Photoshop (Adobe).

## Immunolabeling of brain sections

Following administration of pentobarbital to induce deep anesthesia, animals were transcardially perfused with 4% formaldehyde (freshly prepared from paraformaldehyde) in 0.1 M sodium phosphate buffer pH 7.4 (0.1 M PB). Sagittal brain sections (30 µm thick) were prepared and immunolabeled using free-floating methods as detailed previously (*Rhodes et al., 2004*; *Speca et al., 2014*; *Bishop et al., 2015*; *Palacio et al., 2017*). Sections were permeabilized and blocked in 0.1 M PB containing 10% goat serum and 0.3% Triton X-100 (vehicle) for 1 hr at RT, then incubated overnight at 4°C in primary antibodies (*Table 2*) diluted in vehicle. After 4 × 5 min washes in 0.1 M PB, sections

were incubated with mouse IgG subclass- and/or species-specific Alexa-conjugated fluorescent secondary antibodies (Invitrogen) and Hoechst 33258 DNA stain diluted in vehicle at RT for 1 hr. After 2 × 5 min washes in 0.1 M PB followed by a single 5 min wash in 0.05 M PB, sections were mounted and air dried onto gelatin-coated microscope slides, treated with 0.05% Sudan Black (EM Sciences) in 70% ethanol for 2 min (*Schnell et al., 1999*). Samples were then washed extensively in water and mounted with Prolong Gold (ThermoFisher Cat # P36930). Images of brain sections were taken using the same exposure time to compare the signal intensity directly using an AxioCam HRm high-resolution CCD camera installed on an AxioObserver Z1 microscope with a 10×/0.5 NA lens, and an Apo-Tome coupled to Axiovision software, version 4.8.2.0 (Zeiss, Oberkochen, Germany). Labeling intensity within stratum pyramidale and stratum radiatum of hippocampal area CA1 was measured using a rectangular region of interest (ROI) of approximately 35 μm x 185 μm. Labeling intensity within stratum granulosum and the inner third of stratum moleculare of the dentate gyrus (DG) was measured using a rectangular ROI of approximately 48 μm x 200 μm. To maintain consistency between samples, the average pixel intensity values of ROIs from CA1 were acquired near the border of CA1 and CA2, and those from DG were obtained near the center of the dorsal/suprapyramidal blade of the DG. Signal intensity values from all immunolabels and of Hoechst dye were measured from the same ROI. Background levels for individual labels were measured from no primary controls for each animal and subtracted from ROI values. High magnification confocal images of rat and mouse hippocampus were acquired using a Zeiss LSM880 confocal laser scanning microscope equipped with an Airyscan detection unit and a Plan-Apochromat 63×/1.40 NA Oil DIC M27 objective.

## Immunopurification of Kv2.1 and proteomics

Crosslinked mouse brain samples for immunopurification were prepared as previously described (*Kirmiz et al., 2018a*). Excised brains were homogenized over ice in a Dounce homogenizer containing 5 mL of ice-cold homogenization and crosslinking buffer (in mM): 320 sucrose, 5 NaPO$_4$, pH 7.4, supplemented with 100 NaF, 1 PMSF, protease inhibitors, and 1 DSP (Lomant's reagent, ThermoFisher Cat# 22585). Following a 1 hr incubation on ice, DSP was quenched with 20 mM Tris, pH 7.4 (JT Baker Cat# 4109–01 [Tris base]; and 4103–01 [Tris-HCl]). 2 mL of this homogenate was then added to an equal volume of ice-cold 2x radioimmunoprecipitation assay (RIPA) buffer (final concentrations): 1% (vol/vol) TX-100, 0.5% (wt/vol) deoxycholate, 0.1% (wt/vol) SDS, 150 NaCl, 50 Tris, pH 8.0 and incubated on a tube rotator at 4°C for 30 min. Insoluble material was then pelleted by centrifugation at 12,000 × g for 10 min at 4°C. The supernatant was incubated overnight at 4°C with the anti-Kv2.1 rabbit polyclonal antibody KC (*Trimmer, 1991*). Following this incubation, we added 100 μL of magnetic protein G beads (ThermoFisher Cat# 10004D) and incubated the samples on a tube rotator at 4°C for 1 hr. Beads were then washed 6x following capture on a magnet in ice-cold 1x RIPA buffer, followed by four washes in 50 mM ammonium bicarbonate (pH 7.4). Proteins captured on magnetic beads were digested with 1.5 mg/mL trypsin (Promega Cat# V5111) in 50 mM ammonium bicarbonate overnight at 37°C. The eluate was then lyophilized and resuspended in 0.1% trifluoroacetic acid in 60% acetonitrile.

Proteomic profiling was performed at the University of California, Davis Proteomics Facility. Tryptic peptide fragments were analyzed by LC-MS/MS on a Thermo Scientific Q Exactive Plus Orbitrap Mass spectrometer in conjunction with a Proxeon Easy-nLC II HPLC (Thermo Scientific) and Proxeon nanospray source. Digested peptides were loaded onto a 100 μm x 25 mm Magic C18 100 Å 5U reverse phase trap where they were desalted online, then separated using a 75 μm x 150 mm Magic C18 200 Å 3U reverse phase column. Peptides were eluted using a 60 min gradient at a flow rate of 300 nL per min. An MS survey scan was obtained for the *m/z* range 350–1600; tandem MS spectra were acquired using a top 15 method, where the top 15 ions in the MS spectrum were subjected to HCD (High Energy Collisional Dissociation). Precursor ion selection was performed using a mass window of 1.6 *m/z*, and normalized collision energy of 27% was used for fragmentation. A 15 s duration was used for the dynamic exclusion. MS/MS spectra were extracted and charge state deconvoluted by Proteome Discoverer (Thermo Scientific). MS/MS samples were then analyzed using X! Tandem (The GPM, thegpm.org; version Alanine (2017. 2. 1.4)). X! Tandem compared acquired spectra against the UniProt Mouse database (May 2017, 103089 entries), the cRAP database of common proteomic contaminants (www.thegpm.org/crap; 114 entries), the ADAR2 catalytic domain sequence, plus an equal number of reverse protein sequences assuming the digestion enzyme

trypsin. X! Tandem was searched with a fragment ion mass tolerance of 20 ppm and a parent ion tolerance of 20 ppm. Variable modifications specified in X! Tandem included deamidation of asparagine and glutamine, oxidation of methionine and tryptophan, sulfone of methionine, tryptophan oxidation to formylkynurenin of tryptophan and acetylation of the N-terminus. Scaffold (version Scaffold_4.8.4, Proteome Software Inc, Portland, OR) was used to validate tandem MS-based peptide and protein identifications. X! Tandem identifications were accepted if they possessed -Log (Expect Scores) scores of greater than 2.0 with a mass accuracy of 5 ppm. Protein identifications were accepted if they contained at least two identified peptides. The threshold for peptide acceptance was greater than 95% probability. Data in *Table 1* are presented as spectral counts over three independent experiments, normalized to spectral counts for Kv2.1 peptides returned in each experiment.

## Plasmid constructs

To maintain consistency with previous studies, we use the original (*Frech et al., 1989*) amino acid numbering of rat Kv2.1 (accession number NP_037318.1). The generation of DsRed-Kv2.1 and -Kv2.2 plasmids has been described previously (*Kirmiz et al., 2018b*). GCaMP3-Kv2.1 was generated using Gibson assembly to insert GCaMP3 (*Tian et al., 2009*) into the Kv2.1 RBG4 vector (*Shi et al., 1994*), resulting in fusion of GCaMP3 to the N-terminus of full-length rat Kv2.1. The plasmid encoding $Kv2.1_{S586A}$ has been previously described (*Lim et al., 2000*); the plasmid encoding $Kv2.1_{P404W}$ in the pcDNA4/TO vector was a gift from Dr. Jon Sack (University of California, Davis). The plasmid encoding Kv1.5 has been previously described (*Nakahira et al., 1996*). The plasmids encoding GFP- and RFP-tagged full-length rabbit Cav1.2 $\alpha 1$ subunit (accession number NP_001129994.1), the GFP-tagged short isoform of rat Cav1.3 $\alpha$ subunit (accession AAK72959.1), and PKCα have been previously described (*Moreno et al., 2016*; *Dixon et al., 2015*; *Navedo et al., 2006*). Plasmids encoding untagged full-length mouse Cav1.2, rat Cavβ3, and rat $\alpha_2\delta_1$ were gifts of Dr. Diane Lipscombe (Brown University). The plasmid encoding BFP-Sec61β was a gift from Dr. Gia Voeltz (Addgene plasmid #49154). Plasmid encoding HA-tagged rat Cav1.2 was a gift from Dr. Valentina Di Biase (Medical University of Graz), plasmid encoding human Cav3.1 was a gift from Dr. Edward Perez-Reyes (University of Virginia), and plasmid encoding full-length mouse RyR2 fused with YFP (*Wang et al., 2007*; *Liu et al., 2010*) was a gift of Dr. S.R. Wayne Chen (University of Calgary). The vector encoding human STAC1 was obtained from DNASU (DNASU plasmid # HsCD00445396).

## Live cell imaging

HEK293T cells transfected with RyR2-YFP, LTCC α1 subunit (Cav1.2 or Cav1.3s), Cavβ3, Cavα$_2$δ$_1$, STAC1, and empty vector control (pcDNA3) or $DsRed-Kv2.1_{P404W}$ plasmids in a 1.5:1:0.5:0.5:0.25:1 ratio were seeded to glass bottom dishes (MatTek Cat# P35G-1.5–14 C) approximately 15 hr prior o recording. Total internal reflection fluorescence (TIRF) and widefield microscopy imaging of HEK293T cells and DIV9-10 (transfected with GCaMP3-Kv2.1) or DIV14-21 (loaded with Cal-590 AM) CHNs cultured on glass-bottom dishes was performed in KRB at 37˚C as previously described (*Kirmiz et al., 2018a*; *Kirmiz et al., 2018b*). For imaging of cells loaded with Ca$^{2+}$-sensitive dye, cells were first incubated in regular culture medium to which had been added 1.5 μM Cal-590 AM (AAT Bioquest Cat# 20510) for 45 min or Fluo-4 AM (Invitrogen Cat# F14201) for 25 min at 37˚C. Dye-containing medium was then aspirated, followed by two washes in KRB which had been warmed to 37˚C. Cells were then incubated in KRB for an additional 30 min at 37˚C prior to imaging. Caffeine (Sigma Cat# C0750), thapsigargin (Millipore Cat# 586005), nimodipine (Alomone Cat# N-150), Bay K8644 (Alomone Cat# B-350), and tetracaine (Sigma Cat# T7508) were dissolved in warm KRB at 2x the final concentration and added to cells during imaging by pipette. For GxTX-633 labeling of cells, cells were incubated in 300 nM GxTX-633 dissolved in KRB supplemented with 0.1% BSA for 20 min at 37˚C, followed by a single wash with KRB. Images were acquired on a Nikon Eclipse Ti TIRF/widefield microscope equipped with an Andor iXon EMCCD camera and a Nikon LUA4 laser launch with 405, 488, 561, and 647 nm lasers, using a 100×/1.49 NA PlanApo TIRF objective and NIS Elements software. For *post-hoc* immunolabeling of CHNs, the dish orientation and location of the imaged cell was recorded, after which the CHNs were fixed in ice-cold 4% formaldehyde/4% sucrose in PBS, pH 7.4, and processed for immunolabeling as described above. Recorded CHNs were identified on the basis of expression of GCaMP3-Kv2.1 and/or neurite morphology revealed by immunolabeling

for MAP2. Acquired image stacks were processed and analyzed using Fiji; we used the Fiji plugin xySpark (*Steele and Steele, 2014*) for automated spark detection and analysis.

## Electrophysiology

HEK293T cells transfected with Cav1.2-GFP, Cavβ3, Cavα$_2$δ$_1$, and empty vector control (pcDNA3) or DsRed-Kv2.1$_{P404W}$ plasmids in a 1:0.5:0.5:1 ratio were seeded to microscope cover glasses (Fisher Cat# 12-545-102) approximately 15 hr prior to recording to obtain single cells. Coexpression of Cav1.2 and Kv2.1$_{P404W}$ in HEK293T cells was apparently cytotoxic and thus necessitated seeding of cells at a higher density to obtain viable single cells as compared to control cells expressing Cav1.2 alone. HEK293T cells were patched in an external solution of modified Krebs-Ringer buffer (KRB) containing (in mM): 146 NaCl, 4.7 KCl, 2.5 CaCl$_2$, 0.6 MgSO$_4$, 1.6 NaHCO$_3$, 0.15 NaH$_2$PO$_4$, 8 glucose, 20 HEPES, pH 7.4, approximately 330 mOsm. Transfected cells were identified by the presence of GFP and DsRed expression. $I_{Ca}$ was recorded in transfected cells using the whole-cell voltage clamp patch configuration using fire-polished borosilicate pipettes that had resistances of 2–3 MΩ when filled with an internal solution containing (in mM): 125 Cs-methanesulfonate, 10 TEA-Cl, 1 MgCl$_2$, 0.3 Na$_2$-GTP, 13 phosphocreatine-(di)Tris, 5 Mg•ATP, 5 EGTA, 10 HEPES, adjusted to pH 7.22 with CsOH, approximately 320 mOsm. Currents were sampled at 20 kHz and low-pass–filtered at 2 kHz using an Axopatch 200B amplifier, and acquired using pClamp 10.2 software (Molecular Devices, Sunnyvale, CA). All experiments were performed at room temperature (22–25˚C). Pipette capacitance was compensated using the amplifier, and capacitance and ohmic leak were subtracted online using a P/5 protocol. Current–voltage (*I–V*) relationships were obtained approximately three minutes after obtaining the whole-cell configuration by subjecting cells to a series of 300 ms depolarizing pulses from the holding potential of −70 mV to test potentials ranging from −60 to +100 mV in 10 mV increments. The voltage dependence of $G/G_{max}$ was obtained from the recorded currents by converting them to conductances (*G*) using the equation $G = I_{Ca}/(\text{test pulse potential} – E_{rev(Ca)})$, plotting the normalized values ($G/G_{max}$) versus the test potential, and fitting them to a Boltzmann function. Steady-state inactivation was measured by subjecting cells to a series of 2500 ms conditioning prepulses from the holding potential to potentials ranging from −60 to +100 mV, returning to the −70 mV holding potential for 5 ms, then measuring the peak current elicited by a 300 ms step to the −20 mV test potential. Data were analyzed and plotted using Prism software (Graphpad Software Inc, San Diego, CA). For experiments in which depolarization-induced increases in Ca$^{2+}$-sensitive dye were measured, we included 0.2 mM Rhod-2 (AAT Bioquest Cat# 21068) in the patch pipette solution. Images were acquired at 10 Hz using a through-the-lens TIRF microscope built around an Olympus IX-70 inverted microscope equipped with an oil-immersion ApoN 60×/1.49 NA TIRF objective and an Andor iXON CCD camera using TILLvisION imaging software (TILL Photonics, FEI, Hillsboro, OR).

To measure gating and ionic tail currents, we first determined the reversal potential for $I_{Ca}$ from the *I–V* relationship obtained using the *I–V* protocol described above. Gating currents were then measured by applying a series of depolarizing steps from the holding potential (−70 mV) to potentials ± 5 mV of the reversal potential in 1 mV increments. Currents were sampled at a frequency of 25 kHz and low-pass filtered at 2 kHz. We first obtained recordings in cells perfused with KRB alone, then obtained recordings from the same cell after it had been perfused for two minutes with KRB containing 1 μM nitrendipine (Alomone Cat# N-155). To isolate gating currents and $I_{tail}$ produced by Cav1.2, we subtracted currents measured in the presence of nitrendipine from those measured in KRB alone. The on-gating charge ($Q_{on}$) was then obtained from these records by integrating the gating current within approximately 2 ms of a depolarizing step to the reversal potential, and maximal $I_{tail}$ amplitudes were measured upon repolarization to the holding potential.

Somatic whole-cell patch clamp recordings were acquired from WT and Kv2.1 KO mouse CHNs cultured on microscope cover glasses after 15–16 DIV. Pyramidal neurons were selected based upon their morphological characteristics (*Benson et al., 1994*). Patch pipettes were fashioned and filled with intracellular recording solution as described above. After establishing the whole-cell configuration in KRB, the bath solution was exchanged with an extracellular recording buffer containing (in mM): 135 NMDG, 30 TEA-Cl, 5 BaCl$_2$, 8 glucose, 20 HEPES, adjusted to pH 7.4 with HCl. Series resistance was 9.9 ± 0.9 (WT) and 10.4 ± 0.9 (Kv2.1 KO) MΩ (p=0.694, Student's *t*-test) (before compensation); cell capacitance was 52.9 ± 4.8 (WT) and 58.4 ± 4.0 (Kv2.1 KO) pF (p=0.789, Student's *t*-test). Prior to recording, cell capacitance was canceled, and series resistance was partially (60–70%)

compensated. Recordings of LTCC ionic and gating currents were then performed as described for HEK293T cells. We used 10 μM nimodipine to isolate the contribution of LTCCs to the measured currents.

For simultaneous measurement of the $V_m$ and $Ca^{2+}$ sparks, rat CHNs transfected with GCaMP3-Kv2.1 were recorded using the whole-cell perforated patch clamp configuration. CHNs were patched in KRB using pipettes filled with a solution containing (in mM): 135 K-gluconate, 15 KCl, 5 NaCl, 1 MgCl$_2$, 0.1 EGTA, 10 HEPES, pH adjusted to 7.22 using KOH, and amphotericin B (Millipore Cat# 171375) dissolved in DMSO and added at a final concentration of approximately 50 μg/mL. Upon obtaining a GΩ seal, the amplifier was switched to the current clamp mode to record spontaneous fluctuations in the $V_m$. Measurement of the $V_m$ (sampled at 25 kHz) and widefield image acquisition (acquired at 5 Hz) were triggered simultaneously using the same microscope described above.

## Sparklets

We recorded Cav1.2-mediated $Ca^{2+}$ sparklets using the dual TIRF imaging/patch clamp system described above. HEK293T cells transfected with untagged mouse Cav1.2, pDsRed-monomer-C1 or DsRed-Kv2.1$_{P404W}$, Cavβ3, Cavα$_2$δ$_1$, and rat PKCα (*Navedo et al., 2006*), which increases spontaneous sparklet activity, were loaded *via* the patch pipette with a solution containing (in mM): 0.2 Fluo-5F (Invitrogen Cat# F14221), 87 Cs-aspartate, 20 CsCl, 1 MgCl$_2$, 5 Mg•ATP, 10 HEPES, 10 EGTA, adjusted to pH 7.2 with CsOH. After obtaining a GΩ seal in KRB, the external solution was exchanged with a solution containing (in mM): 110 NaCl, 5 CsCl, 1 MgCl$_2$, 10 glucose, 10 HEPES, 20 CaCl$_2$, pH 7.4 with NaOH. Cells were maintained at a holding potential of −70 mV, and TIRF images were acquired using TILLvisION software. Sparklets were manually detected and analyzed using Fiji software. Sparklet activity was quantified by calculating the $nP_s$ of each site (*Navedo et al., 2006*).

## Experimental design and statistical analysis

For all data sets presented in this study for which statistical analyses were performed, measurements were imported into GraphPad Prism and Microsoft Excel for presentation and statistical analysis. Reported values are mean ± SEM, unless stated otherwise. Exact p-values are reported in each figure or figure legend. Paired data sets were compared using a Student's *t*-test if the data passed a normality test; a non-parametric test was used otherwise.Proteomics on brain samples were collected from three independent sets of age-matched male wild-type and Kv2.1 KO adult mice. For experiments involving HEK293T cells and CHNs, at least two independent cultures were used for experimentation; the number of samples (*n*) indicates the number of cells analyzed and is noted in each figure or figure legend.

## Acknowledgements

We thank Kimberly Nguyen and Grace Or Mizuno for assistance in preparation of rat CHNs, and Dr. Manuel Navedo for helpful discussion and comments on the manuscript. We also thank Drs. Parashar Thapa and Jon Sack for providing GxTX-633. Aspects of the microscopy in this study were performed using the resources of the UC Davis MCB Imaging Facility, and we thank Dr. Michael Paddy for expert advice on imaging. The proteomics experiments in this study were performed using the resources of the UC Davis Proteomics Core Facility, and we thank Michelle Salemi for expert advice on proteomics. This project was funded by National Institutes of Health Grants U01NS099714 (JST), R21NS101648 (JST), R01HL144071 (LFS and JST), T32GM007377 (MK), and F32NS108519 (NCV).

## Additional information

### Funding

| Funder | Grant reference number | Author |
| --- | --- | --- |
| National Institute of Neurological Disorders and Stroke | R21 NS101648 | James S Trimmer |
| National Heart, Lung, and Blood Institute | R01 HL144071 | L Fernando Santana James S Trimmer |

| National Institute of General Medical Sciences | T32 GM007377 | Michael Kirmiz |
| National Institute of Neurological Disorders and Stroke | F32 NS108519 | Nicholas C Vierra |
| National Institute of Neurological Disorders and Stroke | U01 NS099714 | James S Trimmer |

The funders had no role in study design, data collection and interpretation, or the decision to submit the work for publication.

## Author contributions

Nicholas C Vierra, Conceptualization, Resources, Data curation, Formal analysis, Funding acquisition, Validation, Investigation, Visualization, Methodology, Writing—original draft; Michael Kirmiz, Conceptualization, Resources, Data curation, Formal analysis, Validation, Investigation, Visualization, Methodology, Writing—review and editing; Deborah van der List, Resources, Investigation, Methodology, Writing—review and editing; L Fernando Santana, Conceptualization, Resources, Supervision, Funding acquisition, Project administration, Writing—review and editing; James S Trimmer, Conceptualization, Resources, Data curation, Formal analysis, Supervision, Funding acquisition, Validation, Project administration, Writing—review and editing

## Author ORCIDs

Nicholas C Vierra  http://orcid.org/0000-0001-7269-5399
L Fernando Santana  http://orcid.org/0000-0002-4297-8029
James S Trimmer  https://orcid.org/0000-0002-6117-3912

## Ethics

Animal experimentation: This study was performed in strict accordance with the recommendations in the Guide for the Care and Use of Laboratory Animals of the National Institutes of Health. All of the animals were handled according to approved institutional animal care and use committee (IACUC) protocols (#20485 and #21265) of the University of California, Davis. All perfusions were performed under sodium pentobarbital anesthesia, and every effort was made to minimize suffering.

## Decision letter and Author response

Decision letter https://doi.org/10.7554/eLife.49953.032
Author response https://doi.org/10.7554/eLife.49953.033

# Additional files

## Supplementary files

• Transparent reporting form  DOI: https://doi.org/10.7554/eLife.49953.030

## Data availability

All data generated or analysed during this study are included in the manuscript and supporting files.

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
