## [Decision Letter]

**Acceptance summary:**

This study by Vierra and colleagues describes the novel observation that the Kv2.1 voltage-activated potassium channel forms clusters that interact functionally with L-type voltage-activated calcium channels on the plasma membrane and with ryanodine receptors, an intracellular calcium channel that mediates release of calcium from intracellular stores. The authors show that CaV1.2 clusters increase in size with expression of Kv2.1, and that this effect also occurs with a mutant channel in which potassium permeation has been abolished, but not with a mutant channel exhibiting diminished clustering of the Kv2.1 channel. They also find that co-expression of Kv2.1 and CaV1.2 modulates the activity of this calcium channel by increasing the amplitude of calcium currents and shifting opening of the channel to more negative membrane voltages, making it easier for the channel to open in response to membrane depolarization. The editors and reviewers agreed that this is rigorous, well-executed study that presents important and original findings concerning the interaction of different ion channels involved in electrical signaling within neurons. The authors use an impressive number of tools and experimental approaches, the data are all very high quality and the manuscript is well-written. The experimental results all support the main claims of the paper and the authors were responsive to issues raised in review and have done an excellent job of revising the manuscript.

**Decision letter after peer review:**

Thank you for submitting your article "Kv2.1 mediates spatial and functional coupling of L-type calcium channels and ryanodine receptors in neurons" for consideration by *eLife*. Your article has been reviewed by three peer reviewers, and the evaluation has been overseen by Kenton Swartz as the Reviewing Editor and Eve Marder as the Senior Editor. The reviewers have opted to remain anonymous.

The reviewers have discussed the reviews with one another and the Reviewing Editor has drafted this decision to help you prepare a revised submission.

All reviewers found your study on the role of Kv2.1 in functional coupling of Cav1.2 and ryanodine receptors (RyR) to be very interesting. The key findings of the manuscript are that clusters of Kv2.1 dynamically associate Cav1.2 and RyR, these associations result in discrete calcium signaling events (calcium sparks), and that Kv2.1 enhances Cav1.2 open probability. Although the functional significance of these clusters and their regulation remains speculative, the manuscript represents an important advance in our understanding of subcellular neuronal signaling and will be great of interest to the broader scientific community. We ask that you revise your manuscript to address the issues raised by the reviewers, as outlined below. Although we normally compile requests as essential or minor, in this case it may be helpful to see the reviewers remarks in the context of their reviews, and all reviewers have been clear about what is required and what is optional.

Reviewer #1:

The authors investigated the role of Kv2.1 in functional coupling of Cav1.2 and ryanodine receptors (RyR) to endow neurons with somatodendritic subcompartments capable of localized calcium signaling using a combination of antibody labeling with super-resolution microscopy, proteomics, in vitro calcium imaging, and patch clamp electrophysiology. While a nonconducting role for Kv2.1 in maintaining endoplasmic reticulum-plasma membrane junctions (EPJs) has been documented, the phenomenon's relevance to cellular function has generally been unclear. The key findings of the manuscript are that clusters of Kv2.1 dynamically associate Cav1.2 and RyR, these associations result in discrete calcium signaling events (calcium sparks), and that Kv2.1 enhances Cav1.2 open probability. The results look pretty solid, particularly as they were replicated in primary neuronal cultures, heterologous expression systems, and demonstrated in both rats and mice. Furthermore, the findings shed light on previous reports of localized RyR-mediated calcium events in acute hippocampal slice preparations. Although in the end, the functional significance of these clusters and their regulation remains speculative, the manuscript represents an important advancement in the understanding of subcellular neuronal signaling and will be great of interest to the broader scientific community.

1) There are inconsistencies in the quantitative rigor throughout the paper. In particular the lack of quantitative rigor for the results presented in Figure 1 is concerning. The majority of the results presented are strictly qualitative which is peculiar given the authors are equipped to measure colocalization. They do report that spatial distributions of Kv2.1 and Cav1.2 are correlated but is unclear exactly how this was ascertained other than using the analyze particles function of FIJI. Throughout the associated text, the authors make comparative statements "increased labeling in", less intense immunoreactivity", "comparable to". These statements should be bolstered by simple, unbiased image quantification and statistics. Furthermore, it is unclear why these experiments did not include an assessment Cav1.2 and ryanodine receptor spatial association in the Kv2.1-KO animal that the authors have access to and use throughout the paper. Quantification of spatial association in WT cultures and cultures from Kv2.1-KO animals is an important and simple experiment. Figure 2 also suffers similarly. Plots of colocalization like Figure 2K seem anecdotal. The use of CV as a relevant metric is not rationalized.

2) I am disappointed that the authors chose to convey the results of a key experiment, glutamate induced dispersal of Kv2.1 clusters, as a table as opposed to a figure. This seemingly crucial experiment demonstrates the ability of Kv2.1 to dynamically regulate, rather than an epiphenomenon of Cav1.2 and RyR clustering, and so it is unclear why it is relegated to a table. A possible answer is that the results are of marginal significance. If so, perhaps those data should be removed.

3) In a related concern, it would be beneficial to the interpretation of the experiment if the authors induced dispersal of Kv2.1 clusters with an alternative, preferably more direct, mechanism such as application of a PKA inhibitor. Given the expression of multiple metabotropic glutamate receptors in cultured hippocampal neurons, and the potential of metabotropic effects of NMDA receptor activation, the glutamate stimulation may not solely be acting through phosphorylation of Kv2.1. Although not essential to publication, this result would enhance the paper a lot.

4) In the results presented in Figure 3, the authors note large, synchronized calcium spikes associated with spontaneous action potentials. While the authors are likely correct that these represent a large increase in cytoplasmic calcium through action potential-associated voltage-gated calcium channels, this could be easily tested with the application of TTX. Furthermore, a key conclusion of the paper, that Cav1.2 and RyR associations "generate Ca^2+^ release events independently of action potentials" would be best demonstrated if the calcium sparks indeed persist in the presence of TTX. This experiment would also shed light on whether a neuron's spiking activity can affect frequency or amplitude of calcium sparks.

5) As with some other figures, the data in Figure 3E-G are presented anecdotally. E.g., the inset of 3E shows that Ca signals did not correspond to spontaneous depolarizations, but these are merely single traces. Do you infer that the apparent absence of a voltage signal associated with the spark implies that the spark is not being driven by a spatiotemporally localized Ca influx through LTCCs?

6) Figure 4 the black red and blue trace overlap so much one cannot tell anything from the data. For example, the red traces seem to show that thapsigargin did nothing.

Reviewer #2:

This study by Vierra et al. describes the novel observation that Kv2.1 channel form clusters that interact functionally with L-type Ca channels and ryanodine receptors. The authors show that CaV1.2 clusters increase in size with expression of Kv2.1, and that this effect occurs using a non-conducting mutant but is abolished in a non-clustering Kv2.1 mutant. Interestingly, they also find that co-expression of Kv2.1 and CaV1.2 (in HEK cells) increases the amplitude of whole-cell L-type Ca currents and leads to a leftward shift in the voltage-dependence of activation. In agreement with this finding, recordings in cultured hippocampal neurons from Kv2.1 KO mice show lower amplitude L-type Ca currents.

This is rigorous, well-executed study that presents important and original findings. The authors use an impressive number of tools and experimental approaches including immunohistochemistry, super resolution and TIRF imaging, electrophysiology, and mass-spec based proteomics. The data are all very high quality and the manuscript is well-written. The experimental results all support the main claims of the paper and the conclusions are all convincing.

1) The results shown in Figure 2E (effect of Kv2.1 non-clustering and non-conducting on CaV1.2 clusters) are quantified in terms of coefficient of variation (CV), but the rationale for this is not clearly described. Please address this. In addition to CV, the authors should also consider providing analysis/quantification of the absolute cluster size for each mutant tested as shown in Figure 2B.

2) The functional measurements of calcium sparks in Figure 3 and 5 show somewhat conflicting results. Results in Figure 5 show a reduction in spark amplitude with the non-conducting mutant, suggesting that conductance is necessary for increased amplitude. In Figure 3, however, the non-conducting Kv2.1 (P404W) mutant has no effect on amplitude of sparks (in comparison to the conducting Kv2.1). Why is this the case? Please discuss.

3) GCaMP3 and Cal-590 differ in their affinity for Ca^2+^, which may influence the apparent amplitude and size of the Ca sparks. This point may be helpful in addressing Comment 2. Along those lines, the spatial spread of the Ca signal (FWHM) is provided for the GCaMP3 images while the Cal-590 results are only provided as normalized values.

4) Along those lines, does expression of the Kv2.1 and the Kv2.1P404W mutants differ? This is relevant to GCamP3 fusion protein experiments. Please quantify.

5) The example CaV1.2 traces shown in Figure 6A show significant inactivation in the presence of Kv2.1. Was this generally true? Please address.

Reviewer #3:

In this paper by Vierra et al. the role for the potassium channel KV2.1 in the clustering and function of both voltage-gated calcium channels (VGCC) and ryanodine receptors (RyR) is explored in cultured neurons, cell lines, and tissues. The authors find that KV channels promote clustering of VGCC and RyRs. Furthermore, through imaging and ephys the authors show that this functional interaction lowers the activation threshold for calcium channels, promotes local calcium transients, and has a role in modulating the overall calcium signals in these cells. This tripartite interaction is proposed to occur specifically at PM-ER junction sites. In general, I find this paper well done and thorough. The experiments support the author's conclusions and I think it is a generally useful set of data for the field of cellular neuro-biology. I have some specific recommendations and comments that would improve the manuscript.

1) The zoom in Figure 1 (and most of the imaging based figures with small structures) makes it very difficult to evaluate the local clustering of these channels. As this concept is a key idea in the work (e.g. that these three proteins colocalize and functionally interact), can the authors please show more zoomed-in images, more direct measurements (plots) of the fluorescent signals, and less "wide-field" general views?

Also, some standard consistent quantitative way to evaluate co-localization would be helpful throughout the work. Something similar to a Pearson correlation co-efficient or similar type of analysis in most of the co-localization work should be presented across all relevant datasets in the figures.

2) In Figure 1D the KV signal from SIM looks very odd. Is this PM resident ER? Does a general ER marker have this same localization pattern?

3) In Figure 2 are the surface images here just single molecule-level KV? Specifically, is the "spottiness" of the KV images due to low labeling efficiency, low expression, or in fact due to clustering? Again, the size of the images makes it very difficult for me to evaluate the co-localization and cluster size of these signals.

4) The data in Figure 3—figure supplement 2 showing ER morphology at the PM is key. My primary question is how much of the KV and VGCC proteins are inside the ER and how much are exposed at the plasma membrane. Because the effects shown in this paper are due to channels on the plasma membrane surface, a clear separation of these two populations is really important to understand this work. In general, I would like to know how much are functional channels (PM localized channel) and how much are non-functional ER localized channels before they have reached the PM. The data in Figure 2F in many ways addresses this question, but I would like to see these data analyzed and presented in a much more robust way. For example, given the difference between Figure 2A and 2F in the calcium channel signal it appears that most of the clustered VGCC signal is in the ER and not at the surface.

5) A general cartoon model of this complex should be in the main text of the paper as a figure.

[Editors' note: further revisions were requested prior to acceptance, as described below.]

Thank you for resubmitting your work entitled "Kv2.1 mediates spatial and functional coupling of L-type calcium channels and ryanodine receptors in mammalian neurons" for further consideration by *eLife*. Your revised article has been reviewed by three peer reviewers and the evaluation has been overseen by Eve Marder as the Senior Editor, and Kenton Swartz as the Reviewing Editor.

The editors and reviewers think you have done an excellent job of revising the manuscript to address the comments of the reviewers. Two of the reviewers have additional suggestions concerning the Pearson CC analysis, and we ask that you address these remaining issues in revision.

Reviewer #1:

In this revision of Vierra et al. the authors have made an excellent effort to address reviewer concerns. A small specific concern is listed below.

Regarding the added Pearson CC: what is being tested here? Codistribution across the image, or colocalization of pixels? SIM images do not indicate a degree of colocalization expected based on the assertion in the crosslinking analysis of 12 A distances. In fact the SIM images do not clearly show superposition of RYR and Cav1.2, only RYR and nearby 1.3 clusters. Some comment here would help. The Materials and methods do not really discuss the statistical tests, unless I missed it.

Reviewer #3:

I am generally satisfied with the revisions to the paper that have been made. The paper is now improved. I still do have one outstanding issue that the authors might consider addressing.

1) The use of Pearson's correlation coefficient (PCC) analysis was added to only the first and last figure as far as I can tell. Why was this not used throughout the paper? That being said the PCC values, while different from controls, are very low (0.2). A positive control should be up around 0.7-1. Can the authors discuss these low values as they relate to the findings of the paper? This indicates that the vast majority of these channels in neurons are not co-clustered.

The most useful place for this analysis would be in Figure 3 and in Figure 3—figure supplements 1 and 2. Here, the PCC value would appear to be larger than in neurons but I can't tell because only raw line scans are shown and not PCC plots across multiple cells as a comparison.

In summary, I am trying to understand how common this complex is in both neurons and cultured cells. Do only a subset of ER/PM junctions have these complexes? If so how many? Are these functionally unique or different in some way? This would be useful to understand for this and future work.

---

## [Author Response]

Reviewer #1:[…] 1) There are inconsistencies in the quantitative rigor throughout the paper. In particular the lack of quantitative rigor for the results presented in Figure 1 is concerning. The majority of the results presented are strictly qualitative which is peculiar given the authors are equipped to measure colocalization. They do report that spatial distributions of Kv2.1 and Cav1.2 are correlated but is unclear exactly how this was ascertained other than using the analyze particles function of FIJI. Throughout the associated text, the authors make comparative statements "increased labeling in", less intense immunoreactivity", "comparable to". These statements should be bolstered by simple, unbiased image quantification and statistics. Furthermore, it is unclear why these experiments did not include an assessment Cav1.2 and ryanodine receptor spatial association in the Kv2.1-KO animal that the authors have access to and use throughout the paper. Quantification of spatial association in WT cultures and cultures from Kv2.1-KO animals is an important and simple experiment. Figure 2 also suffers similarly. Plots of colocalization like Figure 2K seem anecdotal. The use of CV as a relevant metric is not rationalized.

We thank the reviewer for pointing out these helpful ways to improve the reporting of our findings, and we have taken several steps to improve the quantitative rigor and presentation of the data. First, we have split Figure 1 of the original manuscript into two separate figures, now Figures 1 and 2 of the revised manuscript, and have added additional quantification of Kv2.1 and Cav1.2 protein localization and/or morphology to Figures 1, 2, and 3. We have also used more precise language in the text to clarify which parameters were quantified and which observations were qualitative (e.g., the general labeling pattern of Kv2.1, Cav1.2, and RyRs in cultured neurons as compared to pyramidal neurons in brain sections). We have added a discussion of the use of the CV of immunolabeling fluorescence intensity as a measure of channel clustering in the main body of manuscript. With regard to the analysis performed to assess the spatial distributions of Kv2.1 and Cav1.2, we have now updated the manuscript Results and Materials and methods sections to clarify that we used the Interaction Analysis function that is part of the MosaicSuite plugin for Fiji. We also appreciate the reviewer highlighting the important experiments with the Kv2.1 KO mice, which although present in the original manuscript, were presented as a table instead of paneled figure. Therefore, we have moved these data from the table in the original manuscript to scatter plots showing all values for each analysis, now part of Figure 11, to better highlight the analysis of Cav1.2 and RyR spatial association in WT and Kv2.1 KO neurons.

2) I am disappointed that the authors chose to convey the results of a key experiment, glutamate induced dispersal of Kv2.1 clusters, as a table as opposed to a figure. This seemingly crucial experiment demonstrates the ability of Kv2.1 to dynamically regulate, rather than an epiphenomenon of Cav1.2 and RyR clustering, and so it is unclear why it is relegated to a table. A possible answer is that the results are of marginal significance. If so, perhaps those data should be removed.

We agree with the reviewer that these results are of greater significance than we initially appreciated, especially after discussions with members of the ion channel research community while this manuscript was under review. Additionally, after determining that tetrodotoxin (TTX) treatment produces the opposite effect of glutamate by enhancing the clustering and spatial association of Kv2.1, Cav1.2, and RyRs, we have chosen to present the glutamate dispersal data and the new TTX data in Figure 1. We feel that the new graphs and representative images in Figure 1 will help the reader better assess the results of these experiments.

3) In a related concern, it would be beneficial to the interpretation of the experiment if the authors induced dispersal of Kv2.1 clusters with an alternative, preferably more direct, mechanism such as application of a PKA inhibitor. Given the expression of multiple metabotropic glutamate receptors in cultured hippocampal neurons, and the potential of metabotropic effects of NMDA receptor activation, the glutamate stimulation may not solely be acting through phosphorylation of Kv2.1. Although not essential to publication, this result would enhance the paper a lot.

We thank the reviewer for this excellent suggestion. While we (e.g., Misonou et al., 2004, 2005) and others (e.g., Aras et al., 2009) have previously demonstrated the involvement of calcineurin in glutamate-induced dephosphorylation and dispersal of Kv2.1 clusters, we agree that the pleiotropic effects of glutamate treatment could contribute to the observed dispersal of Cav1.2 clusters from RyR clusters. While we have not previously observed any effects of direct PKA inhibition on Kv2.1 clustering or phosphorylation (J. Trimmer, unpublished results), we reasoned that enhancing Kv2.1

phosphorylation and clustering might be another approach to evaluate the relationship between Kv2.1 phosphorylation/clustering and the of association somatic Cav1.2 with RyRs. Suppression of neuronal excitability with TTX produces enhanced Kv2.1 phosphorylation (Cerda and Trimmer, 2011) and clustering (Romer et al., 2019) and we determined here that it also increases the spatial association of Kv2.1, Cav1.2, and RyRs relative to controls. Thus, acute stimulation of neurons with glutamate causes dispersal of Kv2.1 and Cav1.2 from RyR clusters, while suppression of neuronal activity with TTX promotes the congregation of these proteins. These findings are now presented as graphs and exemplar images of neurons in Figure 1. While the precise mechanism remains unclear, we contend that these observations support our hypothesis that there is a close relationship between the degree of Kv2.1 clustering/phosphorylation and the coupling of Cav1.2 to RyRs at Kv2.1 clusters.

4) In the results presented in Figure 3, the authors note large, synchronized calcium spikes associated with spontaneous action potentials. While the authors are likely correct that these represent a large increase in cytoplasmic calcium through action potential-associated voltage-gated calcium channels, this could be easily tested with the application of TTX. Furthermore, a key conclusion of the paper, that Cav1.2 and RyR associations "generate Ca^2+^ release events independently of action potentials" would be best demonstrated if the calcium sparks indeed persist in the presence of TTX. This experiment would also shed light on whether a neuron's spiking activity can affect frequency or amplitude of calcium sparks.

We agree that the reviewer’s suggested experiment would provide greater insights into the relationship between action potentials and the localized Ca^2+^ release events. We have now included the results of an experiment in which we applied TTX to a neuron which displayed local Ca^2+^ release events at a GCaMP3Kv2.1 cluster and found that it had no impact on the generation of these Ca^2+^ signals. To better present the relationship between the global Ca^2+^ spiking activity of a neuron and the local Ca^2+^ release events, we have also included a temporal raster plot that depicts local Ca^2+^release events as well as the global Ca^2+^ influx events in selected neurons. While we do not have descriptive statistics of the relationship between local Ca^2+^ release events and global Ca^2+^ spikes the raster plots suggest that local Ca^2+^ release events occurred more frequently immediately following a global Ca^2+^ release event. The relationship between neuronal activity and other factors (e.g., activation of metabotropic glutamate receptors) that potentially influence these Ca^2+^ signals will be explored in more detail in future studies.

5) As with some other figures, the data in Figure 3E-G are presented anecdotally. E.g., the inset of 3E shows that Ca signals did not correspond to spontaneous depolarizations, but these are merely single traces. Do you infer that the apparent absence of a voltage signal associated with the spark implies that the spark is not being driven by a spatiotemporally localized Ca influx through LTCCs?

In presenting these data (now Figure 4E) in the revised manuscript we intended to convey that the local Ca^2+^ release events were not temporally aligned with obvious fluctuations in the whole-cell membrane potential. We did not mean to imply that there was not a spatiotemporally localized influx of Ca^2+^ through LTCCs. In contrast, all of our data suggest the opposite: that the Ca^2+^ release events occurring at Kv2.1 clusters required the spontaneous opening of a spatially coupled LTCC. Presumably these spontaneous openings of LTCCs do not have an influence on the *V*_m_ that is detectable in the whole-cell current clamp configuration. Similar results were presented by Manita and Ross in their initial description of similar Ca^2+^ signals in acute hippocampal slices (Manita and Ross, 2009). With regard to the small, non-AP fluctuations in the *V*_m_ apparent in Figure 4E, we suggest that they appear similar to the depolarizing membrane fluctuations corresponding to GABAergic IPSCs, for example as reported by Pitler and Alger (Pitler and Alger, 1992). However, in the absence of further data we can only speculate as to the nature and relationship between these small fluctuations in the *V*_m_ and the local Ca^2+^ signals at Kv2.1 clusters.

6) Figure 4 the black red and blue trace overlap so much one cannot tell anything from the data. For example, the red traces seem to show that thapsigargin did nothing.

We agree that the traces in what is now Figure 5 of the revised manuscript were difficult to see. We have revised these panels (now part of Figure 6 in the revised manuscript) to make the local Ca^2+^ release events more apparent by extending these traces vertically, slightly altering the shading and coloring of the overlaid traces, and adding colored dashed lines to each panel to denote the approximate maximum amplitude of the local Ca^2+^ release events, which unlike the global Ca^2+^ spikes do not exceed these dashed lines.

Reviewer #2:[…] 1) The results shown in Figure 2E (effect of Kv2.1 non-clustering and non-conducting on CaV1.2 clusters) are quantified in terms of coefficient of variation (CV), but the rationale for this is not clearly described. Please address this. In addition to CV, the authors should also consider providing analysis/quantification of the absolute cluster size for each mutant tested as shown in Figure 2B.

We thank the reviewer for noting the inadequate description we originally provided for the use of the CV metric to evaluate ion channel clustering. We have now included a better explanation in the second paragraph of the subsection “Kv2.1 organizes the localization of LTCCs”. We have also included a quantitation of Cav1.2 cluster size when coexpressed with each of the Kv2.1 isoforms, now shown in Figure 3J of the revised manuscript.

2) The functional measurements of calcium sparks in Figure 3 and 5 show somewhat conflicting results. Results in Figure 5 show a reduction in spark amplitude with the non-conducting mutant, suggesting that conductance is necessary for increased amplitude. In Figure 3, however, the non-conducting Kv2.1 (P404W) mutant has no effect on amplitude of sparks (in comparison to the conducting Kv2.1). Why is this the case? Please discuss.

We agree that these results are somewhat conflicting. While we do not have a detailed explanation for this observation, we hypothesize that the high input resistance of HEK293T cells relative to cultured neurons, which possess numerous endogenous ionic conductances (including native Kv2.1 and Kv2.2 channels), enabled K^+^ conductance through GCaMP3-Kv2.1_WT_ and –Kv2.1_S586A_ to promote membrane potential hyperpolarization in HEK293T cells, maintaining a greater electrochemical driving force for extracellular Ca^2+^ and also promoting recovery of Cav1.2 channels from voltage-dependent inactivation. We have now included this discussion in the last paragraph of the subsection “Kv2.1 augments LTCC and RyR2-mediated CICR reconstituted in HEK293T cells”. Considering the large (and growing) number of point mutations in and near the pore region of Kv2.1 associated with epilepsy and neurodevelopmental delay, it will be important for future studies to determine how altered K^+^ conductance at Kv2.1 clusters impacts Ca^2+^ signals at these sites.

3) GCaMP3 and Cal-590 differ in their affinity for Ca^2+^, which may influence the apparent amplitude and size of the Ca sparks. This point may be helpful in addressing Comment 2. Along those lines, the spatial spread of the Ca signal (FWHM) is provided for the GCaMP3 images while the Cal-590 results are only provided as normalized values.

We thank the reviewer for pointing out the different Ca^2+^ binding properties of GCaMP3 and Cal-590, and have now included a brief discussion of this fact in the third paragraph of the subsection “Neuronal Kv2.1 channels functionally associate with endogenous LTCCs and RyRs”. With respect to the presentation of normalized values, we chose to present normalized values as we felt it would assist the reader evaluate the magnitude of the effects in terms of the percent change in each parameter. However, in doing so we overlooked the stylistic inconsistency with the other figures in the manuscript. Therefore, we have replaced the normalized values with the raw values (Figure 6J of the revised manuscript).

4) Along those lines, does expression of the Kv2.1 and the Kv2.1P404W mutants differ? This is relevant to GCamP3 fusion protein experiments. Please quantify.

While we did not observe obvious differences in the expression of GCaMP3-Kv2.1_WT_ and GCaMP3Kv2.1_P404W_ in cultured hippocampal neurons, this was not rigorously tested in situ. However, we have now evaluated whether these two constructs are expressed at different levels in HEK293T cells, which while not primary neurons present a much more facile model to examine expression differences for heterologously expressed constructs. As now shown in Figure 4—figure supplement 1, we did not observe differences in the expression of these two constructs, as assessed by surface labeling of GCaMP3-Kv2.1 and also raw GCaMP3 fluorescence.

We also note that in mouse neurons loaded with Cal-590 that we observed no difference in the amplitude or width of localized Ca^2+^ signals in neurons lacking Kv2.1 – only signal frequency was depressed. We hypothesize that this result is a consequence of the decreased spatial association of Cav1.2 and RyRs, decreased RyR cluster size, and decreased open probability of LTCCs in Kv2.1 KO neurons. Although the Kv2.1 KO mouse neurons could express endogenous Kv2.2 channels, our data support a model which suggests that the impact of Kv2.1 on these spontaneous Ca^2+^ signals in neurons is mediated primarily through its structural role at ER-PM junctions, independent of Kv2.1’s K^+^ conducting function. While Kv2.1-mediated K^+^ conductance is clearly vital for normal neuronal development and function, the functional relationship between Kv2.1 clustering and its K^+^ conductance remains unclear.

5) The example CaV1.2 traces shown in Figure 6A show significant inactivation in the presence of Kv2.1. Was this generally true? Please address.

We thank the reviewer for suggesting that we examine inactivation of Cav1.2 when coexpressed with Kv2.1_P404w_. While we had reported that steady-state inactivation of the peak current was comparable between control cells and cells coexpressing Kv2.1, we did not assess changes in the sustained current. As now shown in Figure 7D of the revised manuscript, we observe greater inactivation of the sustained Cav1.2 current in the presence of Kv2.1. As this difference in inactivation is abolished by coexpression with STAC1 (Figure 7—figure supplement 1D), which interferes with CDI (Campiglio et al., 2018), the greater inactivation of Cav1.2 in the presence of Kv2.1 presumably reflects greater Ca^2+^-dependent inactivation, a result of the Kv2.1-induced potentiation of Cav1.2 activity.

Reviewer #3:[…] I have some specific recommendations and comments that would improve the manuscript.1) The zoom in Figure 1 (and most of the imaging based figures with small structures) makes it very difficult to evaluate the local clustering of these channels. As this concept is a key idea in the work (e.g. that these three proteins colocalize and functionally interact), can the authors please show more zoomed-in images, more direct measurements (plots) of the fluorescent signals, and less "wide-field" general views?Also, some standard consistent quantitative way to evaluate co-localization would be helpful throughout the work. Something similar to a Pearson correlation co-efficient or similar type of analysis in most of the co-localization work should be presented across all relevant datasets in the figures.

We agree that the reviewer’s recommendations would improve the manuscript and assist the reader in the evaluation of the results. Therefore, we have revised the figures as suggested by the reviewer. First, we have taken Figure 1 of the original manuscript and split it into two separate figures (now Figures 1 and 2 of the revised manuscript), have added more high-magnification images, and have increased the size of the original images throughout when feasible. In addition, we have included more quantitation of the imaging data: we now show colocalization analyses in Figure 1E and F, additional quantitation of protein immunolabeling in mouse brain sections in Figure 2C-E, and quantification of Kv2.1 and Cav1.2 protein localization and morphology in Figure 3C, D, H and J).

2) In Figure 1D the KV signal from SIM looks very odd. Is this PM resident ER? Does a general ER marker have this same localization pattern?

We thank the reviewer for highlighting the unusual (relative to most other PM proteins) but typical PM localization pattern of neuronal Kv2.1 channels. In neurons, the colocalization of surface Kv2.1 channels with its ER-resident binding partner VAPB has previously been demonstrated by the Tamkun group (Johnson et al., 2018). In addition, surface labeling of endogenous Kv2 channels in hippocampal slices with a Kv2-specific toxin (GxTX) conjugated to a fluorescent dye (Thapa et al., 2019) supports that these clusters are present on the PM. To better highlight this quality of Kv2 channels for the readers of this manuscript, we have now demonstrated that surface Kv2 channel clusters align very closely with ER resident proteins in HEK293T cells (Figures 3B and D of the revised manuscript), as we have done previously (Kirmiz et al., 2018). We suspect that the super-resolution SIM images (Figure 1G-I) and also the AiryScan confocal images (e.g., Figure 1C) better reveal the details of the unique PM localization of Kv2 channels.

3) In Figure 2 are the surface images here just single molecule-level KV? Specifically, is the "spottiness" of the KV images due to low labeling efficiency, low expression, or in fact due to clustering? Again, the size of the images makes it very difficult for me to evaluate the co-localization and cluster size of these signals.

We agree that the images we presented in the original manuscript were not optimal for evaluation of the ion channels in the PM. To address this point, we have included new and larger TIRF images of cells expressing these proteins throughout the figures of the revised manuscript. In addition, we have included additional experiments and related quantitation to address the “spottiness” highlighted above.

With regard to Kv2.1 channels, we now demonstrate that the Kv2.1 channels are expressed well at the PM, as indicated by the surface GxTX labeling of Kv2.1 now shown in Figure 3 and also Figure 4—figure supplement 1. We suspect the Kv spots to which the reviewer is referring are the high-density surface clusters of Kv2.1 that are much brighter than the surrounding diffuse Kv2 channels.

4) The data in Figure 3—figure supplement 2 showing ER morphology at the PM is key. My primary question is how much of the KV and VGCC proteins are inside the ER and how much are exposed at the plasma membrane. Because the effects shown in this paper are due to channels on the plasma membrane surface, a clear separation of these two populations is really important to understand this work. In general, I would like to know how much are functional channels (PM localized channel) and how much are non-functional ER localized channels before they have reached the PM. The data in Figure 2F in many ways addresses this question, but I would like to see these data analyzed and presented in a much more robust way. For example, given the difference between Figure 2A and 2F in the calcium channel signal it appears that most of the clustered VGCC signal is in the ER and not at the surface.

We thank the reviewer for raising an important point, and we agree that we are observing both surface and also sub-PM LTCCs, for example those localized to trafficking vesicles (Ghosh et al., 2018) and those retained within the ER as suggested by the reviewer. To address this point, we performed three additional experiments to obtain a better understanding of the relative distributions of surface and subPM channels. First, we performed surface immunolabeling of Kv2.1 channels using GxTX (Figure 3B and Figure 4—figure supplement 1), which demonstrates that the large Kv2.1 clusters are present in the PM. Next, we performed additional surface immunolabeling and quantitation of the Cav1.2 construct with an extracellular-facing HA tag (Obermair et al., 2004), and determined that approximately 70% of the immunolabeled Cav1.2-HA signal present in the TIRF field was surface localized. Thus, while we observed a Kv2.1-induced increase in surface Cav1.2-HA channel cluster size (Figure 3H), it is possible that we were also observing an increase in the recruitment of non-PM localized Cav1.2-HA channels (e.g., ER-retained channel). Therefore, to determine whether the change in Cav1.2 channel localization in the TIRF field was associated with a change in its function, we also performed recording of Ca^2+^ sparklets (Figure 8, Videos 7 and 8) to evaluate the single-channel activity of all active LTCCs present in the TIRF footprint. Using this approach, we found that Kv2.1 elevated the open probability of Cav1.2 (in agreement with our electrophysiological data), increased the number of active Cav1.2 channels present in the TIRF footprint, and reduced the nearest-neighbor distance of active Cav1.2 channels in relationship to each other (Figure 8J-L). Together, we believe these data are strong evidence in support of the model that Kv2.1 simultaneously reorganizes PM Cav1.2 and also influences its activity.

5) A general cartoon model of this complex should be in the main text of the paper as a figure.

We thank the reviewer for this suggestion and have included a model in Figure 11I of the revised manuscript.

[Editors' note: further revisions were requested prior to acceptance, as described below.]Reviewer #1:[…] Regarding the added Pearson CC: what is being tested here? Codistribution across the image, or colocalization of pixels? SIM images do not indicate a degree of colocalization expected based on the assertion in the crosslinking analysis of 12 A distances. In fact the SIM images do not clearly show superposition of RYR and Cav1.2, only RYR and nearby 1.3 clusters. Some comment here would help. The Materials and methods do not really discuss the statistical tests, unless I missed it.

We thank the reviewer for highlighting additional points of clarification regarding our image analyses.

For the added PCC values, we compared pixel intensities within the ROI (i.e., somatic immunolabeling). We chose this analysis for most of our images as we felt it was applied more consistently than an object based analysis relying on image segmentation. We also thank the reviewer for pointing out an important observation regarding the SIM images their relationship to the results of the proteomic analysis of crosslinked proteins IPd with Kv2.1. Indeed, the super-resolution images indicate that Kv2.1 and Cav1.2 or Cav1.3 immunolabeling did not co-occur within the same pixels (and suggesting that they are beyond 12 Å reach of the DSP crosslinker). Provided that the immunolabeling is revealing most or all of the somatic LTCCs and Kv2.1 channels (i.e., there is not a population of LTCCs and Kv2.1 channels that overlap yet are not immunolabeled perhaps due to a lack of antibody binding), we hypothesize that the IP of LTCCs with Kv2.1 may be a result of cross-linked intermediary protein serving to “daisy-chain” these proteins together. We have now better highlighted this possibility in the Discussion.

We have also now included additional clarification in the manuscript regarding the spatial analysis we performed to assess the localization of Kv2.1 and LTCCs:

“For these super-resolution images, we performed an object-based analysis (rather than a pixel intensity correlation-based measurement such as PCC) to determine whether the localization of somatic Kv2.1 and LTCCs were co-dependent. The approach we used relied on evaluation of the nearest-neighbor distances (NND) of Kv2.1 and Cav1.2 or Cav1.3 cluster centroids and a comparison of these values to the predicted NNDs if Kv2.1 and LTCCs were randomly distributed (Shivanandan et al., 2013, Helmuth et al., 2010).”

The test statistic from the Interaction Analysis test (part of the Mosaic plugin) was computed from *K* (we used *K*=1000) Monte Carlo samples corresponding to the point distributions of randomly distributed Kv2.1 and LTCC clusters (i.e., randomized images corresponding to the null hypothesis of “no interaction”), and then ranking the actually observed distance distribution against the randomly distributed samples. We rejected the null hypothesis of “no interaction” based on the rank of the observed distribution compared to the ⌈(1 - *α)K*⌉-th test statistic, using an *α=*0.05. A detailed explanation of the statistical method used in this Fiji plugin is provided in (Helmuth et al., 2010).

Reviewer #3:I am generally satisfied with the revisions to the paper that have been made. The paper is now improved. I still do have one outstanding issue that the authors might consider addressing.1) The use of Pearson's correlation coefficient (PCC) analysis was added to only the first and last figure as far as I can tell. Why was this not used throughout the paper? That being said the PCC values, while different from controls, are very low (0.2). A positive control should be up around 0.7-1. Can the authors discuss these low values as they relate to the findings of the paper? This indicates that the vast majority of these channels in neurons are not co-clustered.The most useful place for this analysis would be in Figure 3 and Figure 3—figure supplements 1 and 2. Here, the PCC value would appear to be larger than in neurons but I can't tell because only raw line scans are shown and not PCC plots across multiple cells as a comparison.

We thank the reviewer for raising important points regarding the images analyses in the paper. The rationale for evaluating the PCC values of images of single cells (we did not attempt these analyses on images of neurons in brain sections) is described above in our response to reviewer 1. We have now included additional analyses of the PCC values of Kv2.1 and LTCCs in Figure 3E, Figure 3—figure supplement 1E, and Figure 3—figure supplement 2E. We agree that the PCC values from neurons indicate that the majority of endogenous Kv2.1 and LTCC immunolabeling does not co-cluster, and have now highlighted this in the text of the Results. However, our data indicate that while the absolute PCC values are low, that they reflect a biologically relevant subset of functionally associated Kv2.1, LTCCs, and RyRs whose function is apparent from our live cell imaging results. The spatial association of these channels can be pharmacologically enhanced or dispersed (by TTX or glutamate, respectively), and genetically disrupted by the loss of Kv2.1.

The PCC values of heterologously expressed Kv2.1 and LTCCs in HEK293T cells were higher than was observed for native channels in neurons. While the molecular basis of the greater colocalization of heterologously expressed channels is not clear, we hypothesize that it may be related to the much larger abundance of LTCC channels relative to Kv2.1 when heterologously expressed. Thus, it may be that the affinity of LTCCs for a Kv2.1-mediated ER-PM junction is not extremely high (as suggested by the relatively low PCC values from neurons indicating that only a subset of endogenous channels associated), but by heterologously expressing the channels we are much better able to see the interaction between Kv2.1 and LTCCs. Future studies are aimed at understanding the molecular mechanisms underlying the spatial association of Kv2.1 and LTCCs.

In summary, I am trying to understand how common this complex is in both neurons and cultured cells. Do only a subset of ER/PM junctions have these complexes? If so how many? Are these functionally unique or different in some way? This would be useful to understand for this and future work.

The reviewer raises an important point regarding the relative abundance of ER-PM junctions containing the Kv2.1-LTCC-RyR complexes in neurons. ER-PM junctions are abundant on neuronal somata (occupying approximately 10% of the somatic PM (Wu et al., 2017)) and can possess a variety of protein constituents, including the proteins studied here, and also proteins such as STIM/Orai, E-Syts, junctophilins, etc. As Kv2.1 clusters intrinsically represent ER-PM junctions by nature of their formation (i.e., via interaction with the ER-resident VAPs), our observation that localized Ca^2+^ signals occurred only at a subset of GCaMP3-Kv2.1 clusters (e.g., see Figure 4A and Video 1) suggests that only a subset of Kv2.1 clusters possess the molecular machinery required to generate these Ca^2+^ signals. The reason for this is unclear but we suspect it might underlie the physiological functions of these particular Kv2.1 clusters/EPJs (for example, Ca^2+^-dependent release of signaling molecules such as neuropeptides is known to be associated with Kv2.1 clusters in neuroendocrine cells). Future work is focused on identifying the physiological functions of these Ca^2+^ signals, which we hope will identify why only a subset of the Kv2.1 clusters appear to exhibit these Ca^2+^ signals. Moreover, LTCCs and RyRs can exist at sites distinct from Kv2-containing ER-PM junctions, including other classes of ER-PM junctions. We have now included a brief discussion of how the Kv2.1 clusters relate to other EPJs present in neurons (subsection “Properties of Ca2+ 642 sparks at Kv2.1-associated EPJs”, second paragraph